# Prioritized mass spectrometry increases the depth, sensitivity and data completeness of single-cell proteomics

R. Gray Huffman[1], Andrew Leduc[1], Christoph Wichmann[2], Marco Di Gioia [3], Francesco Borriello[3], Harrison Specht[1], Jason Derks [1], Saad Khan[1], Luke Khoury [1], Edward Emmott [1,4], Aleksandra A. Petelski[1,5], David H. Perlman[6], Jürgen Cox [2], Ivan Zanoni [3] & Nikolai Slavov [1,5] ✉

Major aims of single-cell proteomics include increasing the consistency, sensitivity and depth of protein quantification, especially for proteins and modifications of biological interest. Here, to simultaneously advance all these aims, we developed prioritized Single-Cell ProtEomics (pSCoPE). pSCoPE consistently analyzes thousands of prioritized peptides across all single cells (thus increasing data completeness) while maximizing instrument time spent analyzing identifiable peptides, thus increasing proteome depth. These strategies increased the sensitivity, data completeness and proteome coverage over twofold. The gains enabled quantifying protein variation in untreated and lipopolysaccharide-treated primary macrophages. Within each condition, proteins covaried within functional sets, including phagosome maturation and proton transport, similarly across both treatment conditions. This covariation is coupled to phenotypic variability in endocytic activity. pSCoPE also enabled quantifying proteolytic products, suggesting a gradient of cathepsin activities within a treatment condition. pSCoPE is freely available and widely applicable, especially for analyzing proteins of interest without sacrificing proteome coverage. Support for pSCoPE is available at http://scp.slavovlab.net/pSCoPE.

Macrophages are innate immune myeloid cells performing diverse functions in development, tissue homeostasis and immune response. Despite this diversity, macrophages are traditionally described in terms of dichotomous states (M1, pro-inflammatory; M2, anti-inflammatory). Single-cell measurements, however, have revealed a more complex and continuous spectrum of macrophage polarization in terms of molecular and functional phenotypes[1–3]. Thus, we sought to explore this continuum of polarized states in primary macrophages using single-cell

mass spectrometry (MS). Shotgun MS methods can analyze hundreds of single cells per day and quantify thousands of proteins but remain biased toward abundant proteins[3–11]. This bias reflects an intentionally programmed 'topN' heuristic for selecting the $n$ most abundant peptide precursors for sequence identification and quantification[12], as illustrated in Fig. 1a.

Peptide selection by the topN heuristic is limited by three challenges: (1) abundance bias, which limits the dynamic range of quantified

[1]Departments of Bioengineering, Biology, Chemistry and Chemical Biology, Single Cell Center and Barnett Institute, Northeastern University, Boston, MA, USA. [2]Computational Systems Biochemistry Research Group, Max Planck Institute of Biochemistry, Martinsried, Germany. [3]Boston Children's Hospital and Harvard Medical School, Boston, MA, USA. [4]Centre for Proteome Research, Department of Biochemistry and Systems Biology, University of Liverpool, Liverpool, UK. [5]Parallel Squared Technology Institute, Watertown, MA, USA. [6]Merck Exploratory Sciences Center, Merck Sharp and Dohme Corp., Cambridge, MA, USA. ✉e-mail: nslavov@northeastern.edu

**Fig. 1 | Introducing prioritization to MaxQuant.Live increases identification consistency and protein coverage. a**, Shotgun topN analysis selects the *n* most abundant precursors for isolation and fragmentation (shown in blue). Among the many detected precursors, prioritized analysis first selects the ones with highest priority (shown in solid red) and then from lower-priority tiers (shown with decreasingly saturated red tones). Prioritization can also selectively allocate increased fill times to high-priority peptides of low abundance, as shown in the second cycle of $MS^2$ scans. **b**, Prioritized analysis increases the consistency of peptide identification over default MaxQuant.Live operation for high-priority peptides while also increasing protein coverage per run. The box plots showing

proteins identified per experiment contain six points per analysis method, one for each experiment. **c**, Rates of $MS^1$ detection and $MS^2$ analysis for prioritized precursors from all tiers of the benchmarking experiments displayed in **b**. These box plots contain six points per analysis method, one for each experiment. The fourth panel displays peptides used for retention time (RT) alignment only and not intended for $MS^2$ scans. For all box plots, whiskers display the minimum and maximum values within 1.5 times the interquartile range of the 25th and 75th percentiles, respectively; the 25th percentile, median and 75th percentile are also featured.

proteins; (2) stochasticity, which limits data completeness across single cells; and (3) unidentifiable precursors, the analysis of which wastes instrument time and limits proteome coverage[13]. Such inefficient use of time is particularly limiting for single-cell proteomics due to the long ion-accumulation times needed to sequence and quantify each precursor[3,14]. While no existing method resolves all three challenges, the challenges can be partially mitigated. For example, targeted MS can alleviate challenges (1) and (2) but has remained limited to analyzing hundreds of peptides or fewer[15–23]. Real-time database searching can increase the fraction of sequenced peptide features and alleviate challenge (3), but it has not allowed for selecting peptides of interest[7,24]. Targeting peptides from inclusion lists with real-time retention-time alignment ameliorates challenge (1) but faces a tradeoff between maximizing coverage (and thus duty cycle usage) and maximizing data completeness[25,26]. To simultaneously address all three challenges, we introduce a multi-tiered precursor-selection strategy, dubbed 'prioritization', illustrated in Fig. 1a.

## Results

Prioritized analysis aims to simultaneously maximize the consistency of peptide analysis, proteome coverage and instrument time utilization. To achieve these aims, we built upon the real-time retention-time alignment of MaxQuant.Live[26,27] and introduced priority levels that define the temporal order of peptide analysis. Prioritization aims to maximize data completeness when the duty cycle time is insufficient for

analyzing all peptides from the inclusion list, the precursors of which are detected in survey scans. Two example duty cycles implementing this selection logic are displayed in Fig. 1a.

### Increasing proteome coverage and data completeness

The logic of prioritized peptide acquisition is implemented via new functionality in MaxQuant.Live software that seeks to maximize both data completeness and proteome coverage (Fig. 1a). To maximize proteome coverage, a large inclusion list of previously identified precursors allows filling each and every duty cycle with peptide-like features most likely to be identified. To simultaneously improve data completeness and proteome depth, sets of high-priority precursors are supplied and always given priority for $MS^2$ analysis over precursors from lower-priority levels. The sets of high-priority precursors can be selected based on biological interest, ease of identification, spectral purity or other relevant metrics. Increased accumulation times can be allocated for them, such as for the high-priority peptides in the second duty cycle of Fig. 1a. This increased accumulation time should increase the number of ion copies sampled per $MS^2$ analysis[28,29].

To benchmark the benefits of prioritization, MaxQuant.Live was used to acquire data with and without prioritization enabled while keeping all other parameters constant (Fig. 1b). To reduce sample-related variability, we analyzed injections from a bulk sample diluted to single-cell levels. The inclusion list was composed of the same precursors for the prioritized and non-prioritized analyses by MaxQuant.

Live: over 11,500 precursors selected to be identifiable, along with a comparable number of precursors used only for retention-time calibration. The precursors on the inclusion list were then stratified into three levels of priority by the confidence of their identification and spectral purity in previous analyses. More confidently identified and less co-isolated peptides were assigned to the higher-priority levels. Data completeness for the high-priority group of 4,000 peptides increased to 72% when using prioritization, compared to 49% without prioritization. The fraction of peptides identified in 100% of the six runs at a false discovery rate (FDR) of 1% was 18% without prioritization and 59% with prioritization (Fig. 1b), representing a 228% increased consistency for prioritization. This increased consistency of identification did not impede protein coverage, as prioritization increased the number of quantified proteins per experiment at 1% FDR (Fig. 1b). Consistent with the precursor-selection logic shown in Fig. 1a, prioritization sent precursors to MS$^2$ scans according to priority: 97% of the 4,000 high-priority peptides were sent for MS$^2$ analysis and lower fractions of the lower-priority levels (Fig. 1c). By contrast, MaxQuant.Live without prioritization sent similar fractions (about 63%) of peptides from all lists for MS$^2$ analysis as shown in Extended Data Fig. 1b.

We then applied prioritized Single-Cell ProtEomics (pSCoPE) to single human cells to evaluate the depth and the consistency of proteome coverage relative to shotgun analysis (Fig. 2). We prepared single-cell samples from embryonic kidney 293 (HEK) and melanoma cells using nano-proteomic sample preparation (nPOP)[30,31]. These samples were analyzed either by shotgun or by prioritized methods using 60-min active chromatographic gradients and narrow (0.5-Th) isolation windows for MS$^2$ scans. The narrow precursor isolation window resulted in good quantitative agreement between different peptides originating from the same protein (Extended Data Fig. 2a,b). To maximize coverage, we prioritized peptides that were previously identified with high confidence and low co-isolation; more details can be found in Methods. Relative to shotgun analysis, pSCoPE increased the fraction of MS$^2$ spectra assigned to a confident peptide sequence by over twofold, reaching 84% (Fig. 2a). The remaining 16% of MS$^2$ spectra correspond to sequences having lower confidence of identification in previous experiments used for generating the inclusion list (Extended Data Fig. 3). The increase in productive MS$^2$ scans doubled the number of unique peptides per run (increased by 103%) and increased the number of quantified proteins per single cell by 106% (Fig. 2a).

In addition to increasing the depth of proteome coverage, the prioritized approach increased the dynamic range of proteins quantified in single cells (Fig. 2b). This is enabled by replacing the intensity-based precursor-selection logic of the topN algorithm with a priority-based precursor-selection logic. This allowed for successful prioritization of lower-abundance precursors and resulted in quantifying peptides spanning a wider range of abundances as shown in Fig. 2b. This wider dynamic range includes low-abundance ranges not covered by the shotgun data at all. The median precursor intensity of peptides quantified by pSCoPE is 2.5-fold lower than the shotgun median (Fig. 2b). The abundances of the precursors from each priority level are shown in Supplementary Fig. 1. Furthermore, the high probability with which pSCoPE sends prioritized precursors for MS$^2$ analysis resulted in MS$^2$ scans for 98% of the peptides on a 1,000-peptide high-priority level, 90% of the 4,475 peptides from the top two priority levels and 71% of the 8,621 peptides on the top three priority levels (Extended Data Fig. 5). These efficiencies were achieved while pSCoPE maximized the number of MS$^2$ scans, thus demonstrating the ability of priority levels to increase the probability of analyzing peptides while also increasing the total number of precursors sent for MS$^2$ scans.

Prioritization also increased data completeness (Fig. 2c,d). This increase is particularly pronounced for challenging peptides, as exemplified with a set of 1,000 peptides identified with less than 50% probability in shotgun SCoPE2 sets (Fig. 2c). Prioritizing these peptides in eight representative runs increased data completeness by 171% compared to controlled shotgun analyses (Fig. 2c). The gains in data completeness extended to peptides from all priority levels, albeit at a lower gain of approximately 16%. pSCoPE also increased data completeness at the protein level, reaching 93% for all proteins, which represents a 34% gain over the performance of shotgun analysis (Fig. 2d).

To put these gains in context, we compared them to other methods for increasing data completeness, such as isobaric match between runs (iMBR)[32]. iMBR facilitated the identification and quantification of approximately 170 additional precursors per shotgun experiment compared to 2,595 additional precursors per experiment added by prioritization (Extended Data Fig. 4). In contrast to prioritization, the decrease in missing data afforded by iMBR cannot be applied to peptides pre-selected based on biological considerations.

The pSCoPE sets used for benchmarking performance in Fig. 2 included single cells from two different cell lines and thus allowed us to examine protein variation both between and within each cell line. These cell types were clearly separated by the first principal component (PC) of the data (Fig. 2e). Protein set enrichment analysis (PSEA) performed on the PCs identified enrichment for multiple functional sets of proteins, including those associated with glycolysis and the G2/M transition of the mitotic cell cycle, as shown in Fig. 2e.

## Quantification accuracy of pSCoPE

Next, we benchmarked the quantitative accuracy of pSCoPE by comparing the mixing and measured ratios for a set of peptides spiked in at known levels (Fig. 3). These peptides contained internal trypsin cleavage sites to control for digestion variability. They were spiked into eight single-cell pSCoPE sets using a randomized design detailed in Supplementary Table 1. The spike-in levels were chosen and confirmed to span the abundance range of the peptides quantified in the single cells (Fig. 3a). Furthermore, their abundances covered a 16-fold dynamic range across five spike-in levels. The measured abundances exhibited linear dependence with the spike-in levels, with a slope of 1.06 and a coefficient of determination $R^2 = 0.97$ (Fig. 3b). These results indicate that pSCoPE is able to quantify peptides in single-cell sets with good accuracy and precision.

## Polarized proteome states

Next, we used pSCoPE to explore the molecular and functional heterogeneity of murine bone marrow-derived macrophages (BMDMs) responding to inflammatory stimuli, such as lipopolysaccharide (LPS), the major component of gram-negative bacteria's outer membrane. Macrophages were differentiated using macrophage colony-stimulating factor (M-CSF) and either treated with LPS for 24 h or left untreated. Single macrophage cells were prepared for MS analysis by nPOP[30] and were first analyzed by SCoPE2 (refs. 3,33). Peptides identified with high confidence and exhibiting high variability across the single cells were then prioritized for analysis by pSCoPE, using longer accumulation times and narrow isolation windows, as detailed in the Methods. pSCoPE quantified 1,123 proteins across 373 single primary macrophage cells, achieving 71% data completeness for all proteins (Extended Data Fig. 6) and good quantitative agreement between peptides originating from the same protein (Extended Data Fig. 2c). The PC-analysis (PCA) projection of the data resulted in two clusters corresponding to the treatment conditions (Fig. 4a). Projected bulk samples clustered with the corresponding treatment groups, indicating that the cluster separation reflects treatment response[33]. This treatment-specific clustering is also reflected in the abundance of proteins that vary across treatments but not within a treatment, as exemplified by proteins functioning in the type 1 interferon (IFN)-mediated signaling pathway (Fig. 4a).

The spread of the macrophage clusters along PC1 suggests that proteins varying across treatment groups may also vary within a treatment group (Fig. 4a). Indeed, PSEA using the PCA loadings identified protein sets enriched within PC2, which captures variability within

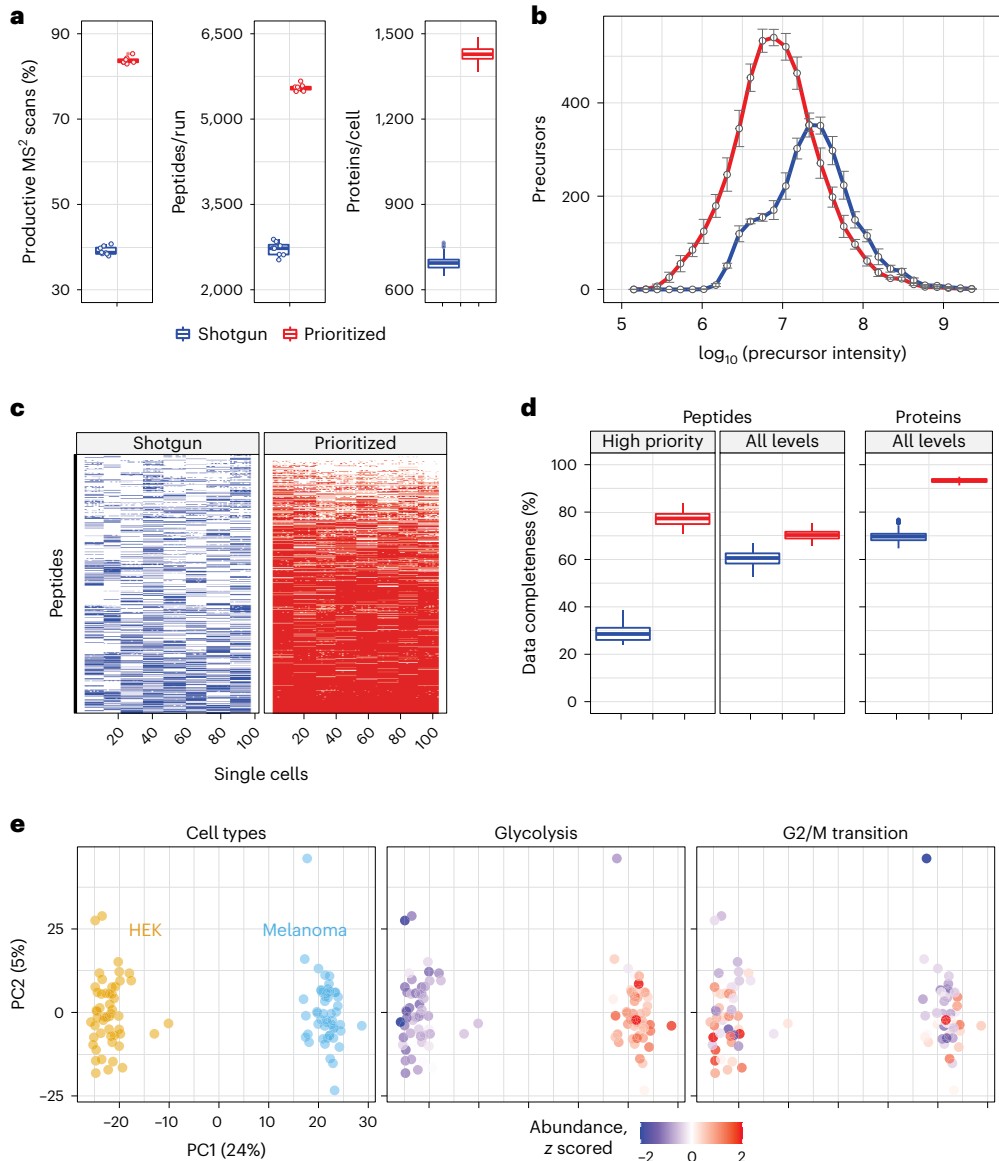

**Fig. 2 | Prioritization increases proteome coverage, sensitivity and data completeness of single-cell protein analysis. a**, Relative to shotgun analysis, prioritization increased the fraction of MS[2] scans assigned to peptide sequences ($n = 8$ experiments per box plot), the number of peptides per run (60-min active gradient; $n = 8$ experiments per box plot) and the number of quantified proteins per single cell ($n = 97$ single cells per box plot). **b**, Prioritized analysis enables increased sensitivity and dynamic range while analyzing more peptides than shotgun analysis with matched parameters. $n = 8$ experiments per analysis method. Data are represented by the median, and error bars denote s.d. Precursor abundances stratified by priority level are displayed in Supplementary Fig. 1. **c**, A heatmap showing data completeness across single cells (columns) for 1,000 peptides (rows) from the top priority tier. **d**, Prioritized analysis increases data completeness at both peptide and protein levels across all priority tiers. $n = 194$ and 206 single cells analyzed by shotgun and prioritization, respectively. **e**, PCA of the single cells associated with **b** cluster by cell type. Protein sets enriched in the PCs are visualized by color coding the single cells by the median protein abundance of the set in each cell. All experiments used 60-min active gradients per run and 0.5-Th isolation windows for MS[2] scans. All peptide and protein identifications were filtered at 1% FDR with additional filtration criteria detailed in Methods. For all box plots, whiskers display the minimum and maximum values within 1.5 times the interquartile range of the 25th and 75th percentiles, respectively; the 25th percentile, median and 75th percentile are also featured.

a condition (Fig. 4b). These sets included proton transport and the phagosome-maturation pathway, both of which were found to be among the most variable protein sets within each condition (Fig. 4a). These finding were recapitulated when PCA was performed without data imputation (Extended Data Fig. 7), suggesting robustness to choices of data analysis[34]. Additionally, color coding each cell in the PCA by its corresponding data completeness indicates that the data completeness is spread evenly across the PCA and does not drive cell clustering (Extended Data Fig. 8).

Next, we directly examined the levels of proteins from gene ontology (GO) groups that changed upon LPS treatment (such as type I IFN signaling) and those that did not change upon treatment but varied within condition (such as proton transport; Fig. 4c). The LPS-treated/untreated fold changes estimated from bulk samples and from single cells correlated positively ($\rho = 0.91$), supporting the accuracy of pSCoPE measurements and affirming that proton-transport proteins do not change much upon LPS treatment. To bolster the confidence that proton-transport proteins vary within a condition, we examined

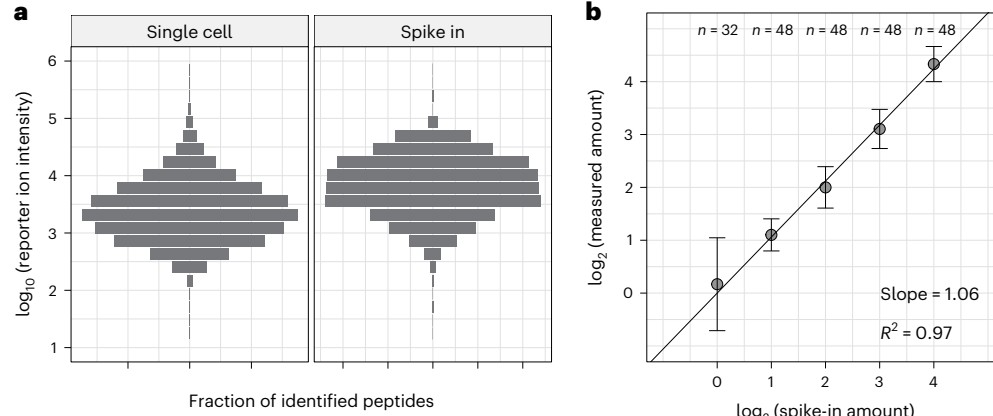

**Fig. 3 | Evaluating quantitative accuracy and precision of pSCoPE with peptide standards. a**, Reporter ion intensities for precursors identified at 1% FDR from single cells and from the spike-in peptides, which were dispensed into the single-cell samples across a 16-fold range. **b**, Normalized reporter ion intensities for all tryptic products from the spike-in peptides plotted against their spike-in amounts, with regression slope and goodness of fit displayed.

Points denote the median, and error bars denote s.d. of the distribution of normalized reporter ion intensities for each spike-in level. The data in **a** and **b** come from eight prioritized experiments, and the numbers of data points for each of the spike-in levels are indicated on the top. The experimental design for this set of analyses can be found in Supplementary Table 1.

their covariation within both LPS-treated and untreated macrophages (Fig. 4d). Significant covariation ($\rho = 0.5$, $P < 10^{-12}$) suggests that the variation of these proton-transport proteins is concerted within a treatment condition and reflects a biological gradient. This conclusion is further supported by the correlation among all detected proteins from the V-ATPase holoenzyme, shown in Supplementary Fig. 3.

To systematically investigate proteome variations within a condition, we performed PCA of each treatment group separately and PSEA on the associated PCA of protein loading. Remarkably, the first PCs of the treated and untreated macrophages correlated strongly ($r = 0.8$, $P < 10^{-15}$), suggesting that within-condition protein variability is similar across the two conditions. This observation is naturally reflected in very similar functional enrichment results for treated and untreated macrophages (Fig. 5a).

These results suggest that 24-h LPS treatment does not fundamentally alter the axes of protein variation for murine BMDMs, such as phagosome maturation, proton transport and protein targeting to the membrane (Fig. 5a). In addition to these shared functional groups, some protein sets vary only within the LPS-stimulated cells, as illustrated by proteins annotated with regulation of inflammatory response, antigen processing and presentation via major histocompatibility class (MHC) II and regulation of translational initiation (Fig. 5a). The coherence of protein variability within functional groups suggests that it is functionally relevant[35], but it does not prove it.

**Connecting protein variation to functional variation**

To examine whether the observed protein heterogeneity has functional consequences, we sought to directly measure the endocytic activity of macrophages and its relationship to such protein heterogeneity. To this end, we measured the uptake of fluorescently labeled dextran and macrophages sorted by flow cytometry from the top and bottom deciles of the fluorescence distribution for analysis by LC–MS/MS (Extended Data Fig. 9). Both the LPS-treated (Fig. 5b) and the untreated (Extended Data Fig. 10) macrophages exhibited large variance in dextran uptake per cell, with the median uptake being higher for LPS-treated cells. The proteomes of subpopulations sorted based on their dextran uptake were analyzed by data-independent acquisition (DIA), which allowed us to identify proteins for which the abundance was significantly different between the most and least endocytically active cells (Fig. 5b and Extended Data Fig. 10). Next, for each cell, we estimated the median abundances of these proteins associated with

endocytic activity and correlated them with the PCs for each treatment condition. For the LPS-treated samples, the proteins associated with high dextran uptake (such as mannose receptor C type 1 (MRC1), stabilin 1 (STAB1) and sorting nexin 17 (SNX17)) were found to be significantly correlated with PC1, while the proteins annotated with low dextran uptake were inversely correlated with PC1 and significantly correlated with PC2 (Fig. 5c). Notably, some proteins (such as MRC1, STAB1 and CD74) exhibit similar association with dextran uptake both in the untreated and LPS-treated macrophages (Fig. 5b and Extended Data Fig. 10).

To measure regulatory mechanisms more directly, we sought to quantify proteolysis, which plays major functional roles in macrophage activation[36–38]. To avoid products of proteolysis that may occur during sample preparation, we focused only on proteolytic products present in the macrophages before trypsin digestion. These products were identified in discovery bulk samples in which amine groups were covalently labeled before trypsin digestion as commonly performed[39,40]. The peptide fragments were matched to annotated proteolytic products in the MEROPS database[41] and analyzed by pSCoPE in single cells. To evaluate the single-cell quantification, the fold changes between LPS-treated and untreated cells were compared to the corresponding bulk estimates (Fig. 6a). The good agreement between the measurements from established bulk methods and pSCoPE supports the accuracy of the single-cell quantification.

To infer the functional association of the validated proteolytic products, we correlated their single-cell abundances to the abundances of pro- and anti-inflammatory protein panels (Fig. 6b): (1) proteins that we identified as differentially abundant between bulk samples of untreated or LPS-treated macrophages and (2) previously reported markers for M1 or M2 macrophages[42]. The cathepsin D-cleaved actin peptide (L104) and the cathepsin E-cleaved citrate synthase peptide (H25) were found to be significantly positively correlated with inflammatory markers. Both peptide fragments annotated with cathepsin D cleavage at L299 were inversely correlated with the set of proteins that were more abundant in LPS-treated BMDMs.

Having established the reliability of single-cell quantification of proteolytic products across conditions, we next examined their abundance within a treatment condition (Fig. 6c). The data indicate that the actin proteolytic products exhibit significant variability within each treatment condition. For example, the actin fragment cleaved at L299 correlates significantly with PC1 (Spearman $r = -0.32$, $P < 2 \times 10^{-5}$;

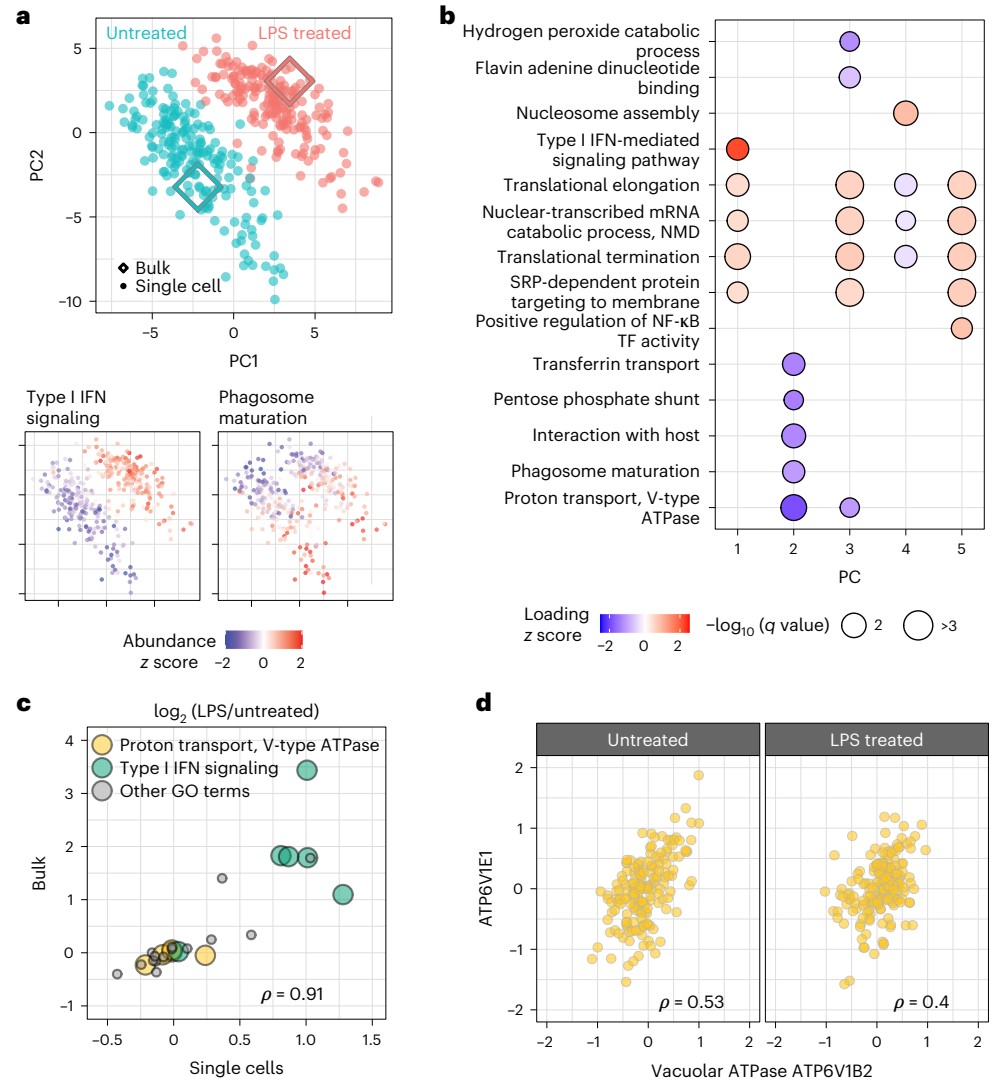

**Fig. 4 | Prioritized analysis of primary macrophages identifies protein variation within and across treatment conditions. a**, PCA of 373 BMDMs and 1,123 proteins color coded by treatment condition. Diamond markers indicate bulk samples projected in the same low-dimensional space as the single cells. The adjoining PCA plots are color coded by the z-scored median relative abundance of proteins corresponding to type I IFN-mediated signaling and phagosome maturation. Performing this analysis without imputation recapitulates these results, as shown in Extended Data Fig. 7. **b**, Protein groups identified by PSEA performed using the PC vectors with protein weights from the PCA shown in **a**. NF-κB, nuclear factor κB; TF, transcription factor; NMD, nonsense-mediated mRNA decay; SRP, signal recognition particle. **c**, The protein fold changes (LPS-treated/untreated macrophages) were estimated both from single cells and from bulk samples. The corresponding estimates correlate positively, with a Spearman correlation of $\rho = 0.91$ computed using all 28 proteins shown ($P = 2 \times 10^{-11}$); all GO terms are labeled in Supplementary Fig. 2. **d**, Proteins functioning in proton transport do not change across conditions but covary within a condition (across 177 single cells). Correlations between all vacuolar ATPase proteins annotated with proton transport are displayed in Supplementary Fig. 3. In **c** and **d**, $\rho$ denotes Spearman correlations, and all associated q values are <$10^{-7}$.

Fig. 6c). These results point to the possibility of using pSCoPE for analyzing proteolytic activity at single-cell resolution.

## Discussion

Our analysis demonstrates the potential of prioritized data acquisition to simultaneously optimize multiple aspects of single-cell proteomics, including the consistency, sensitivity, depth and accuracy of protein quantification (Figs. 1–3). These gains are achieved using multiplexed and widely accessible workflows[33] and a new software module that is freely available. The performance gains by pSCoPE demonstrate the potential of innovations in data acquisition to drive single-cell proteomics[13]. pSCoPE enabled us to quantify molecular and functional diversity of primary macrophages, even of post-transcriptionally modified peptides (Fig. 6). This analysis of post-transcriptionally

modified peptides is enabled by (1) the ability of pSCoPE to selectively send even lowly abundant precursors for MS$^2$ analysis and (2) the narrow isolation windows that reduce the co-isolated signal for such peptides. We expect these methodological benefits to generalize to both single-cell and bulk samples from diverse biological problems[43].

Many MS methods allow for analyzing a pre-selected group of peptides. They range from targeted methods that maximize sensitivity and probability of target quantification[18–23,44] to directed methods that use inclusion lists[23,44]. Some directed methods[25,26] can be used in hybrid mode with a small inclusion list and shotgun analysis (scan while idle). This hybrid mode can be seen as a single level of priority, but it suffers from the low identification rate for MS$^2$ scans acquired in shotgun mode. pSCoPE extends the directed family of methods by

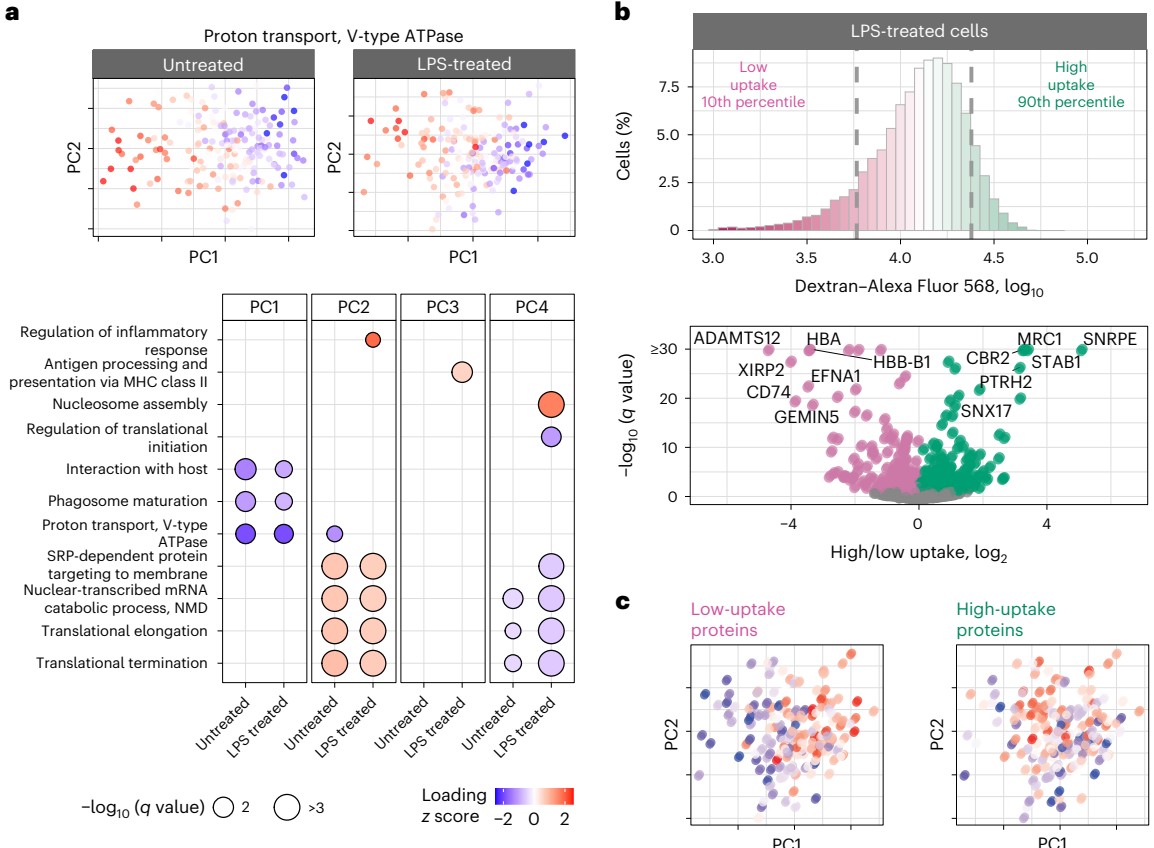

**Fig. 5 | Axes of proteome polarization are similar between untreated and LPS-treated macrophages and correlate with dextran uptake. a**, Untreated and LPS-treated macrophages were analyzed separately by PCA, and PSEA was performed on the corresponding PCs. PCA plots are color coded by the median abundance of proteins annotated with proton transport. **b**, The uptake of fluorescent dextran by LPS-treated macrophages was measured by flow cytometry, and the cells with the lowest and highest uptake were isolated

for protein analysis. The volcano plot displays fold changes for differentially abundant proteins and the associated statistical significance. The corresponding analysis for untreated macrophages is displayed in Extended Data Fig. 10. **c**, LPS-stimulated macrophages were displayed in the space of their PCs and color coded by the median abundance of low-uptake or high-uptake proteins. The low-uptake proteins correlate with PC1 (Spearman $r = 0.55$, $q \leq 3 \times 10^{-15}$), and the high-uptake proteins correlate with PC2 (Spearman $r = 0.33$, $q \leq 2 \times 10^{-5}$).

introducing a generalized tiered approach that allows the prioritization of thousands of peptides for isolation and fragmentation (thus achieving 96% success rate of sending high-priority peptides for MS²) while maximizing the number of analyzed precursors and thus achieving high proteome coverage (Fig. 2). The multi-tiered approach allowed for high identification rates of MS² scans from all priority levels (including the lowest priority level) for an average sequence assignment of 84% at 1% FDR even when using 0.5-Th isolation windows. This prioritized algorithm (Fig. 1) introduced here may also be implemented with other approaches for performing real-time retention-time alignment[45,46] and may be extended to single-cell proteomics multiplexed by non-isobaric mass tags[47].

Prioritized analysis increases the flexibility of experimental designs. For example, it makes precursors selected for quantification less dependent on the composition of the isobaric carrier, which can be particularly advantageous when the carrier material does not perfectly match the analyzed single cells. In such cases, pSCoPE can be used to analyze the relevant proteins even if they are not among the most abundant proteins in the isobaric carriers used. As a second example, prioritization may allow using different collision energies to analyze the same precursor[26]. The combined spectra may improve the localization of peptide modifications and reporter ion release[48]. Applying this approach to selected challenging peptides can improve their analysis without consuming much time or reducing the overall proteome coverage. As a third example, pSCoPE may be used to prioritize not only

peptides but also metabolites and thus enable multi-modal analysis of proteins and peptides in the same single cells.

We measured protein covariation of functionally related proteins within primary macrophages not only between treatment groups but also within a treatment group, as shown in Fig. 4a for phagosome-maturation proteins. The proteins exhibiting such within-condition variability are similar for treated and untreated macrophages (Fig. 5a). This similarity in protein covariation is remarkable because LPS treatment substantially remodels the proteome, and yet protein covariation remains similar for treated and untreated macrophages. A possible explanation for this finding is that protein covariation reflects the topology of regulatory interactions[35], and many of these regulatory interactions remain similar between untreated and LPS-treated macrophages. This interpretation is consistent with the observation that the proteins associated with dextran uptake are similar for the two conditions, as shown in Fig. 5b and Extended Data Fig. 10. Additionally, prioritized analysis enabled the quantification of a proteolytically modified cytoskeletal protein, the cleavage of which is significantly correlated with inflammatory stimulus across single-cell samples. The robustness of the results to choices of data analysis, such as different treatments of missing data (Extended Data Fig. 7), bolsters their validity[34].

Our prioritization approach is broadly accessible as the software is free and compatible with the Thermo Fisher Q Exactive series, Orbitrap Exploris and Orbitrap Eclipse (Supplementary Fig. 4). The newer

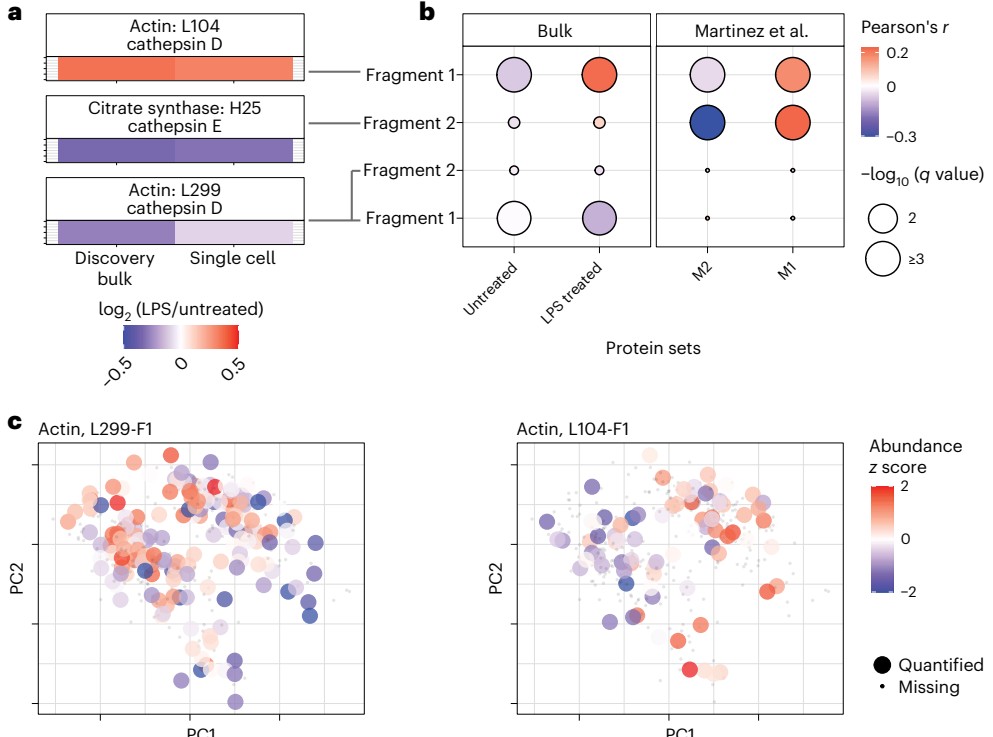

**Fig. 6 | Proteolytic products in individual macrophages correlate with inflammatory markers and vary within treatment groups. a**, A comparison between untreated and LPS-treated ratios of proteolytic products quantified in discovery bulk experiments and in single cells. Annotations are derived from the MEROPS database[41]. **b**, Correlation analysis of proteolytic products with treatment group-specific and macrophage-polarization-specific protein panels. **c**, Data from the untreated and LPS-treated cells were projected by PCA and color coded by the relative abundance of the indicated actin fragments.

instruments have quadruples that are likely to substantially improve the efficiency of isolating ions with narrow isolation windows (0.5 Th) for MS[2] scans and thus achieve higher sensitivity and precision of quantification than those demonstrated here with a Q Exactive classic instrument.

Prioritization can help increase the throughput of single-cell proteomics by enabling consistent analysis of proteins of interest on short chromatographic gradients[14]. All results presented here used 60-min active gradients, although shorter gradients may increase both the throughput and the sensitivity (via narrower elution peaks) while still affording enough time to analyze thousands of prioritized peptides[28]. Thus, pSCoPE may provide accurate and consistent protein quantification across many single cells to support sufficiently powered biological investigations of primary cells and tissues[14,49].

## Online content

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

## Methods

### Implementation of prioritized analysis

To maximize the probability of analyzing high-priority peptides (that is, peptides of high experimental importance) when operating at full duty cycle, we developed a new feature of MaxQuant.Live[26]: multi-tier prioritization. Multi-tier prioritization uses the real-time instrument control capabilities of MaxQuant.Live and adds a priority feature that determines which precursors are analyzed when duty cycle time becomes limiting. The initial priority for each peptide is a user-defined integer number that is by default set to zero. By assigning non-zero values, it is possible to prioritize a single set of peptides or to implement a multi-tier approach, depending on the experimental objectives. During data acquisition, the peptides are selected for fragmentation based on their priority. After each fragmentation event, the corresponding peptide priority value is decremented unless fragmentation occurred outside of the retention-time tolerance. The prioritization feature is part of the latest release of MaxQuant.Live (version 2.1), available at http://MaxQuant.Live/ and http://scp.slavovlab.net/pSCoPE.

The initial user-defined priorities are set via a column in the inclusion list table. This column was added to allow for easy definition of priority for every peptide on the list. The higher the integer number associated with a peptide (and thus its priority level), the higher the probability that it will be chosen for fragmentation when duty cycle is limited. MaxQuant.Live was tested on a Q Exactive (as described below), but it was written to be compatible with all Orbitrap instruments.

### Prioritization workflow

All prioritized single-cell experiments followed the four stages of the workflow displayed in Supplementary Fig. 4 and described below.

(1) Compilation of proteins of interest from literature or prior LC–MS/MS analyses. Detailed information regarding the construction of inclusion lists used in the analysis of the standardized quality-control (SQC) samples, HEK and melanoma samples or BMDM samples can be found in Prioritized inclusion list construction, below.

(2) DIA analysis of a 1× concentrated injection of the combined carrier–reference sample to generate accurate retention times for precursors, which will subsequently be prioritized.
   - This step is enabled by using a spectral library generated from prior DIA analysis of a 5–10× concentrated injection of the combined carrier–reference samples.

(3) Assignment of precursors identified in step 2 to priority levels based on proteins of interest defined in step 1.
   - The minimal set of precursor characteristics needed for a prioritized inclusion list are the mass, expected apex retention time and priority.

(4) Acquire data from SCoPE samples using MaxQuant.Live's prioritization feature and the inclusion list generated in step 3.
   - Performing a test run on a 1× injection of the combined carrier and reference samples can be useful for troubleshooting methods before acquiring data from single cells.

### Benchmarking MaxQuant.Live with and without prioritization enabled

These experimental sets, the results of which are presented in Fig. 1, were designed to benchmark the performance of prioritization against MaxQuant.Live's default global targeting mode with respect to consistency of peptide identification across experiments, as well as protein coverage. The experiments presented in Fig. 1 are a matched set of six experiments acquired via MaxQuant.Live's default global targeting mode and six experiments acquired with prioritization enabled. The parameters for experiments that directly compared MaxQuant.Live's default operation and prioritized analysis were identical, including LC

gradients and data-acquisition parameters. Additional information regarding sample preparation, instrument parameters, MaxQuant.Live parameters, prioritized inclusion list design, analysis of raw data, single-cell data processing and figure generation can be found in the respective sections. The active gradient in all experiments was 60 min.

### Comparing prioritized and shotgun analyses

These experimental sets, the results of which are presented in Fig. 2, were designed to assess the relative performance of shotgun and prioritized methods with respect to sequence coverage and consistency of quantification across single-cell samples. The experiments presented in Fig. 2a are a matched set of eight shotgun analyses and eight prioritized analyses; the experiments presented in Fig. 2b–e are a matched set of eight shotgun analyses and eight prioritized analyses. The parameters for experiments that directly compared shotgun and prioritized analyses were identical, including LC gradients and data-acquisition parameters with the only exception of increasing fill times for selected prioritized precursors as explicitly described in the main text. Additional information regarding sample preparation, instrument parameters, MaxQuant.Live parameters, prioritized inclusion list design, analysis of raw data, single-cell data processing and figure generation can be found in the respective sections. The active gradient in all experiments was 60 min.

### BMDM samples prepared by nPOP

The experiments in Figs. 4–6 were designed to present a use case for prioritized LC–MS/MS methods. Twenty shotgun and 40 prioritized single-cell experiments containing samples from both treatment conditions (untreated or treated for 24 h with LPS) were conducted as part of this module. A side-by-side comparison of the 20 shotgun experiments and the first 20 prioritized experiments can be found in Extended Data Fig. 6. Only the results of the 40 prioritized analyses were included in Figs. 4–6. Additional information regarding sample preparation, instrument parameters, MaxQuant.Live parameters, prioritized inclusion list design, analysis of raw data, single-cell data processing and figure generation can be found in the respective sections. The active gradient in all experiments was 60 min.

### Endocytosis experiments, BMDM samples

For Fig. 5, to identify protein sets associated with endocytosis that were specific to murine BMDMs, bulk samples from each treatment condition (untreated or treated for 24 h with LPS) were incubated with fluorescently labeled dextran, and samples from the top and bottom deciles of dextran uptake were isolated by flow cytometry for downstream LC–MS/MS analysis. Protein sets found to be differential between dextran-uptake deciles were then added to the high-priority level in subsequent prioritized analyses of single-cell BMDM samples. Additional information regarding sample preparation, instrument parameters, raw data analysis and differential protein detection can be found in the respective sections.

### MEROPS experiments, BMDM samples

For Fig. 6, bulk BMDM samples from each treatment condition (untreated or treated for 24 h with LPS) were lysed, cysteine residues were reduced and alkylated, and samples were incubated with tandem mass tag (TMT)pro so that all pre-digestion N termini would be distinguishable from neo-N termini produced by subsequent tryptic digestion. The raw LC–MS/MS data were then searched with a FASTA database containing all murine SwissProt-reviewed sequences, as well as semitryptic peptides consistent with MEROPS-annotated proteolytic cleavage sites. These experiments were used to validate semitryptic MEROPS-annotated peptides observed in the prioritized single-cell samples. Additional information regarding sample preparation, MEROPS database integration, instrument parameters and data analysis can be found in the respective sections.

## Bulk BMDM sample analyses by data-dependent acquisition and data-independent acquisition

Bulk BMDM samples from each treatment condition (untreated or treated for 24 h with LPS) were lysed, digested and labeled with TMTpro for data-dependent acquisition (DDA) analysis as a duplex sample or sequentially analyzed as labeled single-condition samples via DIA. These experiments were used to identify differentially abundant proteins between the treatment conditions, which were then added to the high-priority level in subsequent prioritized analyses of single-cell BMDM samples. Additional information regarding sample preparation, instrument parameters, raw data analysis and differential protein detection can be found in the respective sections.

## BMDM samples prepared via minimal ProteOmic sample Preparation (mPOP) methods

This set of experiments represents an early troubleshooting investigation to both assess the sizes of the BMDMs from each treatment condition by using cellenONE's optical system (Scienion) and contrast against data generated in a prior set of single-cell BMDM samples isolated by flow cytometry that may have experienced sorting issues. The results from this set were not used to generate any of the publication figures and are included merely for completeness, as a subset of identifications from these experiments informed the inclusion list construction of the nPOP-prepared pSCoPE sets. Additional information regarding sample preparation, MEROPS database integration, instrument parameters and raw data analysis can be found in the respective sections.

## Liquid chromatography–mass spectrometry platform

The LC–MS/MS equipment and setup used for all analyses are detailed in the SCoPE2 protocol[33]. Briefly, samples were separated via online nLC on a Dionex UltiMate 3000 UHPLC; 1 µl of sample was loaded onto a 25-cm × 75-µm IonOpticks Aurora Series UHPLC column (AUR2-25075C18A); MS analyses were performed with a Thermo Scientific Q Exactive mass spectrometer; an Active Background Ion Reduction Device (ESI Source Solutions) was used at the ion source to remove background contaminants. In the LC separations, buffer A was 0.1% formic acid in LC–MS-grade water, and buffer B was 80% acetonitrile, 0.1% formic acid in LC–MS-grade water; all buffer B percentages described in the subsequent instrument methods are relative to this concentration.

Instrument methods used in this study can be found in Supplementary Tables 3 and 6. Chromatographic methods used in this study can be found in Supplementary Tables 4 and 7.

## MaxQuant.Live parameters

Instrument method parameters followed the MaxQuant.Live listening scan guidelines at https://maxquantlive.readthedocs.io/en/latest/: two full MS–SIM scans were applied from minutes 25 to 30 to trigger MaxQuant.Live. Both MS–SIM scans had the following parameters in common: resolution of 70,000, AGC target of $3 \times 10^6$ and a maximum injection time of 300 ms. The first MS–SIM scan covered the scan space from 908 Th to the total acquisition time plus 1,000 min; for a total acquisition time of 95 min, the upper bound of the scan range would be 1,070 Th (95 min minus the initial 25 min before acquisition was triggered, plus 1,000 min). The second MS–SIM scan covered the scan space from 909 Th to the $m/z$ corresponding to the MaxQuant. Live method index to call.

MaxQuant.Live parameters used for each sample analysis can be found in Supplementary Tables 11–14.

## Cell culture

**Culturing melanoma cells.** Melanoma cells (WM989-A6-G3, a kind gift from S. Shaffer, University of Pennsylvania) were cultured in TU2% medium, composed of 80% MCDB 153 (Sigma-Aldrich, M7403), 10% Leibovitz's L-15 (Thermo Fisher, 11415064), 2% FBS (MilliporeSigma, F4135), 0.5% penicillin–streptomycin (Thermo Fisher, 15140122) and

1.68 mM calcium chloride (Sigma-Aldrich, 499609). Cells were passaged at 80% confluence in T75 flasks (MilliporeSigma, Z707546) using 0.25% trypsin–EDTA (Thermo Fisher, 25200072).

**Culturing HEK293 cells.** HEK293 cells (CRL-1573, ATCC) were cultured in DMEM, supplemented with 10% FBS (MilliporeSigma, F4135) and 1% penicillin–streptomycin (Thermo Fisher, 15140122). Cells were passaged at 80% confluence in T75 flasks (MilliporeSigma, Z707546) using 0.25% trypsin–EDTA (Thermo Fisher, 25200072).

**Culturing and collecting U937 cells.** U937 cells (CRL-1593.2, ATCC) were grown as suspension cultures in RPMI medium (HyClone 16777-145) supplemented with 10% FBS (MilliporeSigma, F4135) and 1% penicillin–streptomycin (Thermo Fisher, 15140122). Cells were passaged when a density of $10^6$ cells per ml was reached.

U937 cells were collected by pelleting, before washing with 1× PBS at 4 °C. Washed cell pellets were diluted in 1× PBS at 4 °C, and cell density was estimated by counting at least 1,000 cells using a hemocytometer. Cells that were collected for the SQC sample were resuspended in water (Optima LC/MS Grade, Fisher Scientific, W6500).

**Collecting melanoma and HEK cells.** Before collection, medium was removed from cell cultures, which were then rinsed with 0.25% trypsin–EDTA (Thermo Fisher, 25200072) at 4 °C. After rinsing, adherent cultures were incubated with 0.25% trypsin–EDTA at 4 °C (Thermo Fisher 25200072) for 15 min, until cells were detached from the culture vessel. Cold 1× PBS was added to each culture vessel, and the resulting suspension was pelleted by centrifugation at 250g, before being washed with 1× PBS and repelleted at 250g. Washed cell pellets were diluted in 1× PBS at 4 °C, and their density was estimated by counting at least 1,000 cells using a hemocytometer. Cells that were collected for carrier, reference and SQC samples were resuspended in water (Optima LC/MS Grade, Fisher Scientific, W6500) and frozen at −80 °C. Cells that were collected for single-cell sorting on the cellenONE system were diluted in 1× PBS to a concentration of 300 cells per µl and placed on ice.

**Culturing and collecting BMDMs.** C57BL/6J (JAX, 000664) mice were purchased from Jackson Laboratory. BMDMs were differentiated from bone marrow in DMEM (Thermo Fisher Scientific), 30% L929-M-CSF supernatant and 10% FBS. After 7 d, BMDMs were replated at $1 \times 10^6$ cells per ml in DMEM supplemented with 10% FBS, and each plate was either stimulated for 24 h with LPS (serotype O55:B5, Enzo Life Sciences) at 1 µg ml$^{-1}$ or allowed to rest. Before collection, cells were washed twice with 1× PBS and incubated with PBS–2 mM EDTA to detach from the plate. Cells were then centrifuged at 300g for 5 min and washed with 1× PBS before being resuspended. Washed cell pellets were diluted in 1× PBS at 4 °C, and their density was estimated by counting at least 1,000 cells using a hemocytometer. Cells that were collected for carrier and reference samples were resuspended in water (Optima LC/MS Grade, Fisher Scientific, W6500) and frozen at −80 °C. Cells that were collected for single-cell sorting on the cellenONE system were diluted in 1× PBS to a concentration of 300 cells per µl and placed on ice.

## Spike-in peptide selection

Spike-in peptides were used to benchmark the accuracy and precision of reporter ion quantitation in single-cell analyses. These spike-in peptides were selected on the basis of ionizability and identifiability from the search results of an LC–MS/MS analysis of a yeast standard sample.

A DDA analysis of a TMT-labeled yeast standard sample, m13306. raw, was downloaded from MassIVE (MSV000084263) and searched with MaxQuant (version 1.6.7.0). Trypsin/P was selected as the enzyme, and TMT (+224.152478 Da) was enabled as a variable modification on lysines and peptide N termini. All other settings were kept as default. The *Saccharomyces cerevisiae* reference proteome was downloaded

from UniProt and used as the sequence database for this search (uniprot-organism-yeast.fasta).

The evidence.txt file from the MaxQuant search results was imported into the R environment. Peptides containing methionine, glutamine and asparagine were removed from the search results, and peptides less than nine amino acids and greater than 11 amino acids were removed. Peptides in the 25th percentile of the posterior error probability (PEP) distribution and the 75th percentile of precursor intensity and score distributions were selected for further analysis. Peptide sequences present in the human proteome (swis-sprot_human_20181210.fasta) were also filtered out. Four tryptic sequences (AYFTAPSSER, VEVDSFSGAK, TSIIGTIGPK and ELYEVDVLK) from the filtered search results were then selected such that their retention times differed by more than a minute, subjecting them to different groups of co-eluting peptides. These four sequences were then grouped into two pairs, and the pairs were concatenated into single sequences (AYFTAPSSERVEVDSFSGAK and TSIIGTIGPKELYEVDVLK) for synthesis by JPT Peptide Technologies. These concatenated sequences were then used as trypsin-cleavable spike-in peptides to benchmark reporter ion quantitation.

### Sample preparation

**Standards used for evaluating prioritization.** To provide a controlled comparison of MaxQuant.Live's default global targeting method and the prioritized sample-analysis method shown in Fig. 1, a standardized TMT-labeled QC sample was used (hereafter abbreviated as an 'SQC sample').

Serially diluted bulk samples were multiplexed as previously described[3,33]. Each injection of this TMT-labeled sample contained one 50-cell-level carrier channel per cell type (U937, 126C; HEK293, 127N), three single-cell-level channels per cell type (U937, 128C, 129C, 130C; HEK293, 129N, 130N, 131N), one half-cell-level channel per cell type (U937, 131C; HEK293, 132N) and one quarter-cell-level channel per cell type (U937, 132C; HEK293, 133N).

**Single-cell samples.** All single-cell samples were prepared using the droplet nPOP method as detailed in refs. 30,31. In addition to sorted single cells, the SCoPE sets contained negative control samples to be used for downstream quality-control purposes. These negative control samples received all reagents and proceeded through all sample-handling steps, but no single cells were dispensed into these droplets[33]. The distribution of protein-level coefficients of variation (CVs) (that is, quantification variability) associated with the single-cell and control samples for these experiments can be found in Extended Data Fig. 2a–d.

**HEK293 and melanoma single-cell sample preparation.** A ~200-cell carrier and a ~5-cell reference composed of HEK293 and melanoma cell lines were prepared following the guidelines of the SCoPE2 protocol[33]. In addition to serving as the carrier and reference for all single-cell sets prepared by nPOP[31] which were analyzed in the technical section, the combined carrier and reference sample was used in all spectral library-generation and retention-time-calibration experiments for the coverage and consistency experiments shown in Fig. 2.

**Spike-in peptide preparation for HEK293 and melanoma single-cell sample preparation.** AYFTAPSSERVEVDSFSGAK and TSIIGTIGPKELY-EVDVLK were ordered from JPT Peptide Technologies and resuspended at a concentration of 2.5 mM in LC–MS-quality water for storage at −20 °C. Spike-in concentrations of these two peptides were then examined empirically to determine a spike-in level at which their tryptically digested fragments were readily detectable at MS[1], and the associated reporter ion intensities (when serially diluted across a 16-fold range) spanned the full dynamic range of endogenous peptide reporter ion intensities in single-cell samples. The lowest spike-in level was then denoted as the '1×' concentration. For the carrier and reference

samples, spike-in peptides were then added at 400× and 20× concentrations per set, respectively, so that they were 100-fold and fivefold more abundant than the median spike-in level of 4×. For the 14 single-cell and control samples that were part of each SCoPE set, both spike-in peptides were serially diluted, dried down in a speed vacuum and resuspended in 30 µl of LC–MS-quality DMSO, such that each peptide at the following concentrations was added to the indicated number of samples per set: 1× (two samples), 2× (three samples), 4× (three samples), 8× (three samples), 16× (three samples). To achieve this addition, five DMSO aliquots containing the spike-in peptides were dispensed to form the 14-droplet clusters of each nPOP-prepared SCoPE set before cell dispensing; each droplet contained 8 nl of its respective spike-in dilution in DMSO. Spike-in amounts were randomized relative to TMT labels and cell types. Except for the addition of spike-in peptides, the single-cell samples were prepared as detailed in refs. 30,31. Specifically, a single cell was added to each spike-in-containing DMSO droplet and then digested, labeled and quenched. The samples for each SCoPE set were pooled and transferred into the well of a 384-well plate that was loaded into the autosampler for LC–MS/MS analysis.

**BMDM single-cell nPOP sample preparation.** Carrier and reference samples composed of equivalent amounts of untreated and LPS-stimulated murine BMDMs were prepared following the SCoPE2 protocol[29,33], such that the carrier was composed of ~200 cells and the reference was composed of approximately five cells. This sample design was then used in the preparation of single-cell sets by nPOP[30,31], as well as in the generation of spectral libraries and retention-time-calibration experiments for the experiments shown in Figs. 4–6 as well as Extended Data Figs. 6–8.

The LPS-treated (24 h) and the untreated cells were combined within each SCoPE set. The majority (87%) of the labeled sets also contained negative control samples for quality-control purposes. These control samples received all reagents and proceeded through all sample-handling steps, but no single cells were dispensed into these droplets. The distribution of protein-level CVs (that is, quantification variability) associated with the single-cell and control samples for these experiments can be found in Extended Data Fig. 2d.

**BMDM single-cell mPOP sample preparation.** Carrier and reference samples composed of equivalent amounts of untreated and LPS-stimulated murine BMDMs were prepared following the SCoPE2 protocol[29,33], such that the carrier was composed of ~200 cells and the reference was composed of approximately five cells. This sample design was then used in the preparation of single-cell sets by mPOP[50], in which single cells from each condition (untreated and treated with LPS for 24 h) were sorted into a 384-well plate (Thermo, AB1384) with the cellenONE liquid-handling system (Scienion). The mixed carrier and reference sample was also used in the generation of retention-time estimate runs for the set of ten samples analyzed by pSCoPE.

**Endocytosis assay samples.** To facilitate an analysis of functional heterogeneity in the single-cell BMDM samples, markers of endocytic competency were identified from bulk analyses of untreated and LPS-treated (24 h) BMDMs isolated by flow cytometry on the basis of dextran uptake.

**BMDM endocytosis assay.** Murine BMDMs were differentiated and divided into treatment groups, as indicated previously, and incubated with dextran conjugated to Alexa Fluor 568 (Thermo, D22912) at a final concentration of 0.5 mg ml⁻¹ for 45 min at 37 °C. After the incubation period, cells were washed twice with 1× PBS and incubated with PBS–2 mM EDTA to detach from the plate. Before flow cytometry analysis, cells were centrifuged at 300g for 5 min and washed with 1× PBS before being resuspended. Using a Sony MA900 cell sorter, dextran–AF568 fluorescence in the PE–Texas Red channel was then

analyzed for cells from each treatment condition, and a minimum of 70,000 cells from the top and bottom ~10% of the PE–Texas Red fluorescence distribution were then sorted for downstream sample preparation and MS analysis.

**Preparation of endocytosis assay samples for LC–MS/MS analysis.** Each sample isolated by flow cytometry was lysed using a freeze–heat cycle as part of mPOP[50]. After lysis, approximately 70,000 cells worth of lysate was digested for 12 h at 37 °C using 11 ng μl⁻¹ trypsin gold and 150 mM triethylammonium bicarbonate in 65 μl. Samples were then stage tipped[51], and ~10,000 cells worth of digest was injected in 0.1% formic acid for analysis by MS via DIA method 5 using DIA gradient 4, detailed below.

### MEROPS bulk validation experiments, BMDMs
To validate proteolytically regulated substrates detected in single-cell BMDM samples, LC–MS/MS analyses were performed on bulk samples prepared using a workflow previously applied to the identification and quantification of viral protease cleavage products[52].

Murine BMDMs were differentiated, divided into treatment groups and collected, as indicated previously. Samples initially contained 125,000 BMDMs in 62.5 μl of LC–MS water (Optima LC/MS Grade, Fisher Scientific, W6500). SDS (Sigma, L3771-100G) and HEPES (Thermo Fisher Scientific, AAJ63218AE) were added to final concentrations of 1% and 0.1 M, respectively. cOmplete Protease Inhibitor (Roche, Sigma-Aldrich 05892791001) was then added to a 2× final concentration. Samples were then heated to 95 °C for 5 min and subsequently chilled at −80 °C for 10 min. Benzonase (1 U, Millipore, Sigma-Aldrich, E1014-25KU) was added and allowed to incubate at room temperature for 30 min. DTT (500 mM, Pierce, Thermo Fisher, A39255) was added to a final concentration of 15 mM and allowed to incubate for 30 min. Iodoacetamide (Pierce, Thermo Fisher, A39271) was added to a final concentration of 15 mM and incubated at room temperature in the dark for 30 min. DTT was then added a second time to a final concentration of 15 mM and incubated for 1 h. SP3 beads (Cytiva, Fisher Scientific, 09-981-123; Cytiva, Fisher Scientific, 09-981-121) were prepared and mixed following manufacturer recommendations.

Prepared SP3 beads (2.5 μl, 100 μg μl⁻¹) were added to each of the four samples. LC–MS-grade water (17.3 μl) was added to each tube, resulting in a total volume of 141 μl. Ethanol (564 μl, 200 proof, HPLC–spectrophotometric grade, Sigma, 459828-1L) was added to each sample and incubated for 18 min. Samples were then incubated for 5 min on a magnetic stand, the supernatant was removed, and the beads were washed twice with 400 μl of 90% ethanol, after which the remaining supernatant was removed.

Each sample was resuspended in 22.5 μl of 6 M GuCl (Sigma, G-3272), 30 μl of 0.5 M HEPES, pH 8 and TCEP (10 mM final concentration) (Supelco, MilliporeSigma, 646547). Samples were then incubated for 30 min at room temperature. TMTpro (57 μl, Thermo, A44520) at 8 ng μl⁻¹ was then incubated in each sample for 1.5 h, with the untreated condition being labeled with 127C and the LPS-treated condition being labeled with 128N. Samples were then quenched with 6 μl of 1 M Tris (Thermo Fisher, AM9855G) for 45 min. Following quenching, 1.2 μl of SP3 beads (100 μg μl⁻¹) was added to each TMT-labeled sample. Ethanol (484.4 μl, 100%) was added to each sample and allowed to incubate for 15 min. Samples were then placed on a magnetic stand for 5 min, the supernatant was removed, and the beads were washed twice with 600 μl of 90% ethanol. The samples were then centrifuged, and the remaining liquid was removed.

Samples were resuspended in 100 μl to a final concentration of 200 mM HEPES and 12 ng μl⁻¹ trypsin gold (Promega, V5280). Samples were then placed in a bioshaker (Bulldog Bio, VWR, 102407-834) and digested at 37 °C and 200 r.p.m. for 18 h. After digestion, samples were removed from the bioshaker, briefly sonicated, centrifuged, vortexed, centrifuged again and incubated on a magnetic stand for 5 min. The

supernatant was then removed and stored at −80 °C. Before analysis by LC–MS/MS, the samples were stage tipped[51]. Samples were resuspended in 0.1% formic acid at approximately 1 μg worth of digest per μl in glass HPLC inserts (Thermo Fisher, C4010-630) before analysis and then injected and analyzed via DIA method 4 using DIA gradient 3, detailed below (raw files, eGH692–eGH694). TMT labeling was used in these experiments to facilitate identification of neo-N termini produced before tryptic digestion and was not used for multiplexed in-set quantitation; each TMT-labeled sample was analyzed individually.

### Bulk TMTpro-labeled BMDM samples for differential protein analysis
Ten thousand cells from each treatment condition (24 h, treated with LPS and untreated), resuspended in LC–MS water, were frozen at −80 °C for 20 min, before being lysed at 90 °C in a thermal cycler (Bio-Rad, T1000) for 10 min. After lysis, Benzonase was added to a final concentration of 1 U and allowed to incubate for 10 min. Trypsin gold (Promega Trypsin Gold, MS grade, PRV5280) was added to a final concentration of 16 ng μl⁻¹, and triethylammonium bicarbonate (MilliporeSigma, T7408-100ML) was added to a final concentration of 150 mM. The samples were then allowed to digest overnight for 16 h. After digestion, samples were allowed to return to room temperature and were labeled with 85 mM TMT 128N (untreated sample) or 85 mM TMT 127C (LPS-treated sample). The reaction was then quenched with 0.5 μl of 0.5% hydroxylamine (MilliporeSigma, 467804-10ML) for 1 h. Samples were centrifuged briefly to collect liquid following the addition of all reagents. After labeling, about 6,000 cells worth of labeled material from each treatment condition was combined in an MS insert (Thermo Fisher, C4010-630) and dried down in a speed vacuum (Eppendorf) before being reconstituted in 3.3 μl of 0.1% formic acid (Thermo Fisher, 85178) and analyzed via shotgun MS instrument methods 1 and 2, using gradient 1, described below.

Separate samples containing approximately 1,000 cells per injection of the 128N-labeled untreated BMDMs or the 127C-labeled LPS-treated (24 h) BMDMs were injected and analyzed via DIA bulk BMDM analysis instrument method 1, described below. Each TMT-labeled sample was injected separately; TMT reporter ions were not used for in-set quantification in this analysis. Proteins that were differentially abundant between the two conditions analyzed by DIA were identified using the process outlined in Differential protein analysis for DIA samples, described below, and these proteins make up set ζ in the description of the high-priority-level composition in Prioritized inclusion list construction, also found below.

### Spectral library-generating samples
Before performing retention-time-calibration, scout or prioritized experiments, spectral libraries were generated by analyzing bulk injections of SQC sample or mixed carrier and reference samples. These spectral libraries were used to facilitate precursor identification in the lower-abundance retention-time-calibration samples.

For the SQC sample, 1-μl injections of a 10× concentrated aliquot of the SQC sample were analyzed via DIA methods 1 and 2 and DIA gradient method 1, and two subsequent 1-μl injections of a 1× concentrated aliquot of the SQC sample were analyzed via DIA method 1 and DIA gradient method 1. For HEK293 and melanoma samples, 1-μl injections of a 10× concentrated aliquot of the mixed carrier and reference sample were analyzed by DIA instrument methods 1 and 2 and DIA gradient method 1, and two subsequent 1× concentrated aliquots of the mixed carrier and reference sample were injected and analyzed by DIA method 1 and DIA gradient method 1. For the BMDM samples, 1-μl injections of a 5× concentrated aliquot of carrier and reference sample and a 1× concentrated aliquot of carrier and reference sample were sequentially analyzed via DIA instrument method 3 using DIA gradient method 1. Additional information regarding the instrument methods and search engine parameters can be found in the respective sections.

## Scout experiments

Before assembling an inclusion list for prioritized SQC sample or single-cell sample analysis, a prioritized analysis of a 1× concentrated version of the SQC sample or a 1× concentrated version of the mixed carrier and reference sample was performed to generate a set of additional DDA-identifiable precursors. Information regarding the inclusion list construction for these scout experiments, MaxQuant.Live parameters and analysis of raw data can be found in the respective sections.

## Retention-time-calibration experiments

Retention-time-calibration experiments were used to generate accurate retention times for identifiable precursors to be used in subsequent scout experiments and prioritized single-cell analyses.

**SQC samples.** For Fig. 1, a 1-µl injection of a 1× concentrated aliquot of the SQC sample was analyzed via DIA method 1 using DIA gradient 1 and searched using DIA-NN (version 1.8.2 beta 2) with the spectral library generated from the corresponding spectral library-generating experiments (library_TMTpro.tsv, 28,537 precursors). The precursor $m/z$ range was set to 450–1,600 Th, carbamidomethylation of cysteine was deselected as a fixed modification, the protease was set to trypsin, and the neural net classifier was set to double-pass mode. The following command line options were enabled: –no-ifs-removal, –full-unimod and –report-lib-info. All other settings were left as default.

**HEK and melanoma samples.** For Figs. 2 and 3, a 1-µl injection of a 1× concentrated aliquot of the mixed carrier and reference sample was analyzed via DIA method 1 using DIA gradient 1 and searched using DIA-NN (version 1.8.1 beta 23) with the corresponding spectral library-generating experiments (Rebuttal_library.tsv, 32,897 precursors). The precursor $m/z$ range was set to 450–1,600 Th, carbamidomethylation of cysteine was deselected as a fixed modification, the protease was set to trypsin, and the neural net classifier was set to double-pass mode. The following command line options were enabled: –no-ifs-removal and –report-lib-info. All other settings were left as default.

**BMDM nPOP samples.** For Figs. 4–6, a 1-µl injection of a 1× concentrated aliquot of mixed carrier and reference sample was injected and analyzed via DIA method 3 using DIA gradient 1 and searched using Spectronaut (version 15.1) with the spectral library generated from the corresponding spectral library-generating experiments (20210809_120040_Priori_comb_080921.kit). All search parameters were kept as default, except for template correlation profiling enabled for the profiling strategy and minimum $q$-value row selection for profiling row selection, and Biognosys' iRT kit was indicated as not being used.

**BMDM mPOP samples.** A 1-µl injection of a 1× concentrated aliquot of mixed carrier and reference sample was injected and analyzed via DIA method 3 using DIA gradient 2 and searched with Spectronaut[53] (version 15.0) in directDIA mode using a FASTA containing the SwissProt database for *Mus musculus*, as well as MEROPS cleavage fragments generated as indicated in MEROPS database preparation, below (musmusculus_SPonly_MEROPS_012221.fasta, 27,117 protein entries). Trypsin was specified as the enzyme for in silico digestion, TMTpro (+304.2071 Da) was selected as a fixed modification on lysines, and the following variable modifications were used: protein N-terminal acetylation (+42.01056 Da), methionine oxidation (+15.99492 Da) and TMTpro modification of peptide N termini. The results were then prefiltered in Spectronaut to only contain precursors with at least one TMTpro modification. All other search settings were kept as default.

## Prioritized inclusion list construction

A mapping between inclusion lists and samples can be found in Supplementary Tables 8–10.

**Scout experiments associated with the MaxQuant.Live feature contrast.** A set of four prioritized analyses of the 1× mixed carrier and reference sample were conducted to generate a library of DDA-identifiable precursors from an initial DIA retention-time-calibration experiment. The search results from the retention-time-calibration experiment were filtered to include only fully labeled peptides, and peptide sequences with multiple charge states were condensed to the single most confidently identified charge state by PEP. This collection of peptides will be referred to as Group A within this subsection. The data-processing pipeline used in the construction of these inclusion lists is available at https://github.com/SlavovLab/pSCoPE.

- Scout Run 1: peptides from Group A were stratified into high-, medium- and low-priority analysis groups on the basis of precursor intensity, such that peptides in the top third of intensities were placed in the high-priority group (8,154 peptides), peptides in the middle third of intensities were placed in the middle-priority group (7,914 peptides) and peptides in the bottom third of intensities were placed in the low-priority group (7,915 peptides). This forms Inclusion List 1.
- Scout Run 2: peptides from Group A were stratified into high-, medium- and low-priority analysis groups on the basis of precursor intensity, such that peptides in the top third of intensities were placed in the high-priority group (3,775 peptides), peptides in the middle third of intensities were placed in the middle-priority group (6,814 peptides) and peptides in the bottom third of intensities were placed in the low-priority group (7,491 peptides). Peptides previously identified in a scout experiment were placed in a base priority level that was only sent for analysis to keep duty cycles full when no peptides of higher priority were available (5,903 peptides). This forms Inclusion List 2.
- Scout Run 3: peptides from Group A were stratified into high-, medium- and low-priority analysis groups on the basis of precursor intensity, such that peptides in the top third of intensities were placed in the high-priority group (2,558 peptides), peptides in the middle third of intensities were placed in the middle-priority group (5,150 peptides) and peptides in the bottom third of intensities were placed in the low-priority group (7,062 peptides). Peptides previously identified in a scout experiment were placed in a base priority level that was only sent for analysis to keep duty cycles full when no peptides of higher priority were available (9,366 peptides). This forms Inclusion List 3.
- Scout Run 4: peptides from Group A were stratified into high-, medium- and low-priority analysis groups on the basis of precursor intensity, such that peptides in the top third of intensities were placed in the high-priority group (1,677 peptides), peptides in the middle third of intensities were placed in the middle-priority group (4,246 peptides) and peptides in the bottom third of intensities were placed in the low-priority group (6,622 peptides). Peptides previously identified in a scout experiment were placed on a base priority level that was only sent for analysis to keep duty cycles full when no peptides of higher priority were available (11,591 peptides). This forms Inclusion List 4.

**MaxQuant.Live analysis of SQC samples with and without prioritization.** For Fig. 1b,c, the search results from the retention-time-calibration experiment were filtered for use as an inclusion list via the data-processing pipeline available at https://github.com/SlavovLab/pSCoPE. The library of identifiable precursors referred to below was assembled from a set of four prioritized scout runs aimed at assembling a list of DDA-identifiable peptides from the 1× retention-time-calibration run. This library of peptide–spectrum matches (PSMs) was then filtered at 1% FDR, contaminants and reverse matches were removed, and multiple charge states of the same

modified sequence were condensed to the entry with the highest spectral confidence of identification. While the inclusion list below (Inclusion List 5) was used for the default MaxQuant.Live global targeting analyses as well, the prioritization feature was not enabled for those experiments.

- High priority: peptides from the filtered library referenced above (Group A) were then reduced to a subset that included the top four peptides per protein by spectral confidence (Group B). Group B was then filtered to only include peptides with spectral confidences ≤0.05 and precursor purities ≥0.8. An additional 810 peptides were added to this priority level from Group B, such that they were the most confident previously unselected identifications with precursor intensity fractions (PIFs) ≥0.5. This priority level featured 4,000 peptides.
- Middle priority: excluding peptides previously selected for a priority level, Group B was filtered to only include peptides with PEPs ≤0.05 and precursor purities (PIFs) ≥0.5. This priority level featured 4,000 peptides.
- Low priority: excluding peptides previously selected for a priority level, peptides from Group A were then filtered to only include peptides with PEP ≤0.05. This priority level featured 3,679 peptides.
- Retention-time-calibration peptides: all remaining precursors identified in the retention-time-calibration run were selected to participate in the real-time retention-time-alignment algorithm but were disabled from being sent for MS$^2$ analysis. This priority level featured 11,723 precursors.

**Scout experiments associated with HEK and melanoma analyses.** A set of four prioritized analyses of the 1× mixed carrier and reference sample were conducted to generate a library of DDA-identifiable precursors from an initial DIA retention-time-calibration experiment. The search results from the retention-time-calibration experiment were filtered to include only fully labeled peptides, and peptide sequences with multiple charge states were condensed to the single most confidently identified charge state by PEP. This collection of peptides will be referred to as Group A within this subsection. The data-processing pipeline used in the construction of these inclusion lists is available at https://github.com/SlavovLab/pSCoPE.

- Scout Run 1: peptides from Group A were stratified into high-, medium- and low-priority analysis groups on the basis of precursor intensity, such that peptides in the top third of intensities were placed in the high-priority group (9,014 peptides), peptides in the middle third of intensities were placed in the middle-priority group (8,749 peptides) and peptides in the bottom third of intensities were placed in the low-priority group (8,749 peptides). This forms Inclusion List 6.
- Scout Run 2: peptides from Group A were stratified into high-, medium- and low-priority analysis groups on the basis of precursor intensity, such that peptides in the top third of intensities were placed in the high-priority group (4,467 peptides), peptides in the middle third of intensities were placed in the middle-priority group (7,549 peptides) and peptides in the bottom third of intensities were placed in the low-priority group (8,287 peptides). Peptides previously identified in a scout experiment were placed on a base priority level that was only sent for analysis to keep duty cycles full when no peptides of higher priority were available (6,209 peptides). This forms Inclusion List 7.
- Scout Run 3: peptides from Group A were stratified into high-, medium- and low-priority analysis groups on the basis of precursor intensity, such that peptides in the top third of intensities were placed in the high-priority group (2,316 peptides), peptides in the middle third of intensities were placed in the

middle-priority group (6,197 peptides) and peptides in the bottom third of intensities were placed in the low-priority group (7,674 peptides). Peptides previously identified in a scout experiment were placed in a base priority level that was only sent for analysis to keep duty cycles full when no peptides of higher priority were available (10,325 peptides). This forms Inclusion List 8.
- Scout Run 4: peptides from Group A were stratified into high-, medium- and low-priority analysis groups on the basis of precursor intensity, such that peptides in the top third of intensities were placed in the high-priority group (1,467 peptides), peptides in the middle third of intensities were placed in the middle-priority group (4,695 peptides) and peptides in the bottom third of intensities were placed in the low-priority group (7,132 peptides). Peptides previously identified in a scout experiment were placed in a base priority level that was only sent for analysis to keep duty cycles full when no peptides of higher priority were available (12,948 peptides). This forms Inclusion List 9.

**HEK and melanoma samples (Fig. 2a).** The search results from the retention-time-calibration experiment were filtered for use as an inclusion list (Inclusion List 10) via the data-processing pipeline available at https://github.com/SlavovLab/pSCoPE. The library of identifiable precursors referred to below was assembled from the corresponding shotgun analyses and a set of four prioritized scout runs aimed at assembling a list of DDA-identifiable peptides from the 1× retention-time-calibration run. This library of PSMs was then filtered at 1% FDR, contaminants and reverse matches were removed, and multiple charge states of the same modified sequence were condensed to the entry with the highest spectral confidence of identification. Peptides found to be identified in 50% or fewer of the shotgun analyses were excluded from this list of identifiable peptides.

- High priority: peptides from the filtered library referenced above (Group A) were then reduced to a subset that included the top four peptides per protein by spectral confidence (Group B). Group B was then filtered to only include peptides with PEPs ≤0.05 and PIFs >0.8 for inclusion in the high-priority level. This priority level featured 4,013 peptides.
- Middle priority: excluding peptides previously selected for a priority level, Group B was filtered to only include peptides with PEPs ≤0.05 and PIFs >0.7 for inclusion in the middle-priority level. This priority level featured 4,166 peptides.
- Low priority: excluding peptides previously selected for a priority level, Group A was then filtered to only include peptides with PEPs ≤0.05 for inclusion in the low-priority level. This priority level featured 5,407 peptides.
- Retention-time-calibration peptides: all remaining precursors identified in the retention-time-calibration run were selected to participate in the real-time retention-time-alignment algorithm but were disabled from being sent for MS$^2$ analysis. This priority level featured 11,540 precursors.

**HEK and melanoma samples (Fig. 2b–e).** The search results from the retention-time-calibration experiment were filtered for use as an inclusion list (Inclusion List 11) via the data-processing pipeline available at https://github.com/SlavovLab/pSCoPE. The library of identifiable precursors referred to below was assembled from the corresponding shotgun analyses and a set of four prioritized scout runs aimed at assembling a list of DDA-identifiable peptides from the 1× retention-time-calibration run. This library of PSMs was then filtered at 1% FDR, contaminants and reverse matches were removed, and multiple charge states of the same modified sequence were condensed to the entry with the highest spectral confidence of identification.

- High priority: peptides from the filtered library referenced above (Group A) were then reduced to a set that had been identified in 50% or fewer of the associated shotgun experiments. Peptides with greater than 96% missing data in the single-cell samples were filtered out. Next, peptides were subset into groups identified in one, two, three or four of the associated shotgun experiments, and 250 peptides from each of these groups were randomly sampled to form a 1,000-peptide list of difficult-to-identify peptides.
- Medium–high priority: excluding peptides previously selected for a priority level, Group A was subset to include the top four peptides per protein by PEP (Group B). Group B was then filtered to only include peptides with PEPs ≤0.01 and PIFs ≥0.8. This priority level featured 3,475 peptides.
- Medium–low priority: excluding peptides previously selected for a priority level, the remaining library of identifiable precursors was filtered to only include peptides with PEPs ≤0.05 and PIFs ≥0.7. This priority level featured 4,146 peptides.
- Low priority: excluding peptides previously selected for a priority level, the remaining library of identifiable precursors was filtered to only include peptides with PEPs ≤0.05. This priority level featured 5,009 peptides.
- Retention-time-calibration peptides: excluding peptides previously selected for a priority level, the remaining library of precursors identified from the DIA retention-time-calibration run were selected to participate in the real-time retention-time-alignment algorithm but were disabled from being sent for MS[2] analysis. This priority level featured 11,496 precursors.

**HEK and melanoma samples for spike-in analysis.** For Fig. 3, the search results from the retention-time-calibration experiment were filtered for use as an inclusion list (Inclusion List 12) via the data-processing pipeline available at https://github.com/SlavovLab/pSCoPE. The library of identifiable precursors referred to below was assembled from the corresponding shotgun analyses and a set of four prioritized scout runs aimed at assembling a list of DDA-identifiable peptides from the 1× retention-time-calibration run. This library of PSMs was then filtered at 1% FDR, contaminants and reverse matches were removed, and multiple charge states of the same modified sequence were condensed to the entry with the highest spectral confidence of identification. Peptides found to be identified in 50% or fewer of the shotgun analyses were excluded from this list of identifiable peptides.

- High priority: identified precursors corresponding to the yeast-derived spike-in peptides (AYFTAPSSER, VEVDSFSGAK, TSIIGTIGPK and ELYEVDVLK) were selected for this priority level, which featured eight precursors.
- Medium–high priority: peptides from the filtered library referenced above (Group A) were then reduced to a subset that included the top four peptides per protein by PEP (Group B). Group B was then filtered to only include peptides with PEPs ≤0.05 and PIFs >0.8 for inclusion on the medium–high-priority level. An additional 134 peptides with PEPs ≤0.05 and PIFs >0.5 were added from Group B. This priority level featured 4,000 peptides.
- Medium–low priority: excluding peptides previously selected for a priority level, Group B was filtered to only include peptides with PEPs ≤0.05 and PIFs >0.7 for inclusion in the middle-priority level. This priority level featured 4,695 peptides.
- Low priority: excluding peptides previously selected for a priority level, Group A was then filtered to only include peptides with FDR ≤ 1% for inclusion in the low-priority level. This priority level featured 4,203 peptides.

- Retention-time-calibration peptides: all remaining precursors identified in the retention-time-calibration run were selected for this priority level. This priority level featured 9,258 precursors.

**Scout experiment for BMDM nPOP samples.** For Figs. 4–6, the search results from the retention-time-calibration experiment, Group A, were then filtered to meet the following criteria for use as an inclusion list (Inclusion List 13): elution group PEP ≤0.02, elution group $q$ value ≤ 0.05, TMTpro-labeling modifications (+304.2071 Da) on the N terminus or lysine residues. These filtered precursors form Group B.

- High priority: precursors from Group B in the top intensity tertile of Group A. This priority level featured 1,466 precursors.
- Medium priority: precursors from Group B in the middle intensity tertile of Group A. This priority level featured 2,429 precursors.
- Low priority: precursors from Group B in the low-intensity tertile of Group A. This priority level featured 1,808 precursors.
- Retention-time-calibration peptides: excluding peptides previously selected for a priority level, the remaining library of precursors identified from the DIA retention-time-calibration run were selected to participate in the real-time retention-time-alignment algorithm but were disabled from being sent for MS[2] analysis. This priority level featured 3,091 precursors.

**BMDM nPOP samples.** For Figs. 4–6, the prioritized inclusion list for BMDM samples (Inclusion List 14) was constructed by importing the search results from the DIA retention-time-calibration run into the R environment and subsetting the detected peptides into four priority levels. Peptides and proteins of special biological interest or experimental utility were assembled from the following sources: all precursors identified at or below a PEP of 0.05 in the scout experiment (set α); proteins significantly correlated with PC1 (5% FDR) in a cross-condition PCA generated from the 20 initial shotgun analyses of single-cell BMDM samples (set β); proteins significantly correlated with PC1 (5% FDR) in a PCA of the LPS-treated single cells generated from the 20 initial shotgun analyses of BMDM samples (set γ); proteins significantly correlated with PC1 (5% FDR) in a PCA of the untreated single cells generated from 20 initial shotgun analyses of BMDM samples (set δ); proteins with |log$_2$ (fold change)| ≥ 1 between the low and high dextran-uptake conditions of each treatment group analyzed by DIA, which were found to be statistically significant (set E, statistical process described in Differential protein analysis for DIA samples, below); proteins with |log$_2$ (fold change)| ≥ 1 between LPS-treated and untreated bulk BMDM samples analyzed via DIA, which were found to be statistically significant (set ζ, statistical process described in Differential protein analysis for DIA samples); proteins significantly correlated with PC1 (5% FDR, |Spearman correlation| > 0.35) from single-condition PCAs generated from prior mPOP-prepared single-cell BMDM analyses (set η); all precursors identified at 1% FDR in the 20 initial shotgun analyses of the single-cell BMDM samples (set θ); precursors identified in the retention-time-calibration run that were also contained in sets β, γ and δ and identified in fewer than 50% of the corresponding shotgun analyses of single-cell BMDMs (set ι).

Regarding precursor-intensity-dependent fill times, if a high-priority precursor appeared in the bottom intensity tertile, it was allotted an MS[2] fill time of 1,000 ms; if a high-priority precursor appeared in the middle intensity tertile, it was allotted an MS[2] fill time of 750 ms; if a high-priority precursor appeared in the top intensity tertile, it was allotted an MS[2] fill time of 500 ms. These intensity tertiles were calculated across all filtered PSMs from the retention-time-calibration run. All other precursors were allotted an MS[2] fill time of 500 ms.

- High priority: peptides from the retention-time-estimation run were selected to correspond to up to the top 125 most

abundant precursors from set ι; the top 35 most abundant MEROPS-annotated precursors; all precursors in set θ that mapped to proteins in set ζ; up to the top 100 most abundant precursors that were in common between set θ and precursors derived from proteins in set η; up to the top 100 most abundant precursors in common between set θ and precursors derived from proteins in set E; up to the top five most abundant precursors per protein in the intersection between set θ and precursors derived from proteins in set δ; up to the top six most abundant precursors per protein in the intersection between set θ and precursors derived from proteins in set γ; up to the top four most abundant precursors per protein in the intersection between set θ and precursors derived from proteins in set β. This priority level featured 589 precursors.

- Medium priority: excluding peptides previously selected for a priority level, remaining precursors from set θ, the set of peptides identified at 1% FDR in the accompanying single-cell shotgun analyses, were selected for this priority level. This priority level featured 1,353 precursors.
- Low priority: excluding peptides previously selected for a priority level, all remaining peptides identified in the scout experiment, set α, were selected for this priority level. This priority level featured 2,656 peptides.
- Retention-time-calibration peptides: excluding peptides previously selected for a priority level, all remaining peptides identified by the retention-time-calibration experiment were selected for participation in the real-time retention-time alignment algorithm but were not enabled for MS$^2$ analysis. This priority level featured 4,675 peptides.

**mPOP-prepared BMDM troubleshooting samples.** The search results from the DIA retention-time-calibration experiment were first filtered such that all remaining entries had an elution group PEP <0.05 and an elution group $q$ value < 0.05, as well as at least one TMTpro modification (+304.2071 Da) on either the peptide N terminus or the lysine residue. The prioritized inclusion list (Inclusion List 15) for the mPOP BMDM samples was constructed such that the high priority contained the following types of precursors that intersected with the retention-time-calibration experiment identifications: precursors identified in less than 50% of the corresponding 13 shotgun experiments, precursors featuring a MEROPS-annotated cleavage site, precursors mapping to proteins of biological interest (Toll-like receptors, interleukin-associated proteins, lysosomal-associated membrane proteins, IFN-associated proteins, caspases, NF-κB-associated proteins, transcription factors, gasdermin, signal transducers, macrophage scavenger receptors and proteins annotated for macrophage function). Precursors on the high-priority level were allocated fill times dependent upon their precursor intensities in the following manner: precursors in the top abundance tertile were allocated an MS$^2$ fill time of 600 ms, precursors in the middle abundance tertile were allocated an MS$^2$ fill time of 750 ms and precursors in the bottom abundance tertile were allocated an MS$^2$ fill time of 900 ms. Precursors identified in the retention-time-calibration experiment that were part of a previous targeting experiment were placed on the middle-priority level along with the precursors that had been identified in the corresponding mPOP shotgun experiments. The bottom priority level, although redundant with the top and middle levels in composition, served to keep the instrument duty cycles full for optimal elution peak sampling. All precursors enabled for MS$^2$, as well as all remaining precursors identified in the filtered retention-time-calibration run, were enabled for participation in MaxQuant.Live's real-time retention-time-calibration algorithm.

### Analysis of raw MS data
**MEROPS database preparation and FASTA modification.** The MEROPS database was downloaded from https://www.ebi.ac.uk/ merops/download[54] and converted into a .csv file for import into the R environment. All cleavage patterns consistent with trypsin (for example, R or K as the P1 residue) were removed from the database. Next, the SwissProt-annotated *M. musculus* FASTA file was read into the R environment, and the sequence for each protein with an annotated MEROPS cleavage site was split between the P1 and P1' residues. Both halves of the MEROPS-cleaved peptide were then subjected to an in silico tryptic digestion such that the tryptic digest produced a fragment at least six amino acids long. The two semitryptic peptide halves were then added to the existing FASTA as separate entries, including annotations from the MEROPS database for the enzyme, cleavage residue number and whether the peptide fragment contained the neo-C terminus or the neo-N terminus.

### DDA data
**Scout experiments of SQC samples.** The four scout experiments associated with Fig. 1 were searched with MaxQuant (1.6.17.0)[55–57] using a FASTA containing all entries from the human SwissProt database and two yeast-derived spike-in proteins (2022-06-20-reviewed -UP000005640_withSpikeIn.fasta, 20,373 proteins). TMTpro 16-plex was enabled as a fixed modification on N termini and lysines via the reporter ion MS$^2$ submenu. Methionine oxidation (+15.99492 Da) was enabled as a variable modification, and trypsin was selected for the in silico digestion with enzyme mode set to specific. Up to two missed cleavages were allowed per peptide with a minimum length of seven amino acids. Second peptide identifications were disabled, 'calculate peak properties' was enabled, and msScans was enabled as an output file. PSM FDR and protein FDR were set to 1. FDR calculations were performed in the R programming environment by calculating the PEP threshold at which 1% of the entries were decoy identifications.

**SQC samples analyzed by MaxQuant.Live with and without prioritization.** For Fig. 1, the six matched analyses of the standard SQC sample conducted via MaxQuant.Live with and without prioritization enabled were searched with MaxQuant (1.6.17.0) using a FASTA containing all entries from the human SwissProt database and two yeast-derived spike-in proteins (2022-06-20-reviewed-UP000005640_withSpikeIn. fasta, 20,373 proteins). TMTpro 16-plex was enabled as a fixed modification on N termini and lysines via the reporter ion MS$^2$ submenu. Methionine oxidation (+15.99492 Da) was enabled as a variable modification, and trypsin was selected for the in silico digestion with enzyme mode set to specific. Up to two missed cleavages were allowed per peptide with a minimum length of seven amino acids. Second peptide identifications were disabled, calculate peak properties was enabled, and msScans was enabled as an output file. PSM FDR and protein FDR were set to 1. FDR calculations were performed in the R programming environment by calculating the PEP threshold at which 1% of the entries were decoy identifications.

**Shotgun analyses of HEK and melanoma single-cell samples.** For Fig. 2, shotgun analyses of the HEK and melanoma samples were searched with MaxQuant (1.6.17.0) using a FASTA containing all entries from the human SwissProt database and two yeast-derived spike-in proteins (2022-06-20-reviewed-UP000005640_withSpikeIn.fasta, 20,373 proteins). TMTpro 18-plex was enabled as a fixed modification on N termini and lysines via the reporter ion MS$^2$ submenu. Methionine oxidation (+15.99492 Da) and protein N-terminal acetylation (+42.01056 Da) were enabled as variable modifications, and trypsin was selected for the in silico digestion with enzyme mode set to specific. Up to two missed cleavages were allowed per peptide with a minimum length of seven amino acids. Second peptide identifications were disabled, calculate peak properties was enabled, and msScans was enabled as an output file. PSM FDR and protein FDR were set to 1. FDR calculations were performed in the R programming environment by calculating the PEP threshold at which 1% of the entries were decoy identifications.

**Scout experiments of HEK and melanoma carrier and reference materials.** The four scout experiments associated with Figs. 2 and 3 were searched with MaxQuant (1.6.17.0) using a FASTA containing all entries from the human SwissProt database and two yeast-derived spike-in proteins (2022-06-20-reviewed-UP000005640_withSpikeIn. fasta, 20,373 proteins). TMTpro 18-plex was enabled as a fixed modification on N termini and lysines via the reporter ion MS² submenu. Methionine oxidation (+15.99492 Da) was enabled as a variable modification, and trypsin was selected for the in silico digestion with enzyme mode set to specific. Up to two missed cleavages were allowed per peptide with a minimum length of seven amino acids. Second peptide identifications were disabled, calculate peak properties was enabled, and msScans was enabled as an output file. PSM FDR and protein FDR were set to 1. FDR calculations were performed in the R programming environment by calculating the PEP threshold at which 1% of the entries were decoy identifications.

**pSCoPE analyses of HEK and melanoma single-cell samples.** In all three datasets, the prioritized runs corresponding to Figs. 2a–e and 3, the same search settings were used as in the accompanying shotgun datasets, with the exception of only specifying methionine oxidation (+15.99492 Da) as a variable modification in the prioritized analyses. FDR calculations were performed in the R programming environment by calculating the PEP threshold at which 1% of the entries were decoy identifications.

**Isobaric match-between-runs contrast.** For Extended Data Fig. 5, the shotgun analyses of HEK and melanoma samples originally associated with Fig. 2 were searched as described in the associated section, but match between runs was enabled. No changes were made to the search strategy employed for the prioritized analyses originally associated with Fig. 2.

**TMT-labeled bulk BMDM sample analyses for differential protein analysis.** TMT-labeled and mixed bulk samples of LPS-treated (24 h) and untreated BMDMs analyzed via DDA MS methods 1 and 2 and DDA gradient method 1 (wGH215 and wGH216, respectively) were searched with MaxQuant (version 1.6.17.0) using a FASTA containing all entries from the SwissProt database for *M. musculus*, as well as MEROPS-annotated cleavage products generated as indicated previously (musmusculus_SPonly_MEROPS_012221.fasta), for a total of 27,117 protein entries. TMTpro 16-plex was enabled as a fixed modification on N termini and lysines via the reporter ion MS² submenu. Methionine oxidation (+15.99492 Da) and protein N-terminal acetylation (+42.01056 Da) were enabled as variable modifications, and trypsin was selected with specific cleavage. Second peptide identifications were disabled, calculate peak properties was enabled, and msScans was enabled as an output. PSM FDR and protein FDR were set to 1.

**Scout experiment for inclusion list generation for BMDM nPOP samples.** The raw file generated by this prioritized analysis was searched with MaxQuant (version 1.6.17.0) using a FASTA containing all entries from the murine SwissProt database (musmusculus_SPonly_012221. fasta, 17,056 proteins). TMTpro 16-plex was enabled as a fixed modification on N termini and lysines via the reporter ion MS² submenu. Methionine oxidation (+15.99492 Da) and protein N-terminal acetylation (+42.01056 Da) were enabled as variable modifications, and trypsin was selected with specific cleavage. Second peptide identifications were disabled, calculate peak properties was enabled, and msScans was enabled as an output. PSM FDR and protein FDR were set to 1.

**Shotgun analyses of BMDM nPOP samples.** For Figs. 4–6, shotgun analyses of nPoP-prepared murine BMDM samples were searched with MaxQuant (2.0.3.0) using a FASTA containing all entries from the murine SwissProt database with additional entries for cleaved peptides consistent with the MEROPS database appended (musmusculus_SPonly_MEROPS_012221.fasta), for a total of 27,117 protein entries. TMTpro was enabled as a fixed modification on N termini and lysines via the reporter ion MS² submenu. Methionine oxidation (+15.99492 Da) and protein N-terminal acetylation (+42.01056 Da) were enabled as variable modifications, and trypsin was selected for the in silico digestion with enzyme mode set to specific. Up to two missed cleavages were allowed per peptide with a minimum length of seven amino acids. Second peptide identifications were disabled, calculate peak properties was enabled, and msScans was enabled as an output file. PSM FDR and protein FDR were set to 1.

**pSCoPE analyses of BMDM nPOP samples.** For Figs. 4–6, the same search settings were used as for the accompanying shotgun datasets, with the exception of the FASTA database. For the prioritized samples, a reduced version of the murine SwissProt database with appended MEROPS entries was used, which contained only those proteins with peptides on the inclusion list (musmusculus_SPonly_MEROPS_012221_ lim2.fasta; 1,234 proteins). Subsequent to the MaxQuant search, the 20 shotgun-analyzed nPOP-prepared SCoPE experiments, the 40 pSCoPE-analyzed nPOP-prepared SCoPE experiments and their preceding DIA-analyzed retention-time-calibration experiment were analyzed together with DART-ID[58] for retention-time-dependent PSM confidence updating. A DART-ID configuration file is included in the MassIVE repository associated with this publication.

**Shotgun and pSCoPE analyses of BMDM mPOP troubleshooting samples.** Shotgun and pSCoPE analyses of the murine BMDM single-cell samples prepared by mPOP were searched with MaxQuant (1.6.7.0) using a FASTA containing all entries from the murine SwissProt database with additional entries for cleaved peptides consistent with the MEROPS database appended (musmusculus_SPonly_MEROPS_012221. fasta), for a total of 27,117 protein entries. TMTpro was enabled as a fixed modification on N termini and lysines via the reporter ion MS² submenu. Methionine oxidation (+15.99492 Da) and protein N-terminal acetylation (+42.01056 Da) were enabled as variable modifications, and trypsin/P was selected for the in silico digestion with enzyme mode set to specific. Up to two missed cleavages were allowed per peptide with a minimum length of seven amino acids. Second peptide identifications were disabled, calculate peak properties was enabled, and msScans was enabled as an output file. PSM FDR and protein FDR were set to 1.

**DIA data**
DIA analysis was used to identify many precursors and their associated retention times, so that they could be used to compile inclusion lists (Supplementary Fig. 4).

**SQC samples, spectral library-generating search.** One-microliter injections of a 10× concentrated aliquot of the SQC sample were analyzed by DIA instrument methods 1 and 2 using DIA gradient 1, and two 1-µl injections of a 1× concentrated aliquot of the SQC sample were analyzed by DIA instrument method 1 using DIA gradient 1. These four sample analyses were used to construct a spectral library via FragPipe for the analysis of a 1× mixed carrier and reference sample analyzed by DIA instrument method 1 using DIA gradient 1 (that is, a retention-time-calibration experiment). The spectral library contained a total of 28,537 precursors and was generated from a directDIA search using 2022-06-20-decoys-reviewed-contam-UP000005640_SpikeIns. fasta (20,373 proteins). Default search parameters were used from the DIA_speclib_quant workflow, with the following exceptions: the enzyme was set to trypsin; N-terminal acetylation, nQnC and nE were disabled as variable modifications; carbamidomethylation of cysteine residues was disabled as a fixed modification; TMTpro (+304.207146 Da) on peptide N termini and lysine residues was enabled as a fixed modification; 'quantify with DIA-NN' was disabled. The spectral library produced by this search was named library_TMTPro.tsv.

**HEK and melanoma samples, spectral library-generating search.** One-microliter injections of a 10× concentrated aliquot of the mixed carrier and reference samples were analyzed by DIA instrument methods 1 and 2 using DIA gradient 1, and two 1-µl injections of a 1× concentrated aliquot of the mixed carrier and reference samples were analyzed by DIA instrument method 1 using DIA gradient 1. These four sample analyses were used to construct a spectral library via FragPipe for the analysis of a 1× mixed carrier and reference sample analyzed via DIA instrument method 1 using DIA gradient 1 (that is, a retention-time-calibration experiment). The spectral library contained a total of 32,897 precursors and was generated from a directDIA search using 2022-06-20-decoys-reviewed-contam-UP000005640_SpikeIns. fasta (20,373 proteins). Default search parameters were used from the DIA_speclib_quant workflow, with the following exceptions: the enzyme was set to trypsin; N-terminal acetylation, nQnC and nE were disabled as variable modifications; carbamidomethylation of cysteine residues was disabled as a fixed modification; TMTpro (+304.207146 Da) on peptide N termini and lysine residues was enabled as a fixed modification; quantify with DIA-NN was disabled. The spectral library produced by this search was named Rebuttal_library.tsv.

**BMDM samples, spectral library-generating search.** One-microliter injections of both a 5× concentrated aliquot and a 1× concentrated aliquot of mixed carrier and reference samples were sequentially analyzed by DIA instrument method 3 using DIA gradient 1. The two sample analyses indicated above were used to construct a spectral library using Spectronaut (version 15.1) via a directDIA search using musmusculus_SPonly_MEROPS_012221.fasta (27,117 proteins). Methionine oxidation (+15.99492 Da), N-terminal acetylation (+42.01056 Da) and N-terminal TMTpro labeling (+304.2071 Da) were enabled as variable modifications, while TMTpro labeling of lysines was enabled as a fixed modification. The resulting spectral library contained a total of 11,701 precursors. Default search parameters were used, with the following exceptions: 'allow source specific iRT calibration' was enabled, Biognosys' iRT kit alignment peptides were specified as unused, and the profiling strategy was set to template correlation profiling. The spectral library produced by this search was named 20210809_120040_Priori_comb_080921.kit.

**SQC sample, retention-time-calibration experiment.** For Fig. 1b,c, raw data were searched via DIA-NN (version 1.8.2 beta 2) with the sample-specific spectral library (library_TMTPro.tsv, 28,537 precursors) discussed above to provide accurate retention times for the subsequent MaxQuant.Live-enabled prioritized analyses of the SQC sample. The reference FASTA for this spectral library was 2022-06-20-decoys-reviewed-contam-UP000005640_SpikeIns.fasta (20,373 proteins). The protease was set to trypsin, N-terminal M excision was enabled, carbamidomethylation of cysteine residues was disabled, the precursor $m/z$ range was set to 450–1,600 Th, the fragment $m/z$ range was set to 200–2,000 Th, double-pass mode was enabled, and the following command line arguments were enabled: –no-ifs-removal, –full-unimod and –report-lib-info. All other options were kept as default.

**HEK and melanoma samples, retention-time-calibration experiment.** For Figs. 2 and 3, raw data were searched via DIA-NN[54] (version 1.8.1 beta 23) with the sample-specific spectral library (Rebuttal_library. tsv, 32,897 precursors) discussed above to provide accurate retention times for the subsequent MaxQuant.Live-enabled prioritized analyses. The reference FASTA for this spectral library was 2022-06-20-decoys-reviewed-contam-UP000005640_SpikeIns.fasta (20,373 proteins). The protease was set to trypsin, N-terminal M excision was enabled, carbamidomethylation of cysteine residues was disabled, the precursor $m/z$ range was set to 450–1,600 Th, the fragment $m/z$ range was set to 200–2,000 Th, double-pass mode was enabled, and the following command line arguments were enabled: –no-ifs-removal and –report-lib-info. All other options were kept as default.

**BMDM samples, pre-scout retention-time-calibration experiment.** For Figs. 4–6, raw data were searched via Spectronaut (version 15.1) with the BMDM-specific spectral library 20210809_120040_Priori_comb_080921.kit. The reference FASTA for this spectral library was musmusculus_SPonly_MEROPS_012221.fasta (27,117 proteins). The iRT kit alignment peptides were specified as unused, template correlation profiling was selected as the profiling strategy, minimum $q$-value row selection was selected for the profiling row-selection method, and 'allow source specific iRT calibration' was set to true. All other options were kept as default.

**BMDM nPOP samples, retention-time-calibration experiment.** For Figs. 4–6, raw data were searched via Spectronaut (version 15.1) with the following spectral library: 20210809_120040_Priori_comb_080921.kit. The reference FASTA for this spectral library was musmusculus_SPonly_MEROPS_012221.fasta (27,117 proteins). iRT kit alignment peptides were specified as unused, template correlation profiling was selected as the profiling strategy, minimum $q$-value row selection was selected for the profiling row-selection method, and 'allow source specific iRT calibration' was set to true. All other options were kept as default.

**TMTpro-labeled bulk BMDM samples for differential protein analysis.** TMT-labeled, unmixed bulk samples (wGH217/218/220) analyzed by DIA instrument method 3 and DIA gradient method 1 were searched with Spectronaut (version 14.1) in directDIA mode using musmusculus_SPonly_MEROPS_012221.fasta, containing 27,117 proteins. Methionine oxidation (+15.99492 Da), N-terminal acetylation (+42.01056 Da) and N-terminal TMTpro labeling (+304.2071 Da) were enabled as variable modifications, while TMTpro labeling of lysines was enabled as a fixed modification. Trypsin was specified as the enzyme for in silico digestion, and results were filtered to contain only precursors with TMTpro-labeling modifications.

**TMTpro-labeled bulk MEROPS validation samples for BMDM analysis.** For Fig. 6, the raw files from the MEROPS analyses were then searched with Spectronaut's (version 15.4) directDIA analysis feature, using a FASTA containing all entries from the SwissProt database for *M. musculus*, as well as MEROPS cleavage fragments generated as indicated previously, which contained 27,117 protein entries (musmusculus_SPonly_MEROPS_012221.fasta). Cysteine carbamidomethylation was set as a fixed modification, and the following variable modifications were used: protein N-terminal acetylation (+42.01056 Da), methionine oxidation (+15.99492 Da), TMTpro modification (+304.2071 Da) of lysine and peptide N termini. Trypsin enzymatic cleavage rules were enabled, allowing for a minimum peptide length of seven and a maximum peptide length of 52. Up to two missed cleavages were allowed. All other search settings were left at their default values.

**Label-free bulk endocytosis samples for BMDM analysis.** For Fig. 5, raw data from the bulk endocytosis sample analyses were searched with Spectronaut (version 14.10) with directDIA, using a FASTA containing all entries from the SwissProt database for *M. musculus*, as well as selected isoforms and MEROPS-annotated cleavage products generated as indicated previously, for a total of 33,996 protein entries (Mouse_ONLYsp_plusMEROPS_v2.fasta). Peptides with lengths between six and 52 amino acids with up to two missed cleavages were permitted. Trypsin/P was selected for cleavage, and protein N-terminal acetylation (+42.01056 Da) and methionine oxidation (+15.99492 Da) were enabled as variable modifications. A PEP cutoff of 1 was selected, although downstream filtration (PEP ≤ 0.01) was performed in the differential protein-analysis script. All other search settings were kept as default.

**BMDM mPOP samples, retention-time-calibration experiment.** The pre-prioritization retention-time-calibration experiment was searched with Spectronaut (version 15.0) in directDIA mode using a

FASTA containing the SwissProt database for *M. musculus*, as well as MEROPS cleavage fragments generated as indicated previously, containing 27,117 protein entries (musmusculus_SPonly_MEROPS_012221). Trypsin was specified as the enzyme for in silico digestion, TMTpro (+304.2071 Da) was selected as a fixed modification on lysines, and the following variable modifications were used: protein N-terminal acetylation (+42.01056 Da), methionine oxidation (+15.99492 Da) and TMTpro modification of peptide N termini. The results were then prefiltered in Spectronaut to only contain precursors with at least one TMTpro modification. All other search settings were kept as default.

### Processing and normalizing single-cell MS data

**Shotgun and pSCoPE HEK and melanoma analyses.** Shotgun and pSCoPE sets pertaining to Fig. 2a–e were processed separately, with identical data filters applied to each dataset. Single-cell MS data were processed via the SCoPE2 single-cell proteomic pipeline[3,33]. Peptides with precursor ion fractions below 50% or a mean RI intensity across the single cells greater than 10% of the intensity in the carrier channel were removed from the dataset. Peptides were filtered at 1% FDR by determining the PEP threshold at which 1% of the entries were reverse matches. Cells with mean protein CVs greater than 0.4 were filtered out from the dataset. Samples and precursors were then filtered to have less than 99% missing data, before being log transformed and aggregated to protein-level abundance by taking the median abundance of the protein-specific peptides. The following intermediate data frames were generated for subsequent analysis from both datasets: the matrix of single cells by precursors, unfiltered for missingness; the matrix of unimputed protein abundances by single cells, containing missing values. For the sets corresponding to Fig. 2b–e, the following additional data matrices were produced: the complete matrix of protein abundances by single cells, containing imputed values; the batch-corrected complete matrix of protein abundances by single cells, containing imputed values; the re-normalized batch-corrected complete matrix of protein abundances by single cells, containing imputed values.

**pSCoPE analyses of BMDM samples.** For Figs. 4–6, single-cell data from 40 prioritized analyses were processed via the single-cell pipeline[3,33]. Peptides with precursor ion fractions below 50% were removed from the dataset, as were peptides with a mean intensity across the single cells greater than 2% of the intensity in the carrier channel. Peptides were filtered at 1% FDR using the DART-ID[58] q-value column. Cells with mean protein CVs greater than 0.4 were filtered out from the dataset. Samples and precursors were then filtered to have less than 99% missing data, before being log transformed and aggregated to protein-level abundance by taking the median abundance of the protein-specific peptides. The following intermediate data frames were generated for subsequent analysis: the matrix of single cells by precursors, unfiltered for missingness; the matrix of unimputed protein abundances by single cells, containing missing values; the complete matrix of protein abundances by single cells, containing imputed values; the batch-corrected complete matrix of protein abundances by single cells, containing imputed values; the re-normalized batch-corrected complete matrix of protein abundances by single cells, containing imputed values.

**Shotgun and pSCoPE analyses of BMDM samples.** For Extended Data Fig. 6, single-cell data from 20 shotgun analyses and the first 20 prioritized analyses were processed via the single-cell pipeline[3,33] using the same parameters as indicated above for the pSCoPE-only BMDM analyses.

### Data analysis

**Differential protein analysis for DIA samples.** Differential protein abundance was assessed by modeling the distribution of noise as a function of average precursor intensity. To perform this analysis, a single sample was injected and analyzed twice by DIA and then searched with Spectronaut as described above. Precursor-level fold changes between replicate injections of the same sample should cluster around 1:1; deviations from this expected 1:1 ratio reflect noise in the measurement and can be used as a null distribution to test for differential protein abundance. However, because precursor quantitation is more accurate at higher absolute intensities, the null distribution of precursor fold changes was split evenly into 15 bins with respect to the average precursor intensity of the pair. Fold changes between experimental conditions were then calculated and converted to a z score using its corresponding null distribution of fold changes (based on intensity). Lastly, the precursors for each protein were t-tested against a standard-normal null distribution of 10,000 values with mean = 0 and s.d. = 1, and then P values were converted to q values using the Benjamini and Hochberg approach.

**Fig. 1, MaxQuant.Live contrast with and without prioritization Bar plot of consistency of identification.** The evidence.txt file containing data from the six SQC experiments acquired by MaxQuant. Live without prioritization enabled (wGH0727, wGH0729, wGH0731, wGH0733, wGH0735, wGH0737) and with prioritization enabled (wGH0728, wGH0730, wGH0732, wGH0734, wGH0736, wGH0738) were filtered at 1% FDR, and a new column for precursor identity was created by concatenating the modified sequence and charge state of each entry. Entries were then aggregated by precursor on a per-experiment basis, selecting the identification with the lowest PEP. This data frame was then reduced to a column of precursor identities and raw files. The number of experiments in which a precursor was identified was tallied for each analysis method, and these per-platform tallies were then left joined to the inclusion list, and resulting NA values for precursors not identified in any experiment were set to 0. The per-analysis method per-precursor sums were then divided by the total number of experiments in each analysis set (six) and multiplied by 100 to generate the identification rate as a percentage. The data were then subset to only include information for precursors in the high-priority group (4,000 precursors), and the resulting data frame was summarized such that the number of precursors identified with a particular frequency was tallied on a per-analysis method basis; these data were plotted to produce the leftmost plot in Fig. 1b.

**Box plot of proteins quantified per run.** The evidence.txt file containing the six SQC experiments acquired by MaxQuant.Live without prioritization enabled (wGH0727, wGH0729, wGH0731, wGH0733, wGH0735, wGH0737) and with prioritization enabled (wGH0728, wGH0730, wGH0732, wGH0734, wGH0736, wGH0738) were filtered at 1% FDR, and the tally of unique proteins quantified per experiment was made using the Leading.razor.protein column. These tallies were then presented as box plots in the rightmost plot of Fig. 1b.

**Box plot of prioritized peptides detected at MS$^1$ and sent for MS$^2$ analysis.** The MaxQuant.Live log files corresponding to the six experiments for each analysis method were parsed using an R script available at https://github.com/SlavovLab/pSCoPE to extract the list of precursors detected at MS$^1$ and sent for MS$^2$ analysis. The precursors detected at MS$^1$ or sent for MS$^2$ were then summed on a per-experiment and per-priority-level basis and divided by the total number of precursors corresponding to that priority level on the inclusion list before being multiplied by 100 to yield a percentage detection or isolation rate. These per-experiment and per-priority-level detection and isolation percentages were then plotted in Fig. 1c, with each point corresponding to an experiment, of which there were six for each analysis method.

**Fig. 2, HEK and melanoma samples**
**Box plot of productive MS$^2$ scans.** The msms.txt files from the shotgun and pSCoPE MaxQuant search results were filtered at 1% FDR by determining the PEP threshold at which 1% of the entries were reverse

matches. All contaminant and reverse matches were then removed from the resulting data frames. The number of remaining PSMs was tallied and divided by the total number of MS$^2$ scans recorded per experiment, determined from the msmsScans.txt output of MaxQuant. This fraction was multiplied by 100 and presented in Fig. 2a as the percentage of productive MS$^2$ scans per experiment.

**Box plot of peptides per run.** The evidence.txt files from the shotgun and pSCoPE MaxQuant search results were filtered at 1% FDR by determining the PEP threshold at which 1% of the entries were reverse matches. All contaminant and reverse matches were then removed from the resulting data frames, and peptides with multiple charge states were collapsed to a single entry per experiment. The number of PSMs remaining was then tallied on a per-experiment basis and presented as a box plot in Fig. 2a.

**Box plot of proteins per cell.** Using the matrix of unimputed protein abundances by single-cell samples produced by the single-cell pipeline[3,33], the number of proteins per single-cell sample with detectable reporter ion intensities was tallied and presented in Fig. 2a.

**Sensitivity analysis.** Eight matched experiments for shotgun (wGH0643, wGH0644, wGH0645, wGH0646, wGH0647, wGH0648, wGH0649, wGH0650) and pSCoPE (wGH0660, wGH0661, wGH0663, wGH0665, wGH0666, wGH0667, wGH0668, wGH0669) were used in this comparison. The filtered evidence.txt file produced after removal of contaminant and reverse matches and filtration for PIF (>0.5), mean reporter ion ratio between the single-cell samples and the carrier (≤0.1) and FDR (≤1%) were used as the data source for this figure. The distribution of precursor intensities for all PSMs was divided into 30 intensity bins on a per-experiment basis, and the number of identifications per bin per experiment was calculated. The median and s.d. of these per-bin identifications was calculated on a per-platform basis (shotgun or prioritization). The median and s.d. were then represented as circles and error bars, respectively, in the associated figure.

**Consistency heatmap.** The precursors by the single-cell matrix produced after filtration for PIF (>0.5), mean reporter ion ratio between the single-cell samples and the carrier (≤0.1), FDR (≤1%) and CV (≤0.4) were subsetted to contain only those precursors in the high-priority level of the consistency experiment inclusion list, which contained 1,000 precursors identified in 50% or fewer of the shotgun experiments. The precursors were then aggregated to the peptide level, and their abundance measurements were binarized such that precursors with NA intensities for a single-cell sample were given a value of zero while precursors with detected reporter ion intensity were given a value of 1. The resulting binary data frame was then presented as a heatmap with peptides on the y axis and single cells on the x axis to contrast data completeness for difficult-to-identify peptides placed on the top two levels of a prioritized inclusion list. The set of peptides was filtered to only include entries with at least one cell with detected reporter ion intensity, as shown in Fig. 2c.

**Box plots of peptide and protein-level data-completeness contrast.** To produce the peptide-level data-completeness box plot shown in Fig. 2d, the precursors by the single-cell matrix produced after filtration for PIF (>0.5), mean reporter ion ratio between the single-cell samples and the carrier (≤0.1), FDR (≤1%) and CV (≤0.4) were subsetted into two data frames: one containing precursors from the high-priority level of the prioritized inclusion list and one containing precursors that were enabled for MS$^2$ analysis from any priority level. Precursor abundances were aggregated to the peptide level by summing the relative intensities observed for multiple charge states on a per-sample basis; this sum was not used for relative quantitation, only for representing whether detectable reporter ion signal was

observed for a peptide in a single-cell sample, as opposed to an NA value. The fraction of peptides with detected reporter ion signal per single cell was computed for these two priority categories before being multiplied by 100 to produce the percent data completeness per cell.

To generate the protein-level data-completeness box plot shown in Fig. 2d, the matrix of protein abundances by single cells produced by the scp pipeline[3,33] was subsetted to include only the proteins for which precursors were specified for MS$^2$ analysis on the prioritized inclusion list. The fraction of these proteins with detected reporter ion signal per single cell was computed and then multiplied by 100 to produce the percent data completeness per cell.

**HEK and melanoma PCA color coded by median protein set abundance.** PCA was performed on the imputed, batch-corrected and normalized proteins by the cell matrix, using the prcomp function in R. For PCAs color coded by the median protein set abundance, the median abundance of all proteins mapping to a protein set was calculated on a per-cell basis, using the unimputed batch-corrected data matrix. The vector of single-cell protein set abundances was then $z$ scored, and the resulting vector was joined with the vector of PC coordinates (the score vectors) by sample ID. The protein sets presented in this analysis were selected from the results of PC-weight-based PSEA, as described below.

### Fig. 3, HEK and melanoma samples

**Reporter ion-intensity distributions.** The evidence.txt file generated by MaxQuant was filtered to exclude PSMs with PIF <0.5 and PEPs ≥0.02. Scans of type 'MSMS' were also removed, as these lack precursor mass and intensity information. The single-cell and control sample reporter ion intensities from each experiment were tallied across 30 intensity bins on the basis of whether they were associated with endogenous or spike-in proteins. These distributions were then plotted as side-by-side violin plots.

**Regression of normalized reporter ion intensities on spike-in concentrations.** The evidence.txt file generated by MaxQuant was filtered to exclude PSMs with PIF <0.5 and PEPs ≥0.02. Scans of type 'MSMS' were also removed, as these lack precursor mass and intensity information. Finally, only PSMs corresponding to the four sequences (AYFTAPSSER, VEVDSFSGAK, TSIIGTIGPK, ELYEVDVLK) generated by tryptic digestion of the spike-in peptides were selected for comparing measured and spiked-in levels. On a per-experiment and per-precursor basis, the median reporter ion intensity was taken of the three single-cell samples containing 1× concentrations of each spike-in peptide. All reporter ion intensities for a given precursor were then divided by this median to generate relative reporter ion intensities. As a given peptide sequence may be associated with multiple charge states, the per-sample normalized reporter ion intensities were then condensed to a median value for each peptide sequence. The set of all median-normalized reporter ion intensities were then regressed on their respective spike-in amounts, setting the intercept to zero and using partial least squares. In the associated plot, the $\log_2$ (concentration) of zero corresponds to the 1× concentration, with each subsequent increment corresponding to the 2×, 4×, 8× and 16× spike-in concentrations, respectively. The gray circle in the associated plot corresponds to the median-normalized reporter ion intensity for all spike-in peptides across all samples at the concentration level, while the error bars reflect the median ± s.d. of the normalized reporter ion intensities. The text at the top of the plot indicates the number of sequences observed across all eight injections for that concentration. The data shown in this figure correspond to a set of eight prioritized experiments acquired via MaxQuant.Live, in which the spike-in peptides were on the high-priority level and were allowed to fill for the default MS$^2$ fill time of 300 ms.

## Figs. 4–6, BMDM samples

**PCAs color coded by median protein set abundance.** For PCAs color coded by the median protein set abundance (Fig. 4a, Fig. 5a,c and Extended Data Fig. 7), the procedure indicated in the preceding section was followed.

**Protein set enrichment analysis.** For Fig. 4b, PSEA was performed using the vector of PC-associated protein weights produced by PCA analysis of the imputed, batch-corrected and normalized cell × protein matrix, generated from the prcomp function in R. The human gene set database was acquired from GOA[59]. The gene set database was filtered to remove entries corresponding to cellular components, in favor of entries annotated as molecular function and biological process. An additional GO category was generated by subsetting the proton-transport-annotated proteins to include only the V-type ATPase-associated proteins; this GO term was named 'Proton Transport, V-Type ATPase'. Gene-level annotation was used to map gene sets to the protein weights. The factor weights for all proteins matching a protein set were compared against the background distribution of protein weights using the two-tailed Wilcoxon rank-sum test. The following filters were used to determine whether a statistical comparison was made: at least five proteins from a protein set must have been present in the data, at least 10% of the proteins within a protein set must have been present in the data, and protein sets must contain fewer than 200 entries. The median loading per protein set was then transformed to a $z$ score for interpretability. The $P$ values were then converted to $q$ values using the Benjamini and Hochberg approach, and results were filtered to 5% FDR. For single-condition PSEA, the PC-based PSEA performed in Fig. 4b was applied to each treatment group separately in Fig. 5a. For the PSEA performed on the HEK and melanoma samples shown in Fig. 2e, the minimum protein count was raised to 15, due to the higher protein coverage present in that set of experiments.

**Fold-change comparison for selected GO terms.** For Fig. 4c, the unimputed, normalized, batch-corrected protein abundances for single-cell and bulk BMDM samples was subset to feature only proteins that were annotated with the following GO terms: proton transport, V-type ATPase; type I IFN signaling; hydrogen peroxide catabolic process; hydrolase activity, hydrolyzing $O$-glycosyl compounds; mitochondrial electron transport, NADH to ubiquinone; phagocytosis; positive regulation of nitric oxide biosynthetic process; and S100 protein binding. Type I IFN signaling and proton transport, V-ATPase were chosen as they correspond to the GO terms with the largest absolute effect sizes for PC1 and PC2, respectively, in Fig. 4b. The remaining GO terms were selected from a subset of GO terms found to be differentially abundant ($q$ value ≤ 0.05) between treatment conditions using the Wilcoxon rank-sum test on the vectors of unimputed, normalized, batch-corrected single-cell protein abundances. Single-cell protein-abundance measurements were then averaged on a treatment condition basis, and the LPS-to-untreated ratio of these average measurements was calculated. The LPS-to-untreated protein-abundance ratio was also calculated for bulk samples, and ratios for the single-cell and bulk samples were then plotted. Proteins not annotated with proton transport or type I IFN signaling were indicated as 'other GO terms' in the figure legend. A version of this figure with each GO term color coded appears as Supplementary Fig. 2. The Spearman correlation between the vector of single-cell protein fold changes and the vector of bulk protein fold changes was computed.

**Scatterplot of V-type ATPase-associated protein abundances.** For Fig. 4c, the unimputed, normalized, batch-corrected single-cell protein abundances for ATP6V1E1 and ATP6V1B2 were subset by treatment condition and plotted against one another. The Spearman correlation between the vectors of protein abundances for ATP6V1E1 and ATP6V1B2 was calculated on a per-treatment condition basis.

**Endocytosis analysis, histogram.** For Fig. 5b, the vectors of dextran–AF568 MFI and event counts were retrieved from the Sony MA900 instrument used to sort the dextran-uptake subpopulations, and these data were filtered for MFIs greater than 1,000 and less than 50,000. MFIs were then $\log_{10}$ transformed and plotted as a normalized histogram in Fig. 5b and Extended Data Fig. 10 for the LPS-treated samples and untreated samples, respectively.

**Endocytosis analysis, volcano plot.** For Fig. 5b, differentially abundant proteins between the low- and high dextran-uptake samples were identified via the DIA differential protein-analysis script introduced earlier, and the results were plotted, such that proteins with |$\log_2$ (fold change)| > 3 were annotated. The volcano plots associated with the LPS-treated samples and untreated samples are presented in Fig. 5c and Extended Data Fig. 10, respectively.

**Endocytosis analysis, PCA color coded by endocytic proteins.** For Fig. 5c, for each treatment condition (24 h, treated with LPS or untreated), the set of proteins with statistically significant fold changes between the high and low dextran-uptake conditions (|$\log_2$ (fold change)| ≥ 1; fold-change $q$ value ≤ 0.01) was intersected with the set of quantified proteins for the respective set of single-cell samples. The median abundance per cell was calculated for the sets of proteins associated with low dextran uptake or high dextran uptake, each vector of median abundances was then $z$ scored, extreme values were capped at a $z$ score of ±2, and the single-condition PCAs were color coded by these $z$-scored protein abundances. The figures associated with the LPS-treated samples and untreated samples are presented in Fig. 5c and Extended Data Fig. 10, respectively.

**Validation of MEROPS peptide quantitation.** For Fig. 6a, MEROPS substrates that were N-terminally labeled with TMTpro in the bulk discovery experiments and for which the cross-condition fold change was comparable between the bulk and single-cell experiments were taken to be validated measurements.

To benchmark the relative quantitation between the bulk discovery and single-cell samples, the treatment condition-associated bulk samples detailed in the MEROPS bulk validation experiments, BMDMs, above, were filtered to an elution group PEP ≤0.01, and entries mapping to the same precursor species were condensed such that the observation with the highest intensity was taken to be representative. The LPS-treated and untreated samples were then joined by precursor, and the abundances were column (sample) and row (precursor) normalized by median and mean, respectively. The ratio of the normalized precursor abundances between the LPS-treated and untreated samples were then calculated. The matrix of batch-corrected unimputed protein abundances per single cell from the prioritized BMDM analyses was then condensed to a representative abundance by treatment condition by taking the median protein abundance across all cells from a treatment group. The relative protein-abundance ratio between treatment conditions was then computed, and the vector of fold changes was subset for the MEROPS cleavage products detected with TMTpro labeling of the neo-N termini in the bulk experiments.

**Biological annotation of MEROPS peptides.** For Fig. 6b, using the MaxQuant evidence.txt output from the DDA analysis of the bulk TMTpro-labeled duplex sample containing LPS-treated (24 h) (127C) and untreated (128N) BMDM samples (wGH215.raw), proteins that were differentially abundant between treatment conditions were identified in the following way: search results were filtered to contain precursors with PEPs ≤0.02 and PIFs >0.8, and reverse matches and contaminants were filtered out; the reporter ion intensities for the two samples were column and row normalized by their means; the per-protein distributions of relative precursor abundances for each sample were subjected to a two-sided Wilcoxon rank-sum test; $P$ values were FDR corrected

via the Benjamini and Hochberg approach and filtered to 1% FDR; the median relative abundance of the precursors mapping to a given protein were taken to reflect the relative abundance of that protein; differential proteins for which the relative abundance ratio (LPS-treated/ untreated) was >1 were annotated as marker proteins associated with LPS treatment or the untreated condition otherwise.

A second set of marker proteins associated with pro-inflammatory M1-like macrophages or anti-inflammatory M2-like macrophages determined by transcriptomic analysis of monocytes, intermediate macrophages, fully differentiated macrophages, classically activated macrophages and alternatively activated macrophages was also used in this analysis[42]. Genes with a $\log_2$ (M1/M2 ratio) greater than zero in the publication-associated database were annotated as M1 associated, while genes with an M1/M2 ratio less than zero were annotated as M2 associated.

Actin L104 cleaved by cathepsin (fragment 1), citrate synthase H26 cleaved by cathepsin E (fragment 2) and actin L288 cleaved by cathepsin D (fragments 1 and 2), which had been validated via bulk analysis, were then tested for significant associations with either the treatment condition-associated protein panels or the M1- and/or M2-associated protein panels using a permutation test.

The matrix of single cells by batch-corrected unimputed protein abundances was filtered to contain the four MEROPS cleavage products and proteins annotated to the supplied list of marker proteins (either treatment condition specific or macrophage polarization specific), and a protein–protein correlation matrix was produced from this filtered matrix, using Pearson's r as the correlation metric. The median correlation was then calculated between each MEROPS cleavage product and the set of proteins annotated with either of the two reference conditions (treated with LPS or untreated; M2 or M1), and the difference between the median correlation associated with each treatment condition was recorded.

The same procedure was then repeated 10,000 times, permuting the column names of the cells by the protein matrix each time. The P value of the original correlation distance was subsequently determined to be the fraction of times that a correlation distance as extreme as the one initially observed was generated by chance alone. The set of P values was then FDR corrected using the Benjamini and Hochberg approach. If the original P value was zero, meaning no value generated by chance was as extreme as the initially observed value, then a q value of $10^{-5}$ was used.

### Extended data figures
**Fraction of inclusion list precursors detected and analyzed.** For Extended Data Fig. 1, the MaxQuant.Live log files associated with the comparison of MaxQuant.Live with and without prioritization (Fig. 1b,c) were imported into the R environment, and the lists of precursors detected by MaxQuant.Live during the survey scan and subsequently sent for $MS^2$ were extracted from the log files. The unique numeric precursor ID was then matched to the associated inclusion list for each experiment to generate Extended Data Fig. 1. The precursors detected during the survey scan in each experiment are shown in Extended Data Fig. 1a, while the precursors sent for $MS^2$ analysis in each experiment are shown in Extended Data Fig. 1b.

**Quantification variability across single-cell and control samples.** For Extended Data Fig. 2, within the single-cell pipeline[3,33], the CV (that is, the standard deviation scaled by the mean) was computed for the relative abundances of all filtered precursors that mapped to a given leading razor protein on a per-sample basis (precursor-filtration metrics are discussed in detail in the data-filtration and -normalization sections). The mean protein-level CV was then calculated on a per-sample basis, and a CV threshold was chosen that well separated the control samples from the single-cell samples. The distribution of CVs for single-cell and control samples analyzed by shotgun LC–MS/MS methods associated with Fig. 2a–e is shown in Extended Data Fig. 2a; the distribution of CVs

for single-cell and control samples analyzed by pSCoPE associated with Fig. 2a is shown in Extended Data Fig. 2b; the distribution of CVs for single-cell and control samples analyzed by pSCoPE associated with Fig. 2b–e is shown in Extended Data Fig. 2c; the distribution of CVs for single-cell and control samples associated with Figs. 4–6 is shown in Extended Data Fig. 2d.

**Peptide properties and ID rates.** For Extended Data Fig. 3, the shotgun and prioritized search results for the technical consistency experiments (Fig. 2c,d) were imported into the R environment, and the set of precursors not identified at 1% FDR in the prioritized analyses was determined. The median spectral confidence of identification and number of matching fragments for each of the precursors in this set was then calculated across all shotgun experiments using the evidence.txt and msms.txt files, respectively, and plotted in Extended Data Fig. 3.

**Fraction of inclusion list precursors detected and analyzed.** For Extended Data Fig. 4, the MaxQuant.Live log files associated with the technical coverage and consistency experiments (Fig. 2a–e) were imported into the R environment, and the lists of precursors detected by MaxQuant.Live during the survey scan and subsequently sent for $MS^2$ were extracted from the log files. The unique numeric precursor ID was then matched to the associated inclusion list for each experiment to generate Extended Data Fig. 5. The MaxQuant.Live log statistics associated with Fig. 2a are shown in Extended Data Fig. 5a, while those associated with Fig. 2b–e are shown in Extended Data Fig. 5b.

**MaxQuant iMBR and pSCoPE contrast.** For Extended Data Fig. 5, the eight shotgun and eight prioritized sample analyses associated with Fig. 2a were selected for this contrast. While the search parameters associated with the prioritized sample analyses were unchanged, the eight shotgun analyses were searched using the same search parameters as indicated previously, except that iMBR was enabled. The set of identifications generated from the pSCoPE experiments was filtered at 1% FDR, and reverse matches and contaminants were removed. To generate the box plots in Extended Data Fig. 5a, the number of PSMs per experiment of type 'MULTI-MATCH' and 'MULTI-MATCH-MSMS' were summed to generate the number of MBR matches per experiment in the 'All Precursors' facet. Only the PSMs of type 'MULTI-MATCH-MSMS' were used to generate the number of MBR matches per experiment in the 'Precursors with $MS^2$ Scans' facet. To generate the box plots in Extended Data Fig. 5b, the following data-handling steps were carried out: for the shotgun experiments, the set of all PSMs at 1% FDR was selected, all reverse matches and contaminants were removed, and all forward sequences identified by iMBR were added to this data frame; the evidence file for the prioritized analyses was filtered at 1% FDR; the precursors identified by iMBR in the shotgun runs were intersected with the prioritized inclusion list, and this group of peptides was then subset from both the shotgun and prioritized datasets; the number of experiments that each of these precursors was identified in was then tallied and faceted by priority level.

**BMDM technical comparison.** For Extended Data Fig. 6, the matrix of precursor abundances by single-cell samples for 20 shotgun analyses and 20 pSCoPE analyses was condensed to the peptide level by summing the relative intensities across charge states on a sample-specific basis. This was conducted as a means to determine peptides without detectable reporter ion signal in a given sample. The inclusion list was also condensed to the peptide level by associating a peptide sequence with the highest priority level that it appeared in. The fraction of peptides with detected reporter ion signal was calculated on a per-sample and per-priority-level basis and displayed in Extended Data Fig. 6a. To calculate the percent data completeness on a per-protein level, the same procedure was followed for condensing precursors to proteins and for associating proteins with priority levels.

To calculate the number of peptides with detectable reporter ion signal per single cell, the matrix of precursor abundances by single-cell samples for 20 shotgun analyses and 20 pSCoPE analyses was condensed to the peptide level as performed previously, and the number of non-NA values was tallied on a per-single-cell basis. The number of proteins with detectable reporter ion signal per single cell was calculated from the matrix of unimputed protein abundances per cell. These tallies are displayed in Extended Data Fig. 6b.

To generate the histogram of representative precursor abundances and their corresponding fill times, the precursor intensities for all precursors identified in the shotgun and pSCoPE single-cell BMDM analyses were split into tertiles. If a high-priority precursor appeared in the bottom intensity tertile, it was allotted an $MS^2$ fill time of 1,000 ms; if a high-priority precursor appeared in the middle intensity tertile, it was allotted an $MS^2$ fill time of 750 ms; if a high-priority precursor appeared in the top intensity tertile, it was allotted an $MS^2$ fill time of 500 ms.

The matrix of precursor abundances by single-cell samples for 20 shotgun analyses and 20 pSCoPE analyses was subset to contain only those precursors that were allotted fill times of 750 ms and 1,000 ms in the pSCoPE analyses. The percent data completeness was then calculated on a per-single-cell basis for the set of precursors allotted longer fill times present in the filtered matrix of precursors by single-cell samples. The results from these analyses are presented in Extended Data Fig. 6c.

**PCA from unimputed protein-level data.** For Extended Data Fig. 7, to assess whether the qualitative trends observed in the cross-condition PCA or the PSEA based on the PCA-derived protein weight vectors were compromised by imputation, PCA was performed on the correlation matrix generated by the batch-corrected, unimputed cell × protein matrix, and the resulting PCA plot was color coded by cell type or the median relative abundances of the proteins corresponding to type I IFN signaling or phagosome maturation (Extended Data Fig. 7).

**PCA color coded by precursor-level data completeness.** For Extended Data Fig. 8, to assess whether the sample separation observed in the cross-condition PCA (shown in Fig. 4) was driven by missing data, the single-cell data points were color coded by the data-completeness percentage. The post-data-filtration matrix of precursors by single cells (filtration metrics are described in the data-processing and -normalization section) was used to calculate the data completeness on a per-sample basis. The fraction of filtered precursors with observed reporter ion signal relative to the total number of filtered precursors detected across all experiments was then multiplied by 100 to generate the data-completeness percentage.

**Flow cytometry gating parameters and staining controls.** For Extended Data Fig. 9, FSC-A, SSC-A and dextran–PE–Texas Red flow cytometry gate settings and sorting parameters were directly acquired from the Sony MA900 flow cytometry instrument. Additional detail regarding the associated experimental parameters can be found in BMDM endocytosis assay.

**Endocytosis panel for untreated BMDMs.** For Extended Data Fig. 10, this plot was constructed in the same manner as that for the main text figure corresponding to the LPS-treated (24 h) samples.

**Supplementary figures**
**Precursor-intensity comparison for prioritized analyses and fold-change comparison for selected GO terms.** For Supplementary Figs. 1 and 2, the unimputed, normalized, batch-corrected protein abundances for single-cell and bulk BMDM samples were subset to feature only proteins that were annotated to the following GO terms: proton transport, V-type ATPase; type I IFN signaling; hydrogen peroxide catabolic process; hydrolase activity, hydrolyzing *O*-glycosyl compounds; mitochondrial electron transport, NADH to ubiquinone;

phagocytosis; positive regulation of nitric oxide biosynthetic process; and S100 protein binding. Type I IFN signaling and proton transport, V-ATPase were chosen as they correspond to the GO terms with the largest absolute effect sizes for PC1 and PC2, respectively, in Fig. 4b. The remaining GO terms were selected from a subset of GO terms found to be differentially abundant ($q$ value ≤ 0.05) between treatment conditions using a Kruskal–Wallis test on the vectors of unimputed, normalized, batch-corrected single-cell protein abundances. Single-cell protein-abundance measurements were then averaged on a treatment condition basis, and the LPS/untreated ratio of these average measurements was calculated. The LPS/untreated protein-abundance ratio was also calculated for the bulk samples, and the ratios for the single-cell and bulk samples were then plotted. Proteins not annotated to proton transport or type I IFN signaling were indicated as 'other GO terms' in the figure legend. A version of this figure with each GO term color coded appears as Supplementary Fig. 2. The Spearman correlation between the vector of single-cell protein fold changes and the vector of bulk protein fold changes was computed.

**Protein–protein correlations associated with the V-type ATPase.** For Supplementary Fig. 3, the unimputed, normalized, batch-corrected single-cell protein abundances for all V-type ATPase-associated quantified proteins were subset by treatment condition, and Spearman correlations were computed between vectors of single-cell protein abundances.

**Reporting summary**
Further information on research design is available in the Nature Portfolio Reporting Summary linked to this article.

## Data availability
Metadata, raw data and processed data are organized according to community recommendations[34] and are freely available at MassIVE: MSV000090383. Data and code necessary for regenerating all figures are freely available on Zenodo at https://doi.org/10.5281/zenodo.7498141. Data are also available at https://scp.slavovlab.net/Huffman_et_al_2022.

## Code availability
Code and protocols are organized according to community recommendations[34] and are available at https://scp.slavovlab.net/pSCoPE and https://github.com/SlavovLab/pSCoPE. Additionally, the GitHub repository containing all code necessary for regenerating the figures in this publication has been archived on Zenodo at https://doi.org/10.5281/zenodo.7498171.

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

## Acknowledgements

We thank L. Reiter and T. Gandhi for making Spectronaut available for this project and to T. Colombani and M. Jovanovic for discussions and constructive comments. This work was funded by a New Innovator Award from the NIGMS from the National Institutes of Health to N.S. under award number DP2GM123497, an Allen Distinguished Investigator Award through the Paul G. Allen Frontiers Group to N.S., a Seed Networks Award from CZI CZF2019-002424 to N.S. and through a Merck Exploratory Science Center Fellowship, Merck Sharpe & Dohme to N.S and an R01 award from NIGMS from the National Institutes of Health to N.S. under award number R01GM144967.

## Author contributions

Experimental design: R.G.H. and N.S. LC–MS/MS: R.G.H., A.L., H.S., J.D., S.K., A.A.P. and E.E. Sample preparation: R.G.H., A.L., M.d.G. and F.B. Cell culture: A.L., M.d.G., F.B., S.K., L.K. and R.G.H. Raising funding: N.S. Supervision: N.S. Data analysis: R.G.H. and N.S. Software development: C.W. Initial draft: R.G.H. and N.S. Result interpretation: N.S., J.C., I.Z., R.G.H., A.L., H.S., J.D., S.K., M.d.G., A.A.P. and D.H.P. Writing: all authors approved the final paper.

## Competing interests

N.S. is a founding director and CEO of Parallel Squared Technology Institute, which is a nonprofit research institute.

## Additional information

**Extended data** is available for this paper at https://doi.org/10.1038/s41592-023-01830-1.

**Correspondence and requests for materials** should be addressed to Nikolai Slavov.

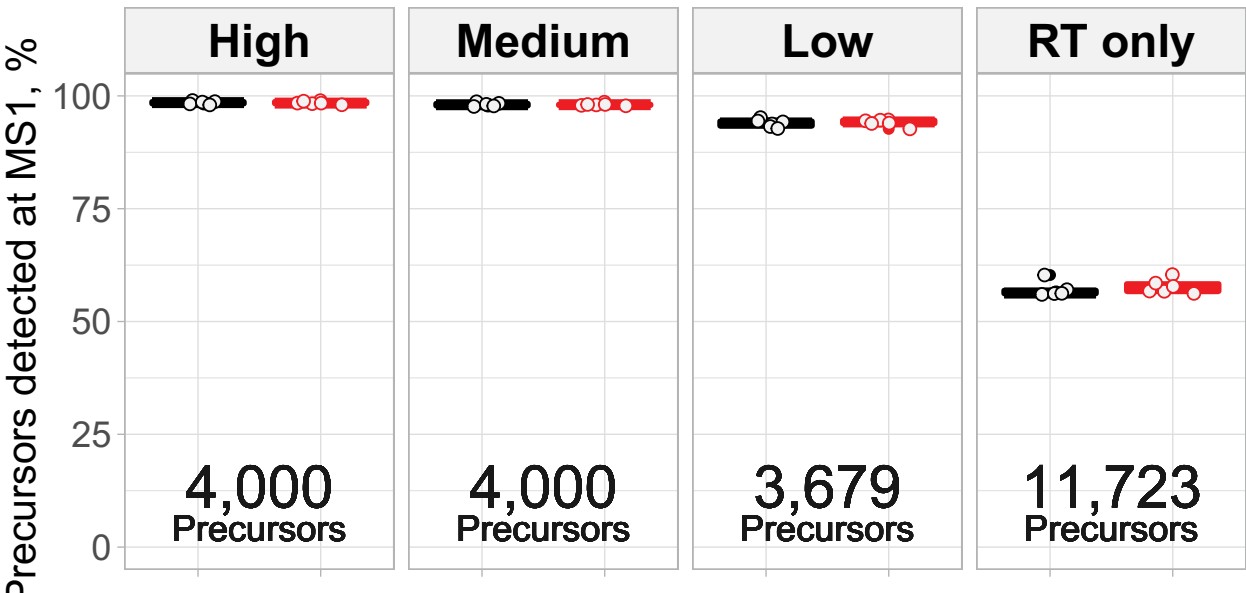

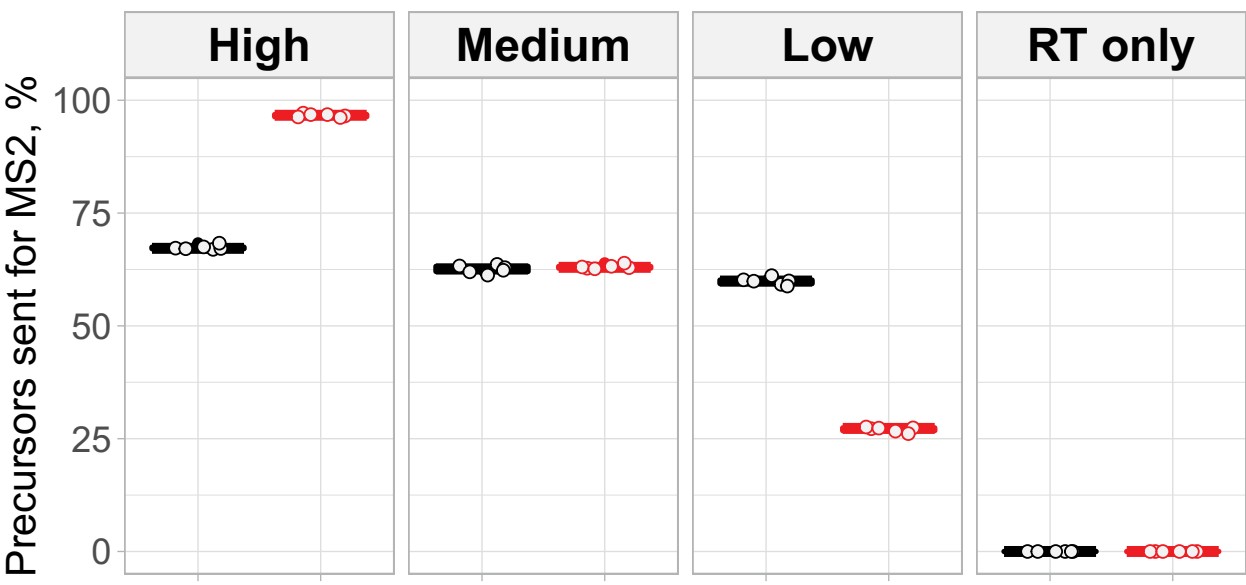

**Extended Data Fig. 1 | Percent of inclusion-list precursors detected and analyzed in platform benchmark runs, for MaxQuant.Live with and without prioritization enabled.** (a) MS1 detection rates for precursors in the platform benchmark experiments displayed in Fig. 1a, b. Data collected using MaxQuant.Live in default mode are shown in black, while data collected using prioritization are shown in red. The precursor count displayed at the bottom of each priority level's facet corresponds to the number of precursors present on that priority level of the inclusion list. (b) MS2 analysis rates for precursors in the platform benchmark experiments displayed in Fig. 1a, b. While the MS1 precursor detection rates are similar for both platforms, the MS2 analysis rates are correlated to the priority levels for prioritized analysis, but not for default MaxQuant.Live analyses. Each boxplot shown above contains 6 data points, one for each LC-MS/MS analysis. For all boxplots, whiskers display the minimum and maximum values within 1.5 times the interquartile range of the 25th and 75th percentiles, respectively; the 25th percentile, median, and 75th percentile are also featured.

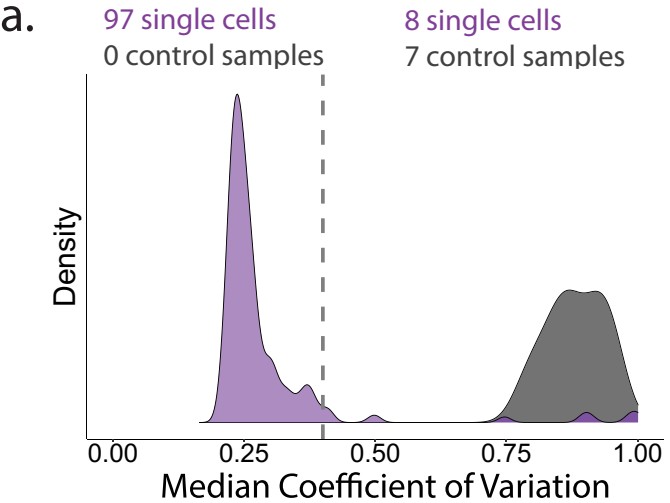

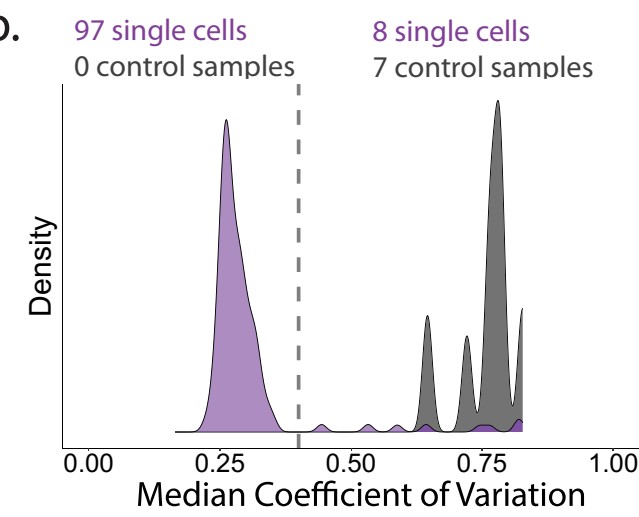

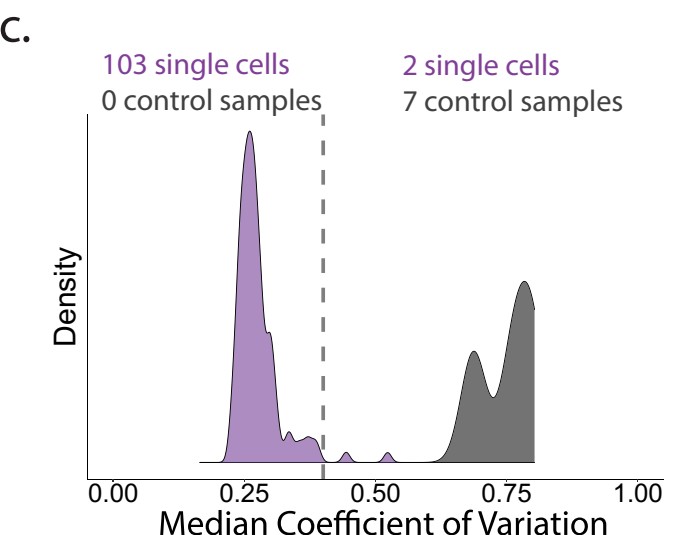

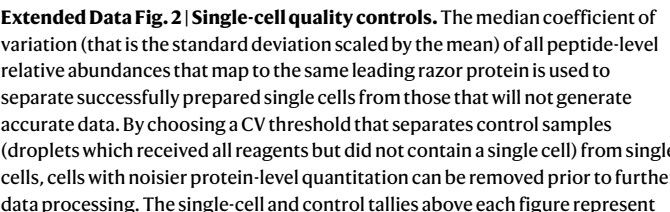

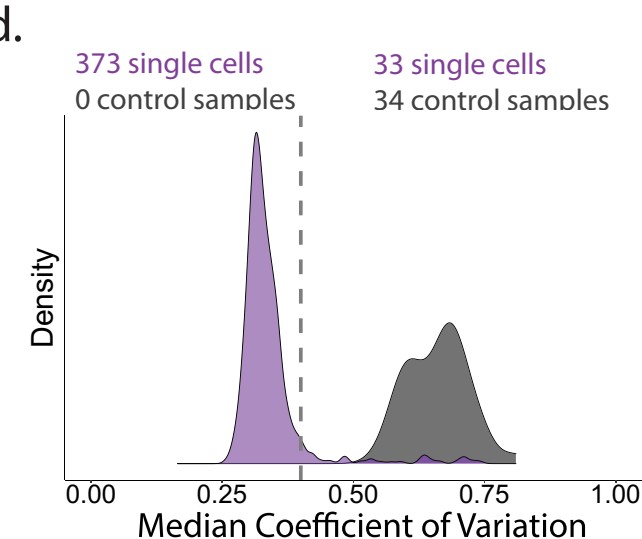

**Extended Data Fig. 2 | Single-cell quality controls.** The median coefficient of variation (that is the standard deviation scaled by the mean) of all peptide-level relative abundances that map to the same leading razor protein is used to separate successfully prepared single cells from those that will not generate accurate data. By choosing a CV threshold that separates control samples (droplets which received all reagents but did not contain a single cell) from single cells, cells with noisier protein-level quantitation can be removed prior to further data processing. The single-cell and control tallies above each figure represent the number of single cells or control wells that passed the CV threshold of 0.4. (a) contains the CV distributions for the single-cell samples associated with Fig. 2a–e, analyzed by shotgun LC-MS/MS methods. (b) contains the CV distributions for the single-cell samples associated with Fig. 2a, analyzed by pSCoPE. (c) contains the CV distributions for the single-cell samples associated with Fig. 2b–e, analyzed by pSCoPE. (d) contains the CV distributions for the single-cell samples associated with Figs. 4–6.

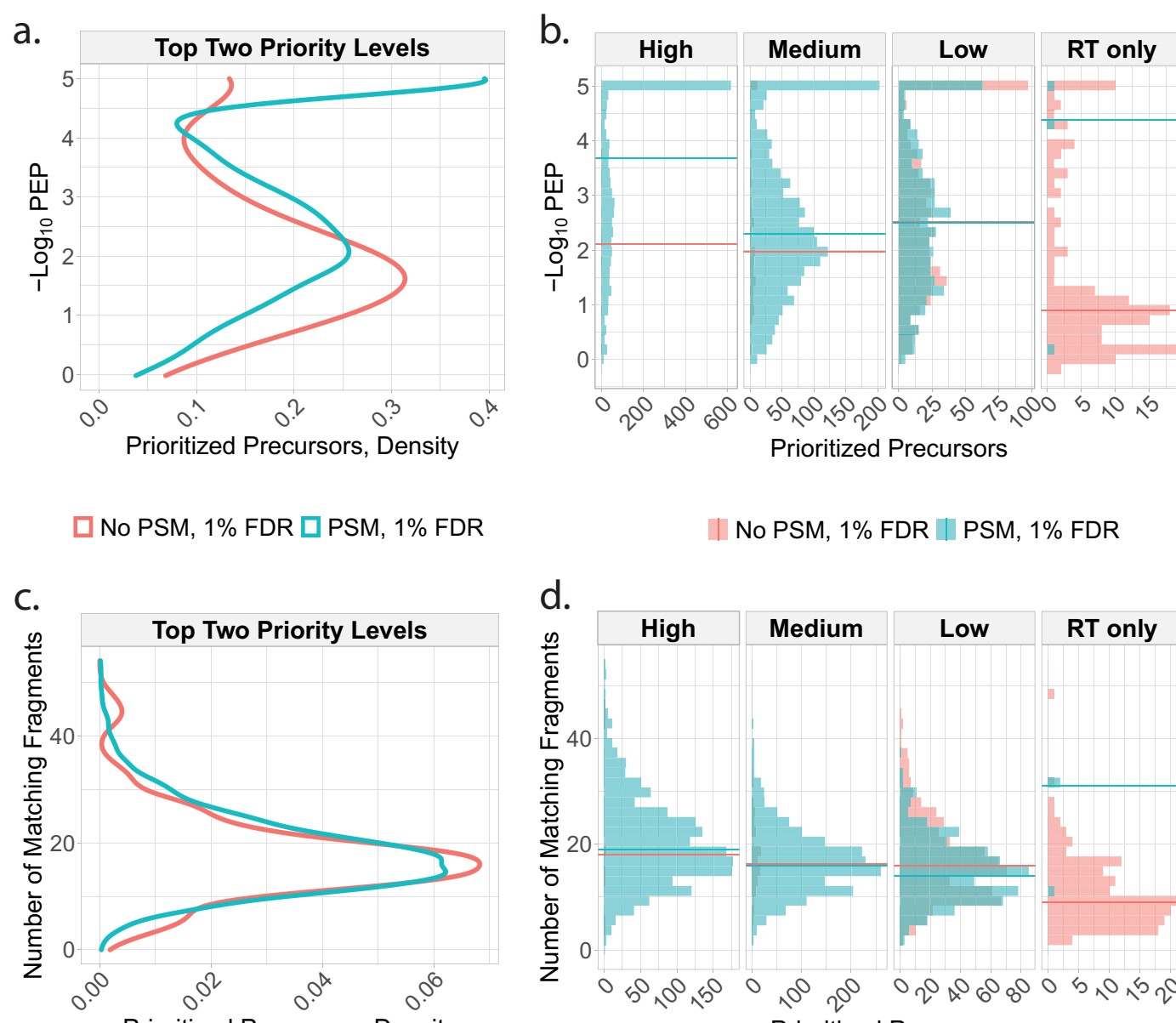

**Extended Data Fig. 3 | Properties of peptides successfully identified in pSCoPE runs.** The precursors from the inclusion list were split into those that resulted in confident PSMs and those that did not, and the properties of each set analyzed based on the shotgun runs used for making the inclusion lists. (a) Confidence of identification (quantified by the posterior error probability; PEP) and number of matching peptide fragments for successful and unsuccessful precursors. The data are shown for all prioritized peptides across all priority tiers. (b) The data from panel a are shown faceted by priority tier. All data shown are from the consistency experiments from Fig. 2c. In previous analyses conducted during a period of suboptimal instrument performance, the number of matching fragments was shown to effectively distinguish between the peptides which were identified at 1\% FDR and those that were not identified, which was reported in version 1 of our preprint. This trend is not observed in the current dataset, which was acquired by the same instrument but with more efficient ion isolation by its quadrupole, (c) and (d).

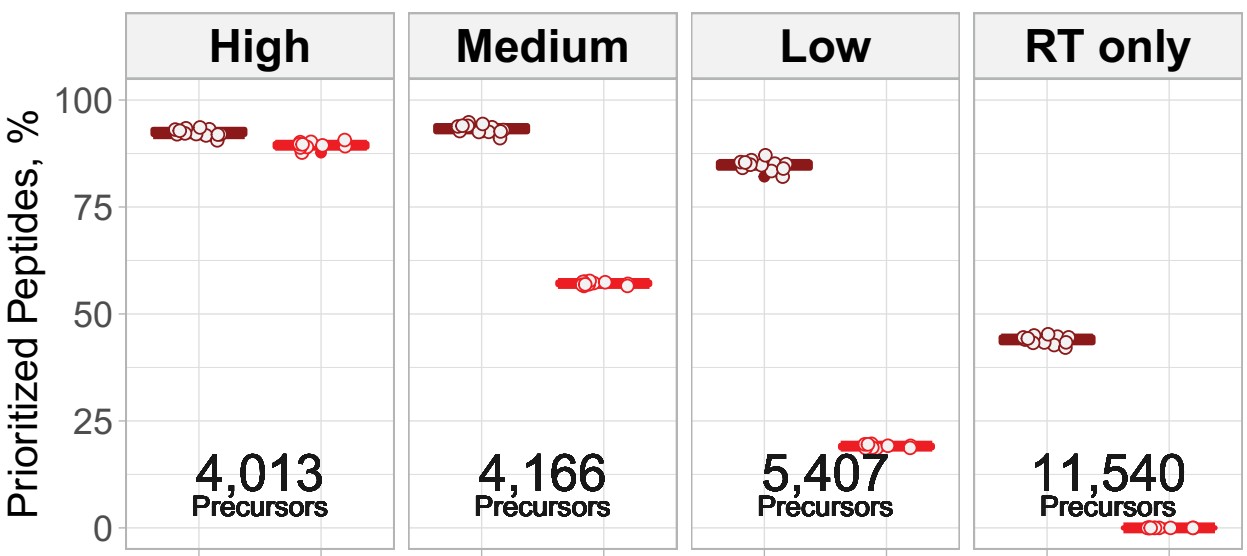

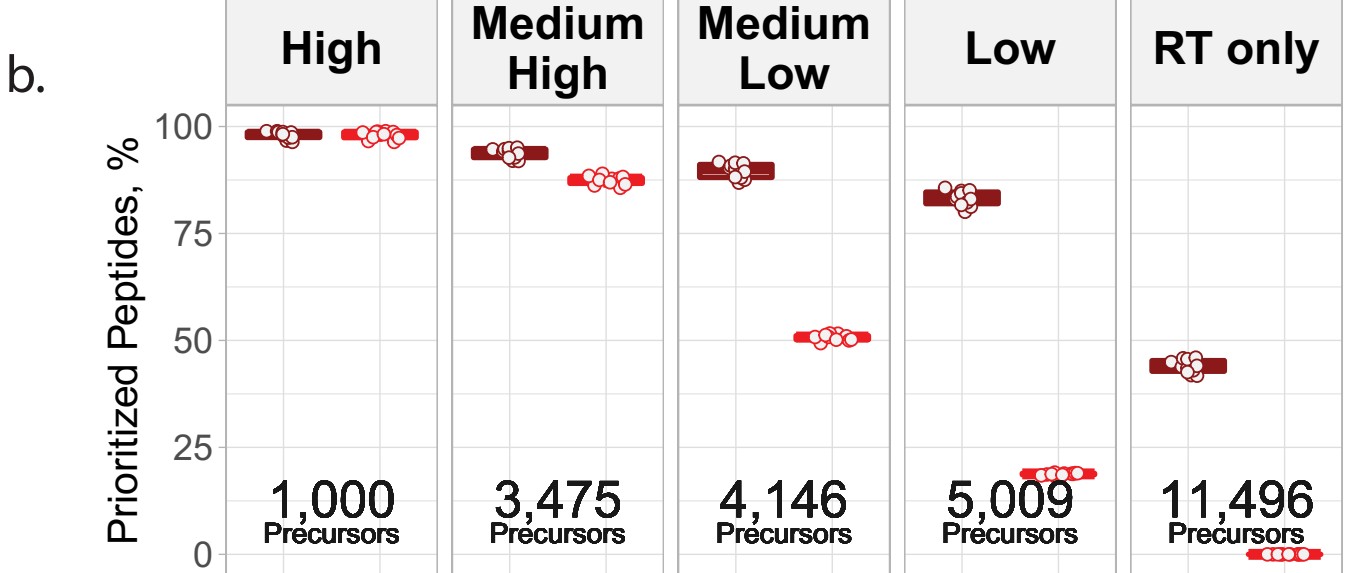

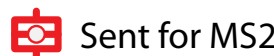

**Extended Data Fig. 4 | Fraction of inclusion-list precursors detected and analyzed in pSCoPE runs.** (a) MS1 detection and MS2 analysis rates for prioritized precursors in the benchmark experiments displayed in Fig. 2a. Each boxplot contains 8 data points, one for each LC-MS/MS analysis. (b) MS1 detection and MS2 analysis rates for prioritized precursors in the benchmark experiments displayed in Fig. 2b–e. Each boxplot contains 8 data points, one for each LC-MS/MS analysis. In both panels, the statistics are shown for each tier along with the number of precursors in the tier. Boxplot whiskers display the minimum and maximum values within 1.5 times the interquartile range of the 25th and 75th percentiles, respectively; the 25th percentile, median, and 75th percentile are also featured.

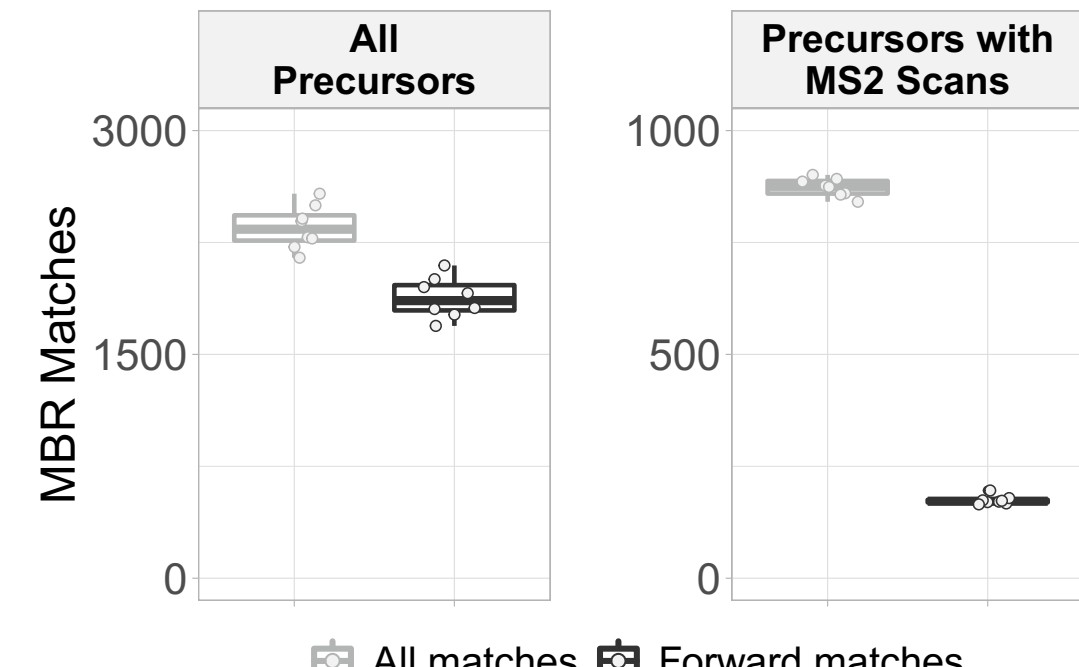

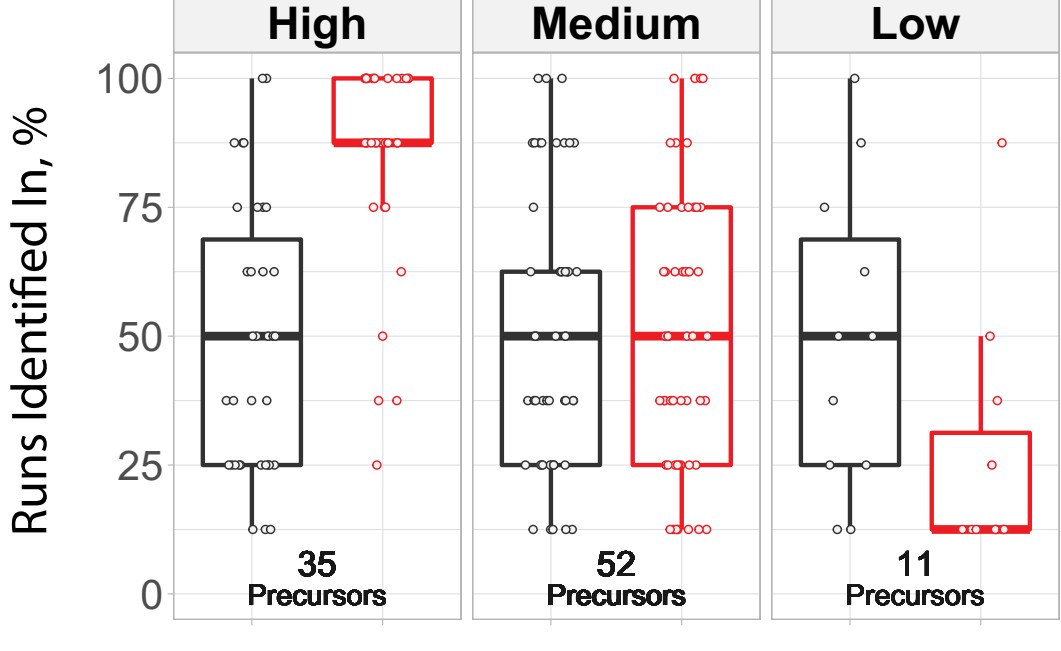

**Extended Data Fig. 5 | pSCoPE outperforms isobaric Match Between Runs (iMBR) for increasing consistency of identification across single-cell experiments.** (a) The 'All Precursors' facet heading indicates the total number of MBR-facilitated precursor identifications in each of 8 shotgun analyses. The 'Precursors with MS2 Scans' facet heading indicates the total number of MBR-facilitated precursor identifications that are associated with MS2 scans, enabling reporter ion quantitation. In both facets, the identifications are segmented into 'All matches', a category which includes matches to reverse sequences, and 'Forward matches', which does not. Each point represents an experiment.

Data derived from shotgun experiments shown in Fig. 2a–e. (b) The intersected precursors between the MBR-facilitated forward sequence matches and the corresponding prioritized analyses were then compared based on consistency of identification across the 8 experiments associated with each acquisition method. Each point represents a precursor. Data derived from shotgun and pSCoPE analyses shown in Fig. 2a. Boxplot whiskers display the minimum and maximum values within 1.5 times the interquartile range of the 25th and 75th percentiles, respectively; the 25th percentile, median, and 75th percentile are also featured.

a.
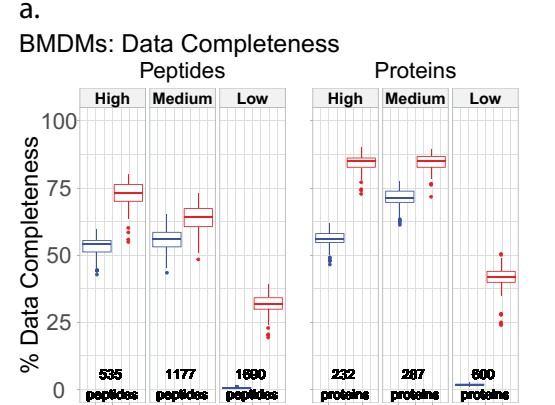

b.
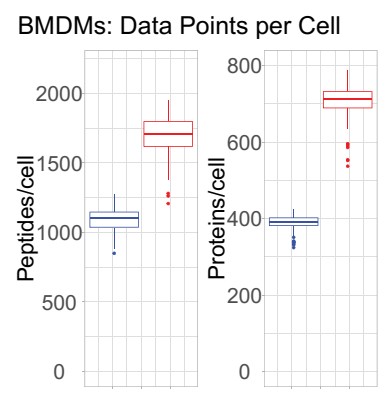

c.
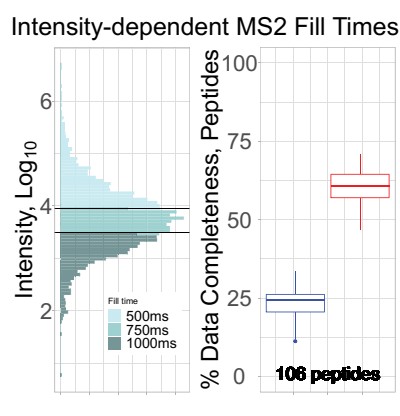

Shotgun Prioritization

**Extended Data Fig. 6 | Data completeness and proteome coverage for BMDMs analyzed by shotgun or prioritized methods.** (a) Percent data completeness tallied for peptides and proteins quantified across twenty shotgun and twenty pSCoPE experiments, faceted by priority tier. n = 175 and 186 single-cells for the prioritized and shotgun analysis methods, respectively. (b) Number of peptides and proteins per single-cell sample across twenty shotgun and twenty pSCoPE experiments. n = 175 and 186 single-cells for the prioritized and shotgun analysis methods, respectively. (c) Illustration of precursor-intensity-dependent MS2 fill times for precursors on the top priority tier. Percent data completeness contrast for precursors which were allotted increased fill times in the pSCoPE analyses. n = 175 and 186 single-cells for the prioritized and shotgun analysis methods, respectively. Boxplot whiskers display the minimum and maximum values within 1.5 times the interquartile range of the 25th and 75th percentiles, respectively; the 25th percentile, median, and 75th percentile are also featured.

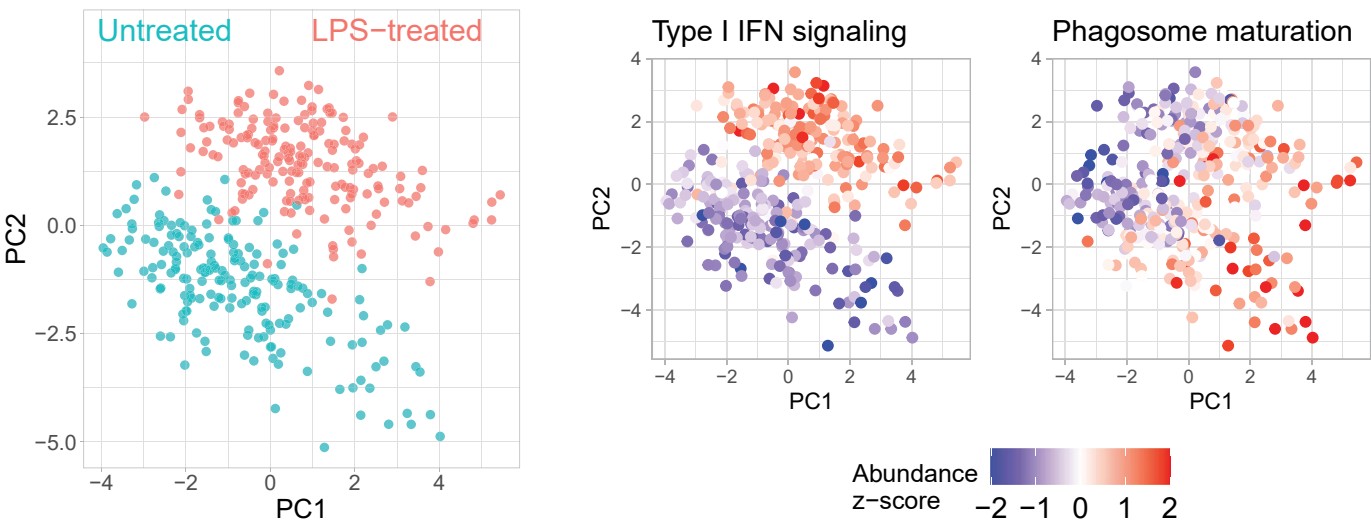

**Extended Data Fig. 7 | PCA of BMDMs using only observed data points.** To evaluate the robustness of our results to uncertainties stemming from missing data, we performed PCA of unimputed BMDM data. The single cells are color-coded by treatment condition, with adjoining PCA plots color-coded by the median relative abundance of proteins corresponding to type I interferon-mediated signaling and phagosome maturation.

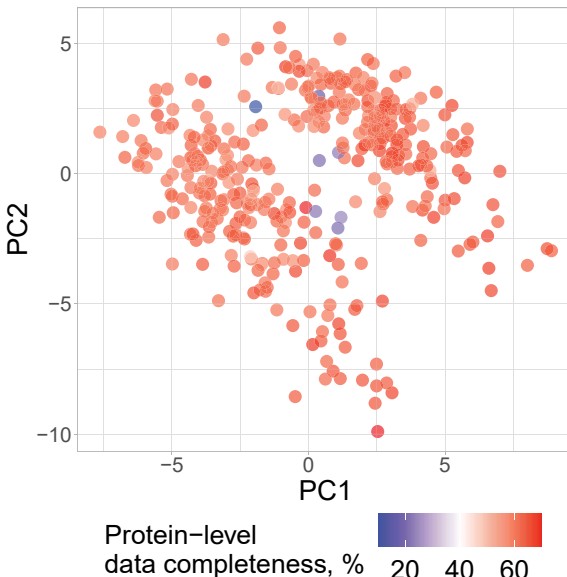

**Extended Data Fig. 8 | PCA color-coded by protein-level data completeness.** To evaluate whether the biological conclusions we drew from our PC-weight-based PSEA could have been influenced by separation due to data completeness, we color-coded our cross-condition BMDM PCA by the percent data completeness on a per-cell basis.

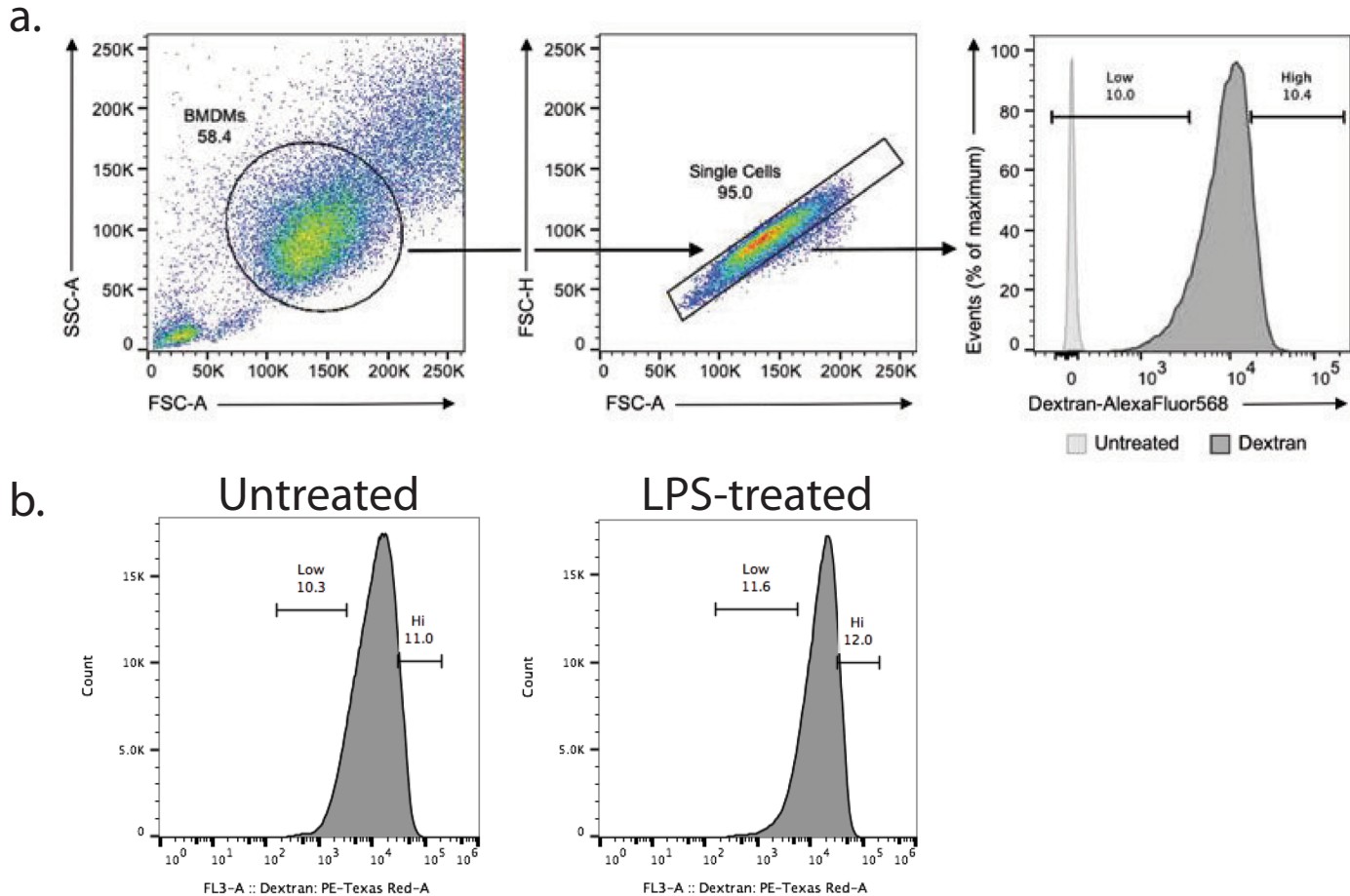

**Extended Data Fig. 9 | FACS gating parameters and staining controls.** (a) FSC-A and SSC-A gates for sorted bone-marrow-derived macrophages and positive/negative staining populations. (b) Dextran:PE-Texas Red gating parameters for isolating the most and least endocytic BMDM populations from each treatment group (untreated and LPS-treated).

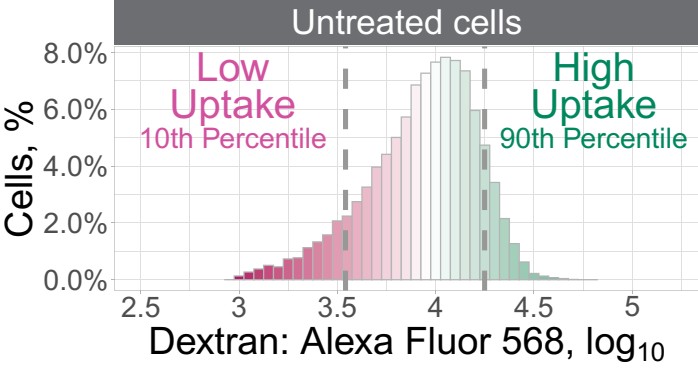

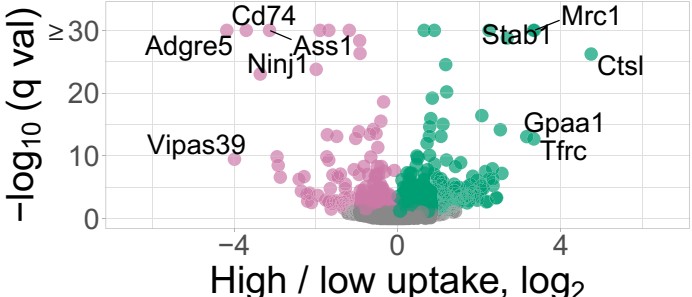

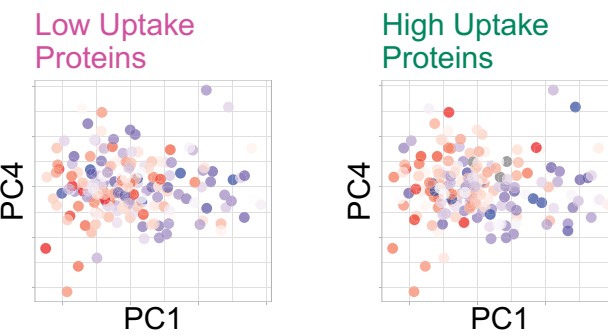

**Extended Data Fig. 10 | Dextran uptake in untreated BMDM samples.** The uptake of fluorescent dextran by the untreated macrophages was measured by FACS, and the cells with the lowest and highest uptake were isolated for protein analysis. The volcano plot displays the fold changes for differentially abundant proteins and the associated statistical significance. The untreated macrophages were displayed in the space of their PCs and color-coded by the median abundance of the low-uptake or the high-uptake proteins. Both the low and the high-uptake proteins correlate inversely to PC1 (low-uptake: Spearman r = −0.29, q <= 6×10⁻⁴; high-uptake: Spearman r = −0.37, q <= 4×10⁻⁶).

# Reporting Summary

## Statistics

For all statistical analyses, confirm that the following items are present in the figure legend, table legend, main text, or Methods section.

| n/a | Confirmed | |
|---|---|---|
| ☐ | ☒ | The exact sample size (*n*) for each experimental group/condition, given as a discrete number and unit of measurement |
| ☐ | ☒ | A statement on whether measurements were taken from distinct samples or whether the same sample was measured repeatedly |
| ☐ | ☒ | The statistical test(s) used AND whether they are one- or two-sided *Only common tests should be described solely by name; describe more complex techniques in the Methods section.* |
| ☐ | ☒ | A description of all covariates tested |
| ☐ | ☒ | A description of any assumptions or corrections, such as tests of normality and adjustment for multiple comparisons |
| ☐ | ☒ | A full description of the statistical parameters including central tendency (e.g. means) or other basic estimates (e.g. regression coefficient) AND variation (e.g. standard deviation) or associated estimates of uncertainty (e.g. confidence intervals) |
| ☐ | ☒ | For null hypothesis testing, the test statistic (e.g. *F*, *t*, *r*) with confidence intervals, effect sizes, degrees of freedom and *P* value noted *Give P values as exact values whenever suitable.* |
| ☒ | ☐ | For Bayesian analysis, information on the choice of priors and Markov chain Monte Carlo settings |
| ☐ | ☒ | For hierarchical and complex designs, identification of the appropriate level for tests and full reporting of outcomes |
| ☐ | ☒ | Estimates of effect sizes (e.g. Cohen's *d*, Pearson's *r*), indicating how they were calculated |

*Our web collection on statistics for biologists contains articles on many of the points above.*

## Software and code

Policy information about availability of computer code

| Data collection | All mass-spec data were collected by Xcalibur™ (V_4.2, V_4.3), which is commercial software by Thermo Fisher Scientific, or MaxQuant.live (version 2.1), a published software platform for data acquisition on Thermo Quadrupole instruments (DOI:https://doi.org/10.1074/mcp.TIR118.001131) |
|---|---|
| Data analysis | Raw DDA mass-spec data were searched by MaxQuant (V_2.0.30, V_1.6.7.0, V_1.6.17.0 ) and all DIA data were searched by Spectronaut (v_15.1, v_15.0, v_14.1, v_14.10 & v_15.6) or DIA-NN (V_1.8.2 beta 2, V_1.8.1 beta 23) The output of MaxQuant, Spectronaut, and DIA-NN were further analyzed by a data processing pipeline implemented in the R programming language, which is available at https://github.com/SlavovLab/pSCoPE. |

For manuscripts utilizing custom algorithms or software that are central to the research but not yet described in published literature, software must be made available to editors and reviewers. We strongly encourage code deposition in a community repository (e.g. GitHub). See the Nature Portfolio guidelines for submitting code & software for further information.

## Data

Policy information about availability of data

All manuscripts must include a data availability statement. This statement should provide the following information, where applicable:
- Accession codes, unique identifiers, or web links for publicly available datasets
- A description of any restrictions on data availability
- For clinical datasets or third party data, please ensure that the statement adheres to our policy

MassIVE MSV000090383 (https://massive.ucsd.edu/ProteoSAFe/dataset.jsp?task=76188b5c13714a6da059d2845fca1c64)

# Field-specific reporting

Please select the one below that is the best fit for your research. If you are not sure, read the appropriate sections before making your selection.

☒ Life sciences          ☐ Behavioural & social sciences          ☐ Ecological, evolutionary & environmental sciences

For a reference copy of the document with all sections, see nature.com/documents/nr-reporting-summary-flat.pdf

# Life sciences study design

All studies must disclose on these points even when the disclosure is negative.

| | |
|---|---|
| Sample size | For the benchmarking experiments associated with Figure 1b/e:<br>1 sample of TMT-labeled serially diluted digest featuring 6 single-cell-level samples (3 U937; 3 HEK293) with 12 replicate injections<br><br>For the benchmarking experiments associated with Figure 2abcde:<br>97 cells associated with the pSCoPE sets shown in Figure 2a (50 HEK293 cells; 47 melanoma cells)<br>103 cells associated with the pSCoPE sets shown in Figure 2b/c/d/e (51 HEK293 cells; 52 melanoma cells)<br>97 cells associated with the shotgun sets shown in Figure 2abcd (46 HEK293 cells; 51 melanoma cells)<br><br>For the spike-in analysis associated with Figure 3:<br>112 samples associated with the pSCoPE sets shown in Figure 2b (51 HEK293 cells; 53 melanoma cells; 8 control samples)<br><br>For the BMDM samples associated with Figures 4/5/6:<br>373 cells associated with the pSCoPE sets (187 untreated BMDMs; 186 24-hr LPS-treated BMDMs)<br><br>Sample sizes were chosen to include at least a dozen single cells for technical experiment to support the validity of the results and a couple hundred single cells from the primary macrophages to give us more statistical power to identify biological differences and regulatory mechanisms. |
| Data exclusions | Single cells were excluded from downstream analysis if their quantification variability (the coefficient of variation of relative peptide abundances which mapped to the same protein) exceeded 0.4, except in the case of the spike-in experiments, where the reporter ion intensities derived from the spike-in peptides were of primary interest. |
| Replication | For the benchmarking experiments associated with Figure 1b/c:<br>6 MaxQuant.Live experiments without prioritization enabled<br>6 MaxQuant.Live experiments with prioritization enabled<br><br>For the benchmarking experiments associated with Figure 2abcde:<br>8 pSCoPE experiments shown in Figure 2a<br>8 pSCoPE experiments shown in Figure 2b/c/d/e<br>8 Shotgun experiments shown in Figure 2a/b/c/d<br><br>For the spike-in analysis associated with Figure 3:<br>8 pSCoPE experiments<br><br>For the BMDM samples associated with Figures 4/5/6:<br>40 pSCoPE experiments<br><br>*Each experiment contained up to 14 single-cell samples |
| Randomization | Single cells, control samples, and TMT-label application were randomized within each 14 drop array that constituted an experimental set for the HEK293 and Melanoma samples shown in Figure 2 and the BMDM samples shown in Figures 4/5/6, prepared by nano-ProteOmic sample Preparation (nPOP). For the HEK293 and melanoma single-cell samples and control samples shown in Figure 3, the spike-in peptide concentration was also randomized across samples. |
| Blinding | Blinding is not relevant to this study because having prior knowledge of which samples are being tested cannot change the raw data or the down-stream results. |

# Reporting for specific materials, systems and methods

We require information from authors about some types of materials, experimental systems and methods used in many studies. Here, indicate whether each material, system or method listed is relevant to your study. If you are not sure if a list item applies to your research, read the appropriate section before selecting a response.

## Materials & experimental systems

| n/a | Involved in the study |
|---|---|
| ☒ | ☐ Antibodies |
| ☐ | ☒ Eukaryotic cell lines |
| ☒ | ☐ Palaeontology and archaeology |
| ☐ | ☒ Animals and other organisms |
| ☒ | ☐ Human research participants |
| ☒ | ☐ Clinical data |
| ☒ | ☐ Dual use research of concern |

## Methods

| n/a | Involved in the study |
|---|---|
| ☒ | ☐ ChIP-seq |
| ☐ | ☒ Flow cytometry |
| ☒ | ☐ MRI-based neuroimaging |

## Eukaryotic cell lines

Policy information about cell lines

| | |
|---|---|
| Cell line source(s) | HEK293 cell lines purchased from ATCC (CRL-1573); melanoma cells (WM989-A6-G3) were a gift from Sydney Shaffer, University of Pennsylvania |
| Authentication | ATCC authentication report based on PCR assays with species-specific primers |
| Mycoplasma contamination | The cells were not tested for mycoplasma contamination |
| Commonly misidentified lines (See ICLAC register) | None |

## Animals and other organisms

Policy information about studies involving animals; ARRIVE guidelines recommended for reporting animal research

| | |
|---|---|
| Laboratory animals | C57BL/6J (Jax 000664) mice (female, 8 weeks old) were purchased from The Jackson Laboratory and used to generate the bone-marrow-derived macrophages |
| Wild animals | Please state here that no wild animals were used in the study. |
| Field-collected samples | Please state here that no field collected samples were used in the study. |
| Ethics oversight | All protocols were approved by the Institutional Animal Care and Use Committee (IACUC) of Boston Children's Hospital. |

Note that full information on the approval of the study protocol must also be provided in the manuscript.

## Flow Cytometry

### Plots

Confirm that:

☒ The axis labels state the marker and fluorochrome used (e.g. CD4-FITC).

☒ The axis scales are clearly visible. Include numbers along axes only for bottom left plot of group (a 'group' is an analysis of identical markers).

☒ All plots are contour plots with outliers or pseudocolor plots.

☒ A numerical value for number of cells or percentage (with statistics) is provided.

### Methodology

| | |
|---|---|
| Sample preparation | C57BL/6J (Jax 000664) mice were purchased from The Jackson Laboratory. Bone-marrow-derived macrophages (BMDMs) were differentiated from bone marrow in Dulbecco's modified Eagle medium (DMEM; Thermo Fisher Scientific), 30% L929-M-CSF supernatant and 10% fetal bovine serum (FBS). After 7 days, BMDMs were replated at 1x10^6 cells/ml in DMEM supplemented with 10% FBS, and each plate was either stimulated for 24 hours with LPS (Serotype O55:B5, Enzo Life Sciences) at 1 ug/ml or allowed to rest. <br><br> Murine BMDMs were then incubated with dextran conjugated to Alexa Fluor 568 (Thermo, D22912) at a final concentration of 0.5 mg/ml for 45 minutes at 37 degrees C. After the incubation period, cells were washed twice with 1x PBS and incubated with PBS-2mM EDTA to detach from the plate. Prior to FACS analysis, cells were spun down at 300g for 5 minutes and washed with 1x PBS before being resuspended. Using a Sony MA900, Dextran-AF568 fluorescence was then analyzed for cells from each treatment condition, and a minimum of 70,000 cells from the top and bottom 10% of the FITC-AF568 fluorescence distribution were then sorted for downstream sample preparation and mass spectrometry analysis. |
| Instrument | Sony MA900 |

| | |
|---|---|
| Software | Cell Sorter Software |
| Cell population abundance | ~80,000 cells per dextran uptake condition (bottom dextran-AF568 decile and top dextran-AF568 decile) per treatment condition (untreated and treated with LPS for 24 hours) |
| Gating strategy | The following gating strategy was used to separate singlets from debris: FSC-A from 75k to 210k, SSC-A from 35k to 170k (visible in supplemental figure 7).<br><br>For the untreated BMDMs, cells from the bottom 10.3% and top 11.0% of dextran:AP568 fluorescence distribution were isolated for the low and high uptake subpopulations, respectively. For the 24-hr LPS-stimulated BMDMs, cells from the bottom 11.6% and the top 12.0% of dextran: AP568 fluorescence distribution were isolated for the low and high uptake subpopulations, respectively |

☒ Tick this box to confirm that a figure exemplifying the gating strategy is provided in the Supplementary Information.

