## [Peer Review File · Nature Methods]

Peer Review Information

Manuscript Title: Prioritized mass spectrometry increases the depth, sensitivity, and data completeness of single-cell proteomics

Corresponding author name(s): Nikolai Slavov

Editorial Notes: n/a

Reviewer Comments & Decisions:

Decision Letter, initial version:

Dear Nikolai,

Your Article, "Prioritized single-cell proteomics reveals molecular and functional polarization across primary macrophages", has now been seen by 3 reviewers. As you will see from their comments below, although the reviewers find your work of considerable potential interest, they have raised a number of concerns. We are interested in the possibility of publishing your paper in Nature Methods, but would like to consider your response to these concerns before we reach a final decision on publication.

We therefore invite you to revise your manuscript to address these concerns. In particular, it would be a good idea to rewrite parts of the paper to focus more on the method, and its development and characterization (however, please keep the application in the main paper as well). Apart from these recommendations, please address all the other concerns raised by the reviewers.

* include a point-by-point response to the reviewers and to any editorial suggestions

* please underline/highlight any additions to the text or areas with other significant changes to facilitate review of the revised manuscript

* address the points listed described below to conform to our open science requirements

* ensure it complies with our general format requirements as set out in our guide to authors at www.nature.com/naturemethods

* resubmit all the necessary files electronically by using the link below to access your home page

[Redacted] This URL links to your confidential home page and associated information about manuscripts you may have submitted, or that you are reviewing for us. If you wish to forward this email to co-authors, please delete the link to your homepage.

We hope to receive your revised paper within 6 weeks. If you cannot send it within this time, please let us know. In this event, we will still be happy to reconsider your paper at a later date so long as nothing similar has been accepted for publication at Nature Methods or published elsewhere.

OPEN SCIENCE REQUIREMENTS

REPORTING SUMMARY AND EDITORIAL POLICY CHECKLISTS

Please note that these forms are dynamic ‘smart pdfs’ and must therefore be downloaded and completed in Adobe Reader. We will then flatten them for ease of use by the reviewers. If you would like to reference the guidance text as you complete the template, please access these flattened versions at <http://www.nature.com/authors/policies/availability.html>.

IMAGE INTEGRITY

DATA AVAILABILITY

All novel DNA and RNA sequencing data, protein sequences, genetic polymorphisms, linked genotype and phenotype data, gene expression data, macromolecular structures, and proteomics data must be deposited in a publicly accessible database, and accession codes and associated hyperlinks must be provided in the “Data Availability” section.

To further increase transparency, we encourage you to provide, in tabular form, the data underlying the graphical representations used in your figures. This is in addition to our data-deposition policy for specific types of experiments and large datasets. For readers, the source data will be made accessible directly from the figure legend. Spreadsheets can be submitted in .xls, .xlsx or .csv formats. Only one (1)

file per figure is permitted: thus if there is a multi-paneled figure the source data for each panel should be clearly labeled in the csv/Excel file; alternately the data for a figure can be included in multiple, clearly labeled sheets in an Excel file. File sizes of up to 30 MB are permitted. When submitting source data files with your manuscript please select the Source Data file type and use the Title field in the File Description tab to indicate which figure the source data pertains to.

Please include a “Data availability” subsection in the Online Methods. This section should inform readers about the availability of the data used to support the conclusions of your study, including accession codes to public repositories, references to source data that may be published alongside the paper, unique identifiers such as URLs to data repository entries, or data set DOIs, and any other statement about data availability. At a minimum, you should include the following statement: “The data that support the findings of this study are available from the corresponding author upon request”, describing which data is available upon request and mentioning any restrictions on availability. If DOIs are provided, please include these in the Reference list (authors, title, publisher (repository name), identifier, year). For more guidance on how to write this section please see: <http://www.nature.com/authors/policies/data/data-availability-statements-data-citations.pdf>

CODE AVAILABILITY

Please include a “Code Availability” subsection in the Online Methods which details how your custom code is made available. Only in rare cases (where code is not central to the main conclusions of the paper) is the statement “available upon request” allowed (and reasons should be specified).

For more information on our code sharing policy and requirements, please see: <https://www.nature.com/nature-research/editorial-policies/reporting-standards#availability-of-computer-code>

MATERIALS AVAILABILITY

SUPPLEMENTARY PROTOCOL

To help facilitate reproducibility and uptake of your method, we ask you to prepare a step-by-step Supplementary Protocol for the method described in this paper. We [encourage authors to share their step-by-step experimental protocols](https://www.nature.com/nature-research/editorial-policies/reporting-standards#protocols) on a protocol sharing platform of their choice and report the protocol DOI in the reference list. Nature Research's Protocol Exchange is a free-to-use and open resource for protocols; protocols deposited in Protocol Exchange are citable and can be linked from the published article. More details can found at www.nature.com/protocolexchange/about.

ORCID

Sincerely,
Arunima

Arunima Singh, Ph.D.
Senior Editor
Nature Methods

Reviewers' Comments:

Reviewer #1:

Remarks to the Author:

The nature and the message of the paper is a very much split. On the one hand it describes a technical extension of a single cell proteomics method, termed prioritized Single Cell ProtEomics (pSCoPE) on the other hand it focuses on quantifying proteome polarization in primary mouse macrophages. The work is of high quality and some of the biological findings of interest. But of interest to two different communities; one the technology geeks that now all try to jump on the single cell proteomics trail, wherein the Slavov lab is really a pioneer, or the biologists interested in the cellular diversity of macrophages and how different biological processes and activations make them proliferate and reprogram.

First judging on the technical merits, the authors convincingly show that using inclusion lists for peptides to be analysed increases reproducibility and even coverage in single cell proteomics and also allows to look for specific processes, such as proteolytic cleavage products in single cells. However, this approach is by itself not very new, albeit maybe the first time applied to single cell proteomics analysis. If that is sufficient novelty I leave to the editor.

The references mentioned for using inclusion lists (14-20) seem to be picked a bit weird (especially 16 and 18) and the concept is already much older, for instance see *Mol Cell Proteomics*. 2008 Oct; 7(10): 1952–1962. doi: 10.1074/mcp.M800218-MCP200 (14-20). The fact that here it is combined with MaxQuant.Live is also not novel as already cited by them in references 22 and 23. It would be fairer if they consider their reported advances more in the context of these earlier approaches, as then it would be a more technical methods paper.

But the reported single cell features are exciting and shed light on the fact that there is more diversity in macrophages than earlier anticipated, and the prioritized Single Cell ProtEomics (pSCoPE) method used can be used to monitor adaptations to treatment and specific features such as proteolysis. In that sense the paper reminded me on the earlier work of the Picotti lab, *Nat Methods*. 2014 Oct;11(10):1045-8. doi: 10.1038/nmeth.3101. entitled "A sentinel protein assay for simultaneously quantifying cellular processes", wherein they analyzed in parallel hundreds of biological processes by a targeted proteomics approach. Compared to that the current data, for a biological perspective, seem to be more a proof-of-concept.

Where does this leave this paper. It is clearly state-of-the-art, it is exciting and the results are trustworthy and solid. The quality of the data and robustness of the methods well described.

Still, I do find it a paper that both on technical merits and biological findings is more incremental than really ground-breaking.

Reviewer #2:

Remarks to the Author:

The authors present a new strategy to improve the efficiency of single-cell proteomics using prioritized mass spectrometric methods (pSCoPE). The authors sought to address three current challenges in the field: abundance bias, stochastic DDA ion selection, and inefficient peptide sequencing. Real-time prioritization of a target list of peptides could alleviate some or all of these challenges and was tested in the context of primary macrophages. Particularly, the authors highlight how information that would be transparent to sequencing methods (i.e., proteolytic cleavage) can be ascertained using SCP with pSCoPE. They then report a 103% improvement in unique peptides per cell with nice PCA discrimination of signaling axes in primary macrophages. While the development of new single cell approaches is still essential and exciting, and the use of real-time methods with MaxQuant.Live is impressive, the novelty of this work seems to be primarily in the application of methods that have been described previously. Based on that and the comments below I believe this work would need to be revised prior to potential publication.

Major Points

- 1) Put bluntly, is this a macrophage method paper or a single cell proteomics method paper?
 - a) From the title and discussion, my assumption was the latter with an application for macrophages. From the introduction, it appears to be a macrophage methods paper. Highlighting work in the main text such as that in Figure S1 would be helpful in defining the benefits of the more broadly appealing single cell route.
- 2) Prioritization of peptides using real-time methods goes back to at least 2014 with the “IDA” strategies from the Coon lab (Bailey et al 2014) and has already been highlighted as a tool within MaxQuant.Live. The Coon lab work, in particular, details how intelligent, retention time-based, selection of precursors reduces missing values across replicates in much the same fashion as the work here.
 - a) Can the authors speak to novelty compared to this previous work perhaps including (but not solely based on) the modern use of Thermo’s API?
- 3) How was prioritization determined for these analyses? Step one of the process is simply described as “Completion of proteins of interest...”. How was this done, what was used, why these proteins/peptides? Are these user defined features are they intensity defined, a combination?

4) Further, it seems that no recent method that similarly aims to address the challenges set out here were tested at all. The only comparison is to a high-resolution MS2 analysis. For example, while peptide inclusion lists are not possible in RTS from Thermo, protein inclusion lists are possible, and the authors already have made use of an Orbitrap Eclipse.

a) Therefore, are peptides list really essential to address the papers three challenges? It seems that either protein inclusion lists with RTS or the author's isobaric matching between runs methods could generate similar results, but neither is tested but one/both are needed as a method (MBR) and real-time (RTS or RETICLE) controls.

5) The authors set up three main challenges in their work: "abundance bias", missing values, and wasted instrument time. pSCoPE addresses two of these, but even after multiple mentions the abundance bias problem is not explored in the text or figures.

6) Method development and optimization could use some additional clarifying details and descriptions. Sticking with either "Top Tier" or "High priority" would be helpful for message clarity.

a) Figure 1a: What is the increased red AUC for low vs. high priority? It seems that this space would be better used to highlight the actual prioritization process from the supporting information rather than detailing differences with a TopN method.

b) Figure 1b/c: There are ostensibly greater than 4500 peptides that should be included in the heatmap in Figure 1c, based on the boxplot in Figure 1b, though only the Top Tier 857 peptides are shown. Why is this?

c) Figure 1d: What does data completeness on the protein level look like for the Top Tier?

d) Providing a summary on what the 'Top Tier' proteins are that are used in these analyses and what the goal for assigning this is needed.

7) Review of the SCoPE2 protocol paper suggests that the number of peptides per run is markedly lower in this work (<2500 peptides per run) than in previous work from these authors (3000-5000 peptides per run, Petelski et al 2021) and the number of proteins quantified with pSCoPE is on par (~1000 proteins per single cell) with the SCoPE2 method. This would suggest that pSCoPE's primary advance is the data complete work, rather than sensitivity for detection. Along with this, it seems that in Figure 4c, the number of missing values for Acting L104-F1 is greater than the ~60% in Figure 1D.

8) Do the authors observe an effect of high differential abundance on data completeness? If not, could this be in part due to taking long fill times with MS2 quantification which can result in considerable quantitative interference for TMT/pro analyses?

Minor Points

I. A table for the various method parameters should be added instead of the iterative text describing single changes in the supporting information.

II. Figure S2. If the X-axis is CV, please label it as such rather than "Quantification Variability".

III. Is Figure S8 the same as Figure 3?

Reviewer #3:

Remarks to the Author:

This paper focuses on the targeting of subset of peptide/protein in a single cell proteomics using TMT labeling experiments by altering duty-cycle of the mass spectrometer to ensure sufficient data points across each peak for key peptides (required for better quantification) while ensure/balancing proteome depth. This is a smart concept where one focuses on a specific peptide precursor but in a broader manner (verse typically smaller number of peptides selected for MRM/PRM experiments) and an approach, they call pSCoPE, could help to reduce missingness and maybe increase sensitivity. The biological questions are very interesting (and appropriate) which is to define the continuum of polarized states in primary macrophages using single-cell MS-based proteomics (and determine underlying protein drivers across this continuum).

1. The data provide shows that pSCoPE is able to reduce missingness, but the impact of this work would be greatly enhanced if the authors also showed that this approach was quantitative. The selection of the peptides to best quantitatively represent the protein of interest is a primary challenge in MRM/PRM experiments and it is a challenge with this hybrid approach outlined in this paper. The authors need to provide information around the quantitative nature of the peptides selected and in particular, the correlation of the selected prioritized peptide precursors with respect to i) linearity of response, ii) reproducibility and iii) correlation with other peptides that could be used to measure. This may need to be done in an artificial SCP experiment and then validated on a subset in their individual single cell data set.
 2. One issue is that as SCP pushes for protein numbers even as the peptide coverage is reduced. In typical MRM/PRM experiments the expectation is two peptides per protein but note, only one peptide is used for quantification (the second peptide is the qualifier). With this new method, can approach be invoked to assist in more resilient quantification. This may be considered outside the scope of the paper but some measure of quantifiability improvement is required.
 3. The authors need to provide the extent that the increase in the number of quantified peptide (meaning those peptides with sufficient data points across their peak) is impacting quantitation at the protein level. Therefore, the authors need to provide the number of quantifiable peptides verse nonquantifiable but identified peptides using shotgun compared to their new method. Furthermore, whether this increase in quantifiable peptide is dependent on the protein or peptide quantity (sensitivity issue) and whether they find higher reproducibly in the quantifiable peptides verse nonquantifiable peptide.
- The authors should also provide the number of quantifiable peptide/protein in the individual macrophage data set as well.
4. The biological insight provided by applying pSCoPE is important (and well investigated in the results) but is not sufficiently discussed in the conclusion. Specifically, the authors must discuss i) the main biological insights provide by this study on macrophage biology, ii) how this study differs from the many

other studies carried out to date and iii) what was the added value did the application of pSCoPE have for this investigation. These insights should also be highlighted in the abstract. In other words, why should macrophage biologist be interested in this study and how has it change their outlook on their field. This may provide value (for SCP) to the broader community. Not much is needed to be added, but a little bit of needed.

Author Rebuttal to Initial comments

Reviewer #1**Remarks to the Author:**

The nature and the message of the paper is a very much split. On the one hand it describes a technical extension of a single cell proteomics method, termed prioritized Single Cell ProtEomics (pSCoPE) on the other hand it focuses on quantifying proteome polarization in primary mouse macrophages. The work is of high quality and some of the biological findings of interest. But of interest to two different communities; one the technology geeks that now all try to jump on the single cell proteomics trail, wherein the Slavov lab is really a pioneer, or the biologists interested in the cellular diversity of macrophages and how different biological processes and activations make them proliferate and reprogram. First judging on the technical merits, the authors convincingly show that using inclusion lists for peptides to be analysed increases reproducibility and even coverage in single cell proteomics and also allows to look for specific processes, such as proteolytic cleavage products in single cells. However, this approach is by itself not very new, albeit maybe the first time applied to single cell proteomics analysis. If that is sufficient novelty I leave to the editor. The references mentioned for using inclusion lists (14-20) seem to be picked a bit weird (especially 16 and 18) and the concept is already much older, for instance see Mol Cell Proteomics. 2008 Oct; 7(10): 1952–1962. doi: 10.1074/mcp.M800218-MCP200 (14-20). The fact that here it is combined with MaxQuant.Live is also not novel as already cited by them in references 22 and 23. It would be fairer if they consider their reported advances more in the context of these earlier approaches, as then it would be a more technical methods paper. But the reported single cell features are exciting and shed light on the fact that there is more diversity in macrophages than earlier anticipated, and the prioritized Single Cell ProtEomics (pSCoPE) method used can be used to monitor adaptations to treatment and specific features such as proteolysis. In that sense the paper reminded me on the earlier work of the Picotti lab, Nat Methods. 2014 Oct;11(10):1045-8. doi: 10.1038/nmeth.3101. entitled “A sentinel protein assay for simultaneously quantifying cellular processes”, wherein they analyzed in parallel hundreds of biological processes by a targeted proteomics approach. Compared to that the current data, for a biological perspective, seem to be more a proof-of-concept. Where does this leave this paper. It is clearly state-of-the-art, it is exciting and the results are trustworthy and solid. The quality of the data and robustness of the methods well described. Still, I do find it a paper that both on technical merits and biological findings is more incremental than really ground-breaking.

Major Points:

1. The references mentioned for using inclusion lists (14-20) seem to be picked a bit weird (especially 16 and 18) and the concept is already much older, for instance see Mol Cell Proteomics. 2008 Oct; 7(10): 1952–1962. doi: 10.1074/mcp.M800218-MCP200 (14-20).

Thank you for suggesting these references. We included all of them in the revised version of our manuscript and removed previous ref 18. We also added explicit discussion to the important results demonstrated by these prior papers.

2. The fact that here it is combined with MaxQuant.Live is also not novel as already cited by them in references 22 and 23. It would be fairer if they consider their reported advances more in the context of these earlier approaches, as then it would be a more technical methods paper.

We agree that such comparisons are essential for demonstrating the novelty and advantages of the prioritization logic that we introduced, and we performed well controlled experiments to generate a new data set explicitly comparing the performance of MaxQuant.live with and without prioritization while keeping all other parameters the same. An excerpt of the associated main figure, extended data figure, and textual reference to the comparison are displayed below:

Figure R1 | Priority tiers increase consistency of identification while increasing protein coverage per sample. (b) Prioritized analysis increases the consistency of peptide identification over default MaxQuant.Live operation for High Priority peptides, while also increasing protein coverage per run. (c) MS1 detection and MS2 analysis rates for prioritized precursors in the benchmarking experiments displayed in panel b.

"To benchmark the benefits of prioritization, MaxQuant.Live was used to acquire data with and without prioritization enabled while keeping all other parameters constant, Fig. R1b. To reduce sample related variability, we analyzed injections from a bulk sample diluted to single-cell levels. The inclusion list was composed of the the same precursors for the prioritized and non-prioritized analysis by MQ.Live: over 11,500 precursors selected to be identifiable, along with a comparable number of precursors used only for retention-time calibration. The precursors on the inclusion list were then stratified into three levels of priority by the confidence of their identification and spectral purity in previous analysis. More confidently-identified and less coisolated peptides were assigned to the higher priority levels. Data completeness for the high-priority group of 4,000 peptides increased to 72% when using prioritization, compared to 49% without prioritization. The fraction of peptides identified in 100% of the 6 runs at 1% FDR was 18% without prioritization and 59% with prioritization (Fig. R1b), representing a 228% increased consistency for prioritization. This increased consistency of identification did not impede protein coverage, as prioritization increased the number of quantified proteins per experiment at 1% FDR, Fig. R1a. Consistent with the prioritization logic shown in Fig. R1a, prioritization sent precursors to MS2 scans according to priority: 97% of the 4000 high-priority peptides were sent for MS2 analysis, and lower fractions of the lower priority tiers, Fig. R1c. In contrast, MaxQuant.Live without prioritization sent similar fractions (about 63%) of peptides from all lists for MS2 analysis as shown in Extended Data Fig. R2b."

Figure R2 | Percent of inclusion-list precursors detected and analyzed in platform benchmark runs, for MaxQuant.Live with and without prioritization enabled

(a) MS1 detection rates for precursors in the platform benchmark experiments displayed in main Fig. 1b,c. Data collected using MaxQuant.Live in default mode are shown in black, while data collected using prioritization are shown in red. The precursor count displayed at the bottom of each priority level's facet corresponds to the number of precursors present on the priority level of the inclusion list. (b) MS2 analysis rates for precursors in the platform benchmark experiments displayed in main Fig. 1b,c. While the MS1 precursor detection rates are similar for both platforms, the MS2 analysis rates are correlated to the priority levels for prioritized analysis, but not for default MaxQuant.Live analyses. Each boxplot shown above contains 6 data points, one for each LC-MS/MS analysis.

Reviewer #2:**Remarks to the Author:**

The authors present a new strategy to improve the efficiency of single-cell proteomics using prioritized mass spectrometric methods (pSCoPE). The authors sought to address three current challenges in the field: abundance bias, stochastic DDA ion selection, and inefficient peptide sequencing. Real-time prioritization of a target list of peptides could alleviate some or all of these challenges and was tested in the context of primary macrophages. Particularly, the authors highlight how information that would be transparent to sequencing methods (i.e., proteolytic cleavage) can be ascertained using SCP with pSCoPE. They then report a 103% improvement in unique peptides per cell with nice PCA discrimination of signaling axes in primary macrophages. While the development of new single cell approaches is still essential and exciting, and the use of real-time methods with MaxQuant.Live is impressive, the novelty of this work seems to be primarily in the application of methods that have been described previously. Based on that and the comments below I believe this work would need to be revised prior to potential publication.

Major Points

1. Put bluntly, is this a macrophage method paper or a single cell proteomics method paper?

- a. From the title and discussion, my assumption was the latter with an application for macrophages. From the introduction, it appears to be a macrophage methods paper. Highlighting work in the main text such as that in Figure S1 would be helpful in defining the benefits of the more broadly appealing single cell route.

We think our paper reports both novel methodology for single-cell proteomics and new biological observations regarding macrophage polarization. Thanks to comments by you and the other reviewers, we appreciated that the first version of our article had left important methodological aspects underdeveloped, and we added them to this revision so that the methodological novelty is not adversely affected by the second part of the paper reporting new macrophage biology. We think the biological findings provide useful context and evidence for the utility of the methodology that we introduce. Specifically, we added two additional methodological main figures (Fig. 1 and Fig. 3 in the revised manuscript). Based on your suggestion we added a revised Fig. 1, which includes the results from the previous figure S1 (panel c). The new Fig. 1 is excerpted above as Fig. R2 in our response to Reviewer 1's second question, which includes data from a new experiment (panel b) and a refined version of the previous analysis of precursors identified at MS1 and isolated for MS2 (panel c).

2. Prioritization of peptides using real-time methods goes back to at least 2014 with the "IDA" strategies from the Coon lab (Bailey et al 2014) and has already been highlighted as a tool within MaxQuant.Live. The Coon lab work, in particular, details how intelligent, retention time-based, selection of precursors reduces missing values across replicates in much the same fashion as the work here.

- a. Can the authors speak to novelty compared to this previous work perhaps including (but not solely based on) the modern use of Thermo's API?

Thank you for pointing out that we had insufficiently compared and benchmarked pSCoPE to other real-time methods, which indeed have a long history. In the revised paper, we referenced and discussed all real-time methods suggested by you and the other reviewers, especially the pioneering work by Bailey *et al.*. Furthermore, we added a well-controlled comparison between the performance of MaxQuant.Live with and without prioritization enabled while keeping all other parameters

constant, Fig. R1. We also added explicit discussion of the advantages of our approach, both to the introduction and the discussion sections. Below is an excerpt from the discussion section and an elaboration on the comparison between pSCoPE and IDA.

Some directed methods^{1,2} can be used in hybrid mode with a small inclusion list and shotgun analysis (scan while idle). This hybrid mode can be seen as a single level of priority, but it suffers from the low identification rate for MS2 scans acquired in shotgun mode. The multiple levels of priority introduced here allow for 84% sequence assignment rate for all MS2 spectra (not only the high priority ones) taken at full duty cycle, Fig. 2a.

While IDA enables a top-N DDA method to run when there are no precursors from the inclusion list available, pSCoPE only sends for analysis precursors that have previously been found to be confidently identifiable from targeted screening runs. This is an especially important aspect for the analysis of low-input and single-cell samples, as such analyses require long MS2 fill times (≥ 250 ms) on a QE instrument to generate confident identifications and accurate reporter-ion quantitation. Given these long MS2 fill times, maximizing the number of MS2 scans that generate confident identifications becomes even more important than in non-single-cell analyses, as there are fewer MS2s taken. Shotgun analyses of multiplexed single-cell samples often generate confident identifications for 40% of all MS2 scans taken; however, when MS2 analysis is restricted to precursor species that have previously resulted in identifications with high spectral confidence, this percentage climbs to 84%, effectively doubling instrument productivity, Fig. R3. This approach also more-than-doubles the number of unique modified peptide sequences identified per run (when multiple charge states are condensed to a single representative modified sequence), and increases the number of proteins quantified per single cell by 106%.

Figure R3 | Prioritized precursor selection enables more productive instrument usage and greater coverage on a peptide and protein basis (a) Relative to shotgun analysis, prioritization increased the fraction of MS2 scans assigned to peptide sequences, the number of peptides per run (65-min active gradient), and the number of quantified proteins per single cell.

15

We also referenced the previous foundational work in the introduction, excerpted below:

Peptide selection by the topN heuristic is limited by three challenges: (i) abundance bias, which limits the dynamic range of quantified proteins; (ii) stochasticity, which limits data completeness across single cells; and (iii) unidentifiable precursors, whose analysis wastes instrument time and limits proteome coverage³. Such inefficient use of time is particularly limiting for single-cell proteomics due to the long ion accumulation times needed to sequence and quantify each precursor^{4,5}. While no existing method resolves all 3 challenges, the challenges can be partially mitigated. For example, targeted MS can alleviate challenges (i) and (ii) but has remained limited to analyzing hundreds of peptides or fewer⁶⁻¹³. Real-time database searching can increase the fraction of sequenced peptide features and alleviate challenge (iii), but it has not allowed for selecting peptides of interest^{4,15}. Targeting peptides from inclusion lists with real-time retention-time alignment ameliorates challenge (i) but faces a trade-off between maximizing coverage (and thus duty cycle usage) and maximizing data completeness^{1,2}. To simultaneously address all three challenges, we introduce multitiered prioritized selection of peptide precursors using the strategy shown in Fig. 2a.

3. How was prioritization determined for these analyses? Step one of the process is simply described as “Compilation of proteins of interest...”. How was this done, what was used, why these proteins/peptides? Are these user defined features are they intensity defined, a combination?

For the experiments reported in Figures 1, 2 and 3, we prioritized peptides that were confidently identified in previous runs and had low level of co-isolation. For Figures 4, 5, and 6 we prioritized peptides that exhibited large variability within primary macrophages (thus likely to reflect biological changes) or are directly related to the expected biological responses. We added these brief explanations to the main text and very detailed descriptions to the Methods section. For each prioritized experiment, we have dedicated areas in the Methods section detailing the selection of precursors for each priority tier.

4. Further, it seems that no recent method that similarly aims to address the challenges set out here were tested at all. The only comparison is to a high-resolution MS2 analysis. For example, while peptide inclusion lists are not possible in RTS from Thermo, protein inclusion lists are possible, and the authors already have made use of an Orbitrap Eclipse.

a. Therefore, are peptide lists really essential to address the papers three challenges? It seems that either protein inclusion lists with RTS or the author’s isobaric matching between runs methods could generate similar results, but neither is tested but one/both are needed as a method (MBR) and real-time (RTS or RETICLE) controls.

This is an excellent point. Based on the reviewer’s suggestion, we have compared the performance of prioritization to shotgun analyses searched using MaxQuant’s isobaric match-between-runs feature, and the results are discussed in the main paper (excerpt below), while the corresponding figure is present in the extended figures section.

Figure R4 | pSCoPE outperforms isobaric Match Between Runs (IMBR) for increasing consistency of identification across single-cell experiments

(a) The "All Precursors" facet heading indicates the total number of MBR-facilitated precursor identifications in each of 8 shotgun analyses. The "Precursors with MS2 Scans" facet heading indicates the total number of MBR-facilitated precursor identifications that are associated with MS2 scans, enabling reporter ion quantitation. In both facets, the identifications are segmented into "All matches", a category which includes matches to reverse sequences, and "Forward matches", which does not. Each point represents an experiment. Data derived from shotgun experiments shown in publication Fig. 2a,b,c,d,e (b) The intersected precursors between the MBR-facilitated forward sequence matches and the corresponding prioritized analyses were then compared based on consistency of identification across the 8 experiments associated with each acquisition method. Each point represents a precursor. Data derived from shotgun and pSCoPE analyses shown in publication Fig. 2a

Other methods for increasing data completeness post-acquisition, such as isobaric match-between-runs (IMBR), facilitated the identification and quantification of approximately 170 precursors per experiment whose spectra alone were unable to support a PSM, Fig. R4; the decrease in missing data afforded by such approaches, however, cannot be selectively applied to experimentally interesting groups of peptides and requires an unidentified MSMS to be effective.

5. The authors set up three main challenges in their work: "abundance bias", missing values, and wasted instrument time. pSCoPE addresses two of these, but even after multiple mentions the abundance bias problem is not explored in the text or figures.

Thank you for suggesting to address this important point, which was missing from our first submission. We have since added a panel to Fig. 2 demonstrating the shift in precursor abundances relative to shotgun for all peptides identified at 1% FDR and addressed this claim in the accompanying text, both excerpted below. In the excerpted figure, each acquisition method was applied to 8 SCoPE sets, and the median number of peptides identified at 1% FDR was calculated for each of 30 precursor intensity bins. The standard deviation is shown as an error bar around the median.

"pSCoPE also directly facilitated the successful analysis of lower abundance precursors, shifting the median precursor intensity of peptides identified at 1% FDR by over 2.5-fold lower, relative to shotgun"

6. Method development and optimization could use some additional clarifying details and descriptions. Sticking with either "Top Tier" or "High priority" would be helpful for message clarity.

Thank you for this recommendation. We have changed all mentions to "High priority".

a. Figure 1a: What is the increased red AUC for low vs. high priority? It seems that this space would be better used to highlight the actual prioritization process from the supporting information rather than detailing differences with a TopN method.

The increased area under the curves in the original figure was an illustration of increased fill times for high-priority peptides, but we have since reworked the figure based on your suggestions. The excerpted figure, below, offers a better illustration of the prioritized precursor selection logic:

Figure R5 | Prioritization schematic: Shotgun TopN analysis selects the *N* most abundant precursors for isolation and fragmentation (shown in blue). Among the many detected precursors, prioritized analysis first selects the ones with highest priority (shown in solid red) and then from lower priority tiers (shown with decreasingly saturated red tones). Prioritization can also selectively allocate increased fill times to High Priority peptides of low abundance, as shown in the second cycle of MS2 scans.

b. Figure 1b/c: There are ostensibly greater than 4500 peptides that should be included in the heatmap in Figure 1c, based on the boxplot in Figure 1b, though only the Top Tier 857 peptides are shown. Why is this?

The data shown in the original version of **Fig 1c** and **Fig. 1b** were derived from separate experimental sets: the first aimed at demonstrating improved depth of coverage, the second aimed at demonstrating improved consistency of identification for a set of difficult-to-identify peptides. We have since generated new experimental sets that demonstrated improved performance, and have presented the data in a revised figure, with sub-panels a., c., and d. excerpted below:

Figure R6 | Benchmarking prioritization against shotgun

Although 13,630 precursors were prioritized as a part of the experiments shown in **Fig. 2c**, we have shown the data for the 1,000 peptides on the high priority tier to highlight the performance of the method for difficult-to-identify species. The performance for peptides and proteins across all priority levels is presented in **Fig. 2d**. As in the previous version, the performance of prioritization with respect to productive MS2 scans, peptides identified per run, and proteins quantified per cell as shown in **Fig. 2a** is derived from a separate group of experiments meant to demonstrate the method's performance for maximizing depth of coverage.

These aims of maximizing coverage and consistency are not mutually exclusive, though, and the performance of the consistency experiments with respect to the same metrics shown in **Fig. 2a**, above, are quite comparable.

c. Figure 1d: What does data completeness on the protein level look like for the Top Tier?

We did not report protein completeness per tier because a protein may have peptides distributed across different tiers. In the new Fig. 2d (corresponding to old Fig. 1d), the protein data completeness for all proteins is 94%, and even higher for proteins having peptides in the top tier.

d. Providing a summary on what the 'Top Tier' proteins are that are used in these analyses and what the goal for assigning this is needed.

Thank you for this suggestion. We have included such summaries, including detailed descriptions of the inclusion list assembly logic in the Methods section.

7. Review of the SCoPE2 protocol paper suggests that the number of peptides per run is markedly lower in this work (<2500 peptides per run) than in previous work from these authors (3000-5000 peptides per run, Petelski et al 2021) and the number of proteins quantified with pSCoPE is on par (1000 proteins per single cell) with the SCoPE2 method. This would suggest that pSCoPE's primary advance is the data complete work, rather than sensitivity for detection.

Thank you for pointing to the need to also compare our results to previously published ones. The discrepancy in this case arises from comparing the number of PSMs before or after DART-ID¹⁶ update. Petelski, *et al.* reported between 2,000 – 2,500 confident PSMs from a MaxQuant search, and this number increased to about 3,000 and above after DART-ID incorporated retention time estimates as reproduced in Fig. R7. Our current paper reports in Figures 1 - 3 PSMs without DART-ID update. Petelski *et al.*, 2021 also reports the number of confident PSMs without DART-ID update (green bars in Fig. R7) and these are below 2,500 precursors per run. Indeed, after analyzing the data for figure 3 from the Nature Protocol and subjecting the PSMs to the same data filtration (e.g, collapsing charge states to unique peptide sequences) and FDR calculation as performed in pSCoPE, we find that a median value of 2,082 peptides were identified per run at 1% FDR in the SCoPE2 Protocol, which actually under-performs relative to the shotgun analyses of the HEK and Melanoma samples from Figure 2a of the current pSCoPE manuscript, Fig. R8. The prioritized analysis outperforms both shotgun methods by a large margin.

Figure R7 | Figure excerpt from Petelski, et al., 2021

Figure R8 | Sensitivity contrast

Huffman et al. manuscript: Shotgun	Huffman et al. manuscript: Prioritized	Petelski et al., 2021: Shotgun
2790	5549	2081.5

Table 1 | Median number of peptides identified per experiment.

8. Along with this, it seems that in Figure 4c, the number of missing values for Actin L104-F1 is greater than the 60% in Figure 1D.

Regarding the data completeness of Actin L104-F1, the reviewer is absolutely correct: this peptide was only quantified in 24% of the single-cell samples (about 90 primary macrophage cells). The BMDM study's performance, shown in extended data figure 4, is actually closer to a median value of 75% data completeness for high-priority peptides, although not all peptides in the high-priority group will have this level of data completeness. The data completeness for Actin L104-F1 is lower as it is lowly abundant, and it was not confidently identified in all pSCOPE sets.

21

9. Do the authors observe an effect of high differential abundance on data completeness? If not, could this be in part due to taking long fill times with MS2 quantification which can result in considerable quantitative

interference for TMT/pro analyses?

Yes, we observe that lowly abundant peptides and proteins have more missing data, even for confidently identified peptides due to their reporter ion intensities being below the limit of detection. This is quite clearly visible with the lower number of data points for the lowest level of spike-in peptides shown in the current version of Fig. 3, shown below as Fig. R10. As expected by the reviewer, we also see more missing data for variable proteins in the cell states in which these proteins are lowly abundant.

Minor Points

I. A table for the various method parameters should be added instead of the iterative text describing single changes in the supporting information.

We have provided such tables, which made the Method section much more readable. Thank you for this suggestion.

II. Figure S2. If the X-axis is CV, please label it as such rather than "Quantification Variability".

Thank you for this suggestion. We have replaced the X-axis titles for this figure, excerpted below:

III. Is Figure S8 the same as Figure 3?

The previous Figure S8 (current Extended Data Figure 10) contains data solely corresponding to the untreated BMDMs, whereas the previous Figure 3b/c (current Figure 5b/c) presents data solely from the LPS-treated BMDMs. We emphasized this in the revised paper.

Figure R9 | Single-cell quality controls The median coefficient of variation (i.e. the standard deviation scaled by the mean) of all peptide-level relative abundances that map to the same leading razor protein is used to separate successfully prepared single cells from those that will not generate accurate data. By choosing a CV threshold that separates control samples (droplets which received all reagents but did not contain a single cell) from single cells, cells with noisier protein-level quantitation can be removed prior to further data processing. The single cell and control tallies above each figure represent the number of single cells or control wells that passed the CV threshold of 0.4. (a) contains the CV distributions for the single-cell samples associated with Fig. 2a,b,c,d, analyzed by shotgun LC-MS/MS methods. (b) contains the CV distributions for the single-cell samples associated with Fig. 2a, analyzed by pSCoPE. (c) contains the CV distributions for the single-cell samples associated with Fig. 4, Fig. 5, and Fig. 6. (d) contains the CV distributions for the single-cell samples associated with Fig. 4, Fig. 5, and Fig. 6.

Reviewer # 3:**Remarks to the Author:**

This paper focuses on the targeting of subset of peptide/protein in a single cell proteomics using TMT labeling experiments by altering duty-cycle of the mass spectrometer to ensure sufficient data points across each peak for key peptides (required for better quantification) while ensure/balancing proteome depth. This is a smart concept where one focuses on a specific peptide precursor but in a broader manner (verse typically smaller number of peptides selected for MRM/PRM experiments) and an approach, they call pSCoPE, could help to reduce missingness and maybe increase sensitivity. The biological questions are very interesting (and appropriate) which is to define the continuum of polarized states in primary macrophages using single-cell MS-based proteomics (and determine underlying protein drivers across this continuum).

Major Points:

1. The data provided shows that pSCoPE is able to reduce missingness, but the impact of this work would be greatly enhanced if the authors also showed that this approach was quantitative. The selection of the peptides to best quantitatively represent the protein of interest is a primary challenge in MRM/PRM experiments and it is a challenge with this hybrid approach outlined in this paper. The authors need to provide information around the quantitative nature of the peptides selected and in particular, the correlation of the selected prioritized peptide precursors with respect to:

- i) linearity of response
- ii) reproducibility
- iii) correlation with other peptides that could be used to measure.

This may need to be done in an artificial SCP experiment and then validated on a subset in their individual single cell data set.

These are excellent suggestions. In order to benchmark linearity of response, reproducibility, and correlation with other peptides, we used spike-in peptides added to the single-cell and control samples of TMT sets. Each peptide ranged in abundance over a 16-fold dynamic range across 5 spiked-in levels. The measured abundances exhibited linear dependence with the spiked-in levels with a slope of 1.06 and a coefficient of determination $R^2 = 0.97$, as shown in new Fig. 3b and reproduced below as Fig. R10. These results indicate that pSCoPE is able to quantify peptides in single-cell sets with good accuracy and precision.

2. One issue is that as SCP pushes for protein numbers even as the peptide coverage is reduced. In typical MRM/PRM experiments the expectation is two peptides per protein but note, only one peptide is used for quantification (the second peptide is the qualifier). With this new method, can the approach be invoked to assist in more resilient quantification? This may be considered outside the scope of the paper but some measure of quantifiability improvement is required.

Prioritization can be easily used to maximize the number of quantified peptides per protein. We did not directly aim for this in our workflow; rather if we had more than 4 peptides per protein, we placed these excess peptides on lower priority tiers, as explained in the Methods. Thus we had multiple peptides for many proteins (Fig. R11) though not for all, and a different inclusion list construction could have easily increased the number of peptides / protein at the expense of lower proteome coverage. In the data associated with Fig. 2a of our revised article, there were 262 proteins in common between the prioritized and shotgun analyses which were only supported by a single peptide in the shotgun analyses, and whose peptides/protein count was increased to ≥ 2 in the pSCoPE analyses, shown in rebuttal associated Fig. R11 below. The median

Figure R10 | Evaluating quantitative accuracy and precision of pSCoPE with peptide standards. (a) Reporter ion intensities for precursors identified at 1% FDR from single cells and from the spike-in peptides, which were dispensed into the single-cell samples across a 16-fold range. (b) Normalized reporter ion intensities for all tryptic products from the spike-in peptides plotted against their spike-in amounts, with regression slope and goodness of fit displayed. The data in both panels comes from 8 prioritized experiments.

number of peptides per protein for this set of proteins was 3 for the pSCoPE analyses. For the proteins in this set, the single-cell peptide abundances for each peptide were correlated to one another on a per-protein basis, and the median correlation for each protein's peptides is presented in the bottom panel of Fig. R11. The median Pearson correlation for these proteins is 0.44.

Figure R11 | The top panel shows the number of peptides per protein in the pSCoPE data for proteins quantified by a single peptide in the corresponding shotgun analyses. The bottom panel shows the correlations between different peptides originating from the same protein. The vertical line indicates the median, $r = 0.44$. The low correlations correspond to proteins with low fold changes across the single cells, and thus low signal.

3. The authors need to provide the extent that the increase in the number of quantified peptide (meaning those peptides with sufficient data points across their peak) is impacting quantitation at the protein level. Therefore, the authors need to provide the number of quantifiable peptides versus nonquantifiable but identified peptides using shotgun compared to their new method. Furthermore, whether this increase in quantifiable peptide is dependent on the protein or peptide quantity (sensitivity issue) and whether they find higher reproducibility in the quantifiable peptides versus nonquantifiable peptide. The authors should also provide the number of quantifiable peptide/protein in the individual macrophage data set as well.

Our implementation of prioritization did not aim to sample precursors at multiple points across their elution peaks via repeat MS2 analyses. Rather, we used precise retention-time alignment to sample precursors at the apex of their elution profile (as seen from the DO-MS reports), thus achieving good quantitative accuracy as shown in main Fig. 3 (reproduced here as Fig. R10), and maximizing the number of analyzed peptides. If desired, one may use our prioritized approach to sample precursors at multiple time points though this will come at the cost of proportional decrease in the number of analyzed precursors.

4. The biological insight provided by applying pSCoPE is important (and well investigated in the results) but is not sufficiently discussed in the conclusion. Specifically, the authors must discuss:

- i) the main biological insights provide by this study on macrophage biology
- ii) how this study differs from the many other studies carried out to date
- iii) what was the added value did the application of pSCoPE have for this investigation.

These insights should also be highlighted in the abstract. In other words, why should macrophage biologist be interested in this study and how has it change their outlook on their field. This may provide value (for SCP) to the broader community. Not much is needed to be added, but a little bit is needed.

Thank you for this great suggestion. We aimed to add these clarifications and discussion points as appropriate. Below is an excerpt of a paragraph that we added to the Discussion:

We measured protein covariation of functionally related proteins within primary macrophages not only between treatment groups but also within a treatment group, as shown Fig. 4a for phagosome maturation proteins. The proteins exhibiting such within-condition variability are similar for the treated and untreated macrophages, Fig. 5a. This similarity in protein covariation is remarkable because LPS treatment substantially remodels the proteome, and yet protein covariation remains similar for treated and untreated macrophages. A possible explanation for this finding is that protein covariation reflects the topology of regulatory interactions¹⁷, and many of these regulatory interactions remain similar between the untreated and LPS-treated macrophages. This interpretation is consistent with the observation that the proteins associated with dextran uptake are similar for the two conditions as shown in Fig. 5b and Extended Data Fig. 10. Additionally, prioritized analysis enabled the quantification of a proteolytically modified cytoskeletal protein whose cleavage is significantly correlated to inflammatory stimulus across single-cell samples. The robustness of the results to choices of data analysis, such as different treatments of missing data (Extended Data Fig. 7), bolsters their validity¹⁸.

References

1. Bailey, D. J., McDevitt, M. T., Westphall, M. S., Pagliarini, D. J. & Coon, J. J. Intelligent Data Acquisition Blends Targeted and Discovery Methods. *Journal of Proteome Research* **13**. Publisher: American Chemical Society, 2152–2161. issn: 1535-3893. <https://doi.org/10.1021/pr401278j> (2022) (Apr. 2014).
2. Wichmann, C. *et al.* MaxQuant.Live enables global targeting of more than 25,000 peptides. *Molecular & Cellular Proteomics* **18**, 982–994 (2019).
3. Slavov, N. Driving Single Cell Proteomics Forward with Innovation. *Journal of Proteome Research* **20**, 4915–4918. <https://doi.org/10.1021/acs.jproteome.1c00639> (2021).
4. Specht, H. *et al.* Single-cell proteomic and transcriptomic analysis of macrophage heterogeneity using SCoPE2. *Genome Biology* **22** (2021).
5. Slavov, N. Scaling Up Single-Cell Proteomics. *Molecular & Cellular Proteomics* **21**, 100179. issn: 1535-9476 (2022).
6. Jaffe, J. D. *et al.* Accurate Inclusion Mass Screening: A Bridge from Unbiased Discovery to Targeted Assay Development for Biomarker Verification. English. *Molecular & Cellular Proteomics* **7**. Publisher: Elsevier, 1952–1962. issn: 1535-9476, 1535-9484. [https://www.mcponline.org/article/S1535-9476\(20\)31278-0/abstract](https://www.mcponline.org/article/S1535-9476(20)31278-0/abstract) (2022) (Oct. 2008).
7. Aebersold, R. & Mann, M. Mass spectrometry-based proteomics. *Nature* **422**, 198 (2003).
8. Soste, M. *et al.* A sentinel protein assay for simultaneously quantifying cellular processes. en. *Nature Methods* **11**. Number: 10 Publisher: Nature Publishing Group, 1045–1048. issn: 1548-7105. <https://www.nature.com/articles/nmeth.3101> (2022) (Oct. 2014).
9. Picotti, P. & Aebersold, R. Selected reaction monitoring-based proteomics: workflows, potential, pitfalls and future directions. *Nature methods* **9**, 555 (2012).
10. Picotti, P. *et al.* A complete mass-spectrometric map of the yeast proteome applied to quantitative trait analysis. *Nature* **494**, 266–270 (2013).
11. Peterson, A. C., Russell, J. D., Bailey, D. J., Westphall, M. S. & Coon, J. J. Parallel reaction monitoring for high resolution and high mass accuracy quantitative, targeted proteomics. *Molecular & Cellular Proteomics* **11**, 1475–1488 (2012).
12. Erickson, B. K. *et al.* A strategy to combine sample multiplexing with targeted proteomics assays for high-throughput protein signature characterization. *Molecular cell* **65**, 361–370 (2017).
13. Manes, N. P. & Nita-Lazar, A. Application of targeted mass spectrometry in bottom-up proteomics for systems biology research. *Journal of proteomics* **189**, 75–90 (2018).
14. Schweppe, D. K. *et al.* Full-Featured, Real-Time Database Searching Platform Enables Fast and Accurate Multiplexed Quantitative Proteomics. en. *J. Proteome Res.* **19**, 2026–2034 (May 2020).
15. Furtwängler, B. *et al.* Real-Time Search Assisted Acquisition on a Tribrid Mass Spectrometer Improves Coverage in Multiplexed Single-Cell Proteomics. en. *Mol. Cell. Proteomics* (2022).
16. Chen, A. T., Franks, A. & Slavov, N. DART-ID increases single-cell proteome coverage. *PLOS Computational Biology* **15**, 1–30 (July 2019).
17. Slavov, N. Learning from natural variation across the proteomes of single cells. *PLOS Biology* **20**, 1–4. <https://doi.org/10.1371/journal.pbio.3001512> (Jan. 2022).
18. Gatto, L. *et al.* Initial recommendations for performing, benchmarking, and reporting single-cell proteomics experiments. <https://arxiv.org/abs/2207.10815> (2022).

Decision Letter, first revision:

Dear Nikolai,

Thank you for submitting your revised manuscript "Prioritized single-cell proteomics reveals molecular and functional polarization across primary macrophages" (NMETH-A48778A). It has now been seen by the original referees and their comments are below. The reviewers find that the paper has improved in revision, and therefore we'll be happy in principle to publish it in Nature Methods, pending revisions to satisfy the referees' final requests and to comply with our editorial and formatting guidelines.

TRANSPARENT PEER REVIEW

Nature Methods offers a transparent peer review option for new original research manuscripts submitted from 17th February 2021. We encourage increased transparency in peer review by publishing the reviewer comments, author rebuttal letters and editorial decision letters if the authors agree. Such peer review material is made available as a supplementary peer review file. Please state in the cover letter 'I wish to participate in transparent peer review' if you want to opt in, or 'I do not wish to participate in transparent peer review' if you don't. Failure to state your preference will result in delays in accepting your manuscript for publication.

ORCID

Sincerely,
Arunima

Arunima Singh, Ph.D.
Senior Editor
Nature Methods

Reviewer #1 (Remarks to the Author):

The authors have gone in depth to provide much more technical details about the performed experiments and provided more check and balances of the data, that is very much appreciated. Still many of these technical advances have been reported by others, just not analyzing single cell samples. I keep of the impression that this paper shows only incremental technical advances, and too little biological advances.

Reviewer #2 (Remarks to the Author):

The authors' revised manuscript for the pSCoPE method using MaxQuant.Live goes a long way to addressing my main concerns. Particularly, the new figures detailing the accessibility of low abundance precursors using their method relative to shotgun methods and the improvement over IMBR. The greater focus on the methods development is appreciated and I believe improves the manuscript. While I wish that was that, this clarified scope has generated some additional questions and comments which I include below.

Comments and Questions

- 1) The authors nicely added data addressing the question of dealing with 'abundance bias'. I would like to see a slight addition to the new Figure 2b (perhaps in supplement) that highlighted the same red distribution for the prioritized precursors but for each tier. I believe the audience would like to see how that is broken down.
- 2) While used in several places, the authors' meaning of 'full duty cycle' is not well defined. As in their shotgun analysis with a Top-N method is a 'full duty-cycle' method. Specifying that the interpretation here is that injection times are dynamically adjusted to improve quantitation is an important point that

should be solidified in a sentence or two. To my best estimate, that is one of the most important aspects of this method.

3) For the carrier proteomes, it seems that the authors used several different carrier amounts (50-200 cells).

a) No details are provided about how this might affect prioritization.

b) The authors mention a 50-cell carrier for the U937 vs. HEK293 experiment, was this same channel arrangement used for subsequent analyses (e.g., skipping 127C and 128N channels)? A simple table (such as that found in Table S1) highlighting how macrophage/treated samples were distributed over TMTpro channels would be a very useful guide for future users and should be included.

4) In Figure 6, what correlation metric are the authors using? R, R²? Is this Spearman, Pearson? I assume R and Spearman based on the text, but this should be added to the Figure legend if correct.

a) Why do the authors believe the correlation coefficients (Assuming R) are less than an absolute value of 0.3?

5) Much of the 'raw' quantitative data is obscured by PCA plots.

a) Adding a small cohort of scatter plots for the untreated/treated single cells vs bulk proteome data would be helpful here to see the actual trends that are mentioned or alluded to through the PC and correlation plots.

6) 'Top tier', rather than 'high priority' still appears several times throughout the manuscript.

7) Perhaps a minor point, but re-reading this manuscript several times now, calling the proposed method 'prioritized Single Cell Proteomics' seems to limit acknowledgement of the extensibility of the work. I understand the branding of 'SCoPE' but could these methods not be applied to nearly any low input, or even bulk, sample? Lowercase 'p' in front of proteomics methods often also denote phosphoproteomics variations of methods which may confuse some readers.

8) What is SQC? The acronym is used throughout the methods section but, as best as I can tell, never defined.

9) What is the difference between the GitHub site for pSCoPE (<https://github.com/SlavovLab/pSCoPE>) and the lab specific link?: scp.slavovlab.net/pSCoPE

a) In general, for long term accessibility, it is better to use (and point people to) public data repositories rather than lab specific ones for data sharing.

10) On page 7, this sentence is confusing owing to having additional/adding three times in one sentence: "iMBR facilitated the identification and quantification of approximately 170 additional precursors per shotgun experiment compared to 2,595 additional precursors per experiment added by prioritization, Extended Data Fig. 4."

Reviewer #3 (Remarks to the Author):

The authors have been very responsive and have addressed all of my concerns.

Comment 1. I very much like the spiked in TMT experiment. This data set is super cool and can be used by others to test different data analysis methods and thus, has broader use than just this paper. Thank you.

Comment 2 and 3. The new figures R10 and R11 are useful and address my suggestion.

Comment 4. I am pleased with the additional paragraph that emphasis some biological interesting points.

Author Rebuttal, first revision:

Reviewer #2 (Remarks to the Author):

The authors' revised manuscript for the pSCoPE method using MaxQuant.Live goes a long way to addressing my main concerns. Particularly, the new figures detailing the accessibility of low abundance precursors using their method relative to shotgun methods and the improvement over IMBR. The greater focus on the methods development is appreciated and I believe improves the manuscript. While I wish that was that, this clarified scope has generated some additional questions and comments which I include below.

Comments and Questions

1) The authors nicely added data addressing the question of dealing with 'abundance bias'. I would like to see a slight addition to the new Figure 2b (perhaps in supplement) that highlighted the same red distribution for the prioritized precursors but for each tier. I believe the audience would like to see how that is broken down.

Thank you. We added the distributions of abundances as Supplemental Fig. S2, which is also reproduced below.

2) While used in several places, the authors' meaning of 'full duty cycle' is not well defined. As in their shotgun analysis with a Top-N method is a 'full duty-cycle' method. Specifying that the interpretation here is that injection times are dynamically adjusted to improve quantitation is an important point that should be solidified in a sentence or two. To my best estimate, that is one of the most important aspects of this method.

Thank you for this suggestion. We agree that the full duty cycle is not as well defined for pSCoPE as it is for shotgun. The closest equivalent for pSCoPE is MS2 scans acquired during the full lifetime of the cycle. Thus, we simplified our writing to explain in simpler and concrete terms our approach without referring to full duty cycle. Below is an excerpt of this description:

These efficiencies were achieved while pSCoPE maximized the number of MS2 scans, thus demonstrating the ability of tiers (prioritization) to increase the probability of analyzing peptides while also increasing the total number of precursors sent for MS2 scans.

- 3) For the carrier proteomes, it seems that the authors used several different carrier amounts (50-200 cells).
- a) No details are provided about how this might affect prioritization.

The SQC sample featuring U937 and HEK293 digest used twin carriers, each corresponding to 50 cells of one cell type (126C and 127N); the carrier for the HEK293 and melanoma single-cell analyses featured in Figures 2 and 3 used a single carrier channel (126C) composed of digest equivalent to 100 cells of each cell type for a total of 200-cells worth of input; the BMDM analyses followed a similar design with digest from 100 untreated and 100 LPS-stimulated cells being combined in a single 200-cell carrier sample (126C). This information has been included in the method section.

The impact of carrier size on single-cell quantitation has been investigated in several publications. They concluded that when the AGC threshold is set sufficiently high so that MS2 scans do not terminate before their full fill times, single-cell reporter-ion quantitation is not strongly affected by the carrier. We designed our experiments with this consideration in mind, making sure that out fill times are not affected by the carriers.

In terms of prioritization, arbitrarily increasing carrier sizes may enable very lowly abundant precursors of interest to be detected at the survey scan level and identified at the MS2-level from the carrier-derived backbone fragments, but single-cell quantitation for these lowly abundant species will be less accurate, as indicated by the reduced quantitative precision for the lowest spike-in level of the peptide spike-in experiment (Figure 3).

We added the following sentences to the discussion section:

“Prioritized analysis increases the flexibility of experimental designs. For example, it makes precursors selected for quantification less dependent on the composition of the isobaric carrier, which can be particularly advantageous when the carrier material does not match perfectly the analyzed single cells. In such cases, pSCoPE can be used to analyze the relevant proteins even if they are not among the most abundant proteins in the isobaric carriers used.”

- b) The authors mention a 50-cell carrier for the U937 vs. HEK293 experiment, was this same channel arrangement used for subsequent analyses (e.g., skipping 127C and 128N channels)? A simple table (such as that found in Table S1) highlighting how macrophage/treated samples were distributed over TMTpro channels would be a very useful guide for future users and should be included.

In general, two reporter-ion channels are skipped between the carrier and reference channels and the first single-cell channel in SCoPE experiments to reduce the impact of isotopic impurities from the TMT tags used for the carrier and reference channels contaminating single-cell quantitation. For the BMDM sample analyses, which were conducted at a resolving power of 140k, we noticed isotopic contamination from the carrier channel in the 129N channel (R15), so no single-cell samples in this channel were used in the downstream analysis, and this caveat has been noted in the Methods section.

We have added supplemental tables containing the experimental design for the single-cell channels of the BMDM experiments to the supplemental section.

4) In Figure 6, what correlation metric are the authors using? R, R²? Is this Spearman, Pearson? I assume R and Spearman based on the text, but this should be added to the Figure legend if correct.

In Figure 6b, we used the Pearson correlation coefficient. We have added this information to the figure legend, as well as the Methods section.

“The matrix of single cells by batch-corrected unimputed protein abundances was filtered to contain the four MEROPS cleavage products and proteins annotated to the supplied list of marker proteins (either treatment-condition specific or macrophage-polarization specific), and a protein-protein correlation matrix was produced from this filtered matrix, using Pearson's r as the correlation metric.”

a) Why do the authors believe the correlation coefficients (Assuming R) are less than an absolute value of 0.3?

The correlations for some proteins are much larger than 0.3. However, the heatmap reports the median correlation for the entire set of proteins, and the median is tempered by the presence of proteins with divergent correlations. We clarified this in the text.

5) Much of the 'raw' quantitative data is obscured by PCA plots.

a) Adding a small cohort of scatter plots for the untreated/treated single cells vs bulk proteome data would be helpful here to see the actual trends that are mentioned or alluded to through the PC and correlation plots.

Thank you for this recommendation. We have added two panels to Figure 4 that more directly feature the raw data. Fig. 4c compares the protein fold changes estimated from the single-cell samples to those estimated from the bulk samples for a selection of GO terms; a spearman correlation of 0.91 was observed for this comparison.

Fig.4d illustrates the covariation of two subunits of the V-type ATPase across single-cells within each treatment condition, providing a more granular view of the intracondition variability identified by the PCA-based PSEA for proton transport, shown in Fig. 4b. An additional supplemental figure S3 presenting the protein-protein correlations for all quantified subunits of the V-ATPase has also been generated.

6) 'Top tier', rather than 'high priority' still appears several times throughout the manuscript.

All mentions of 'top tier' have been replaced by 'high priority.' Thanks for catching this!

7) Perhaps a minor point, but re-reading this manuscript several times now, calling the proposed method 'prioritized Single Cell Proteomics' seems to limit acknowledgement of the extensibility of the work. I understand the branding of 'SCoPE' but could these methods not be applied to nearly any low input, or even bulk, sample? Lowercase 'p' in front of proteomics methods often also denote phosphoproteomics variations of methods which may confuse some readers.

Indeed, the prioritized analysis method that we developed is generally applicable to proteomics workflows, not limited to single cells. We have made mention of this point to the first paragraph of the discussion section, thanks to your recommendation:

"Additionally, the benefits of prioritization are not limited to the analysis of single-cell samples, but the improvements to consistency of identification and precursor selectivity are applicable to the analysis of low-input and bulk multiplexed samples."

8) What is SQC? The acronym is used throughout the methods section but, as best as I can tell, never defined.

We had failed to include a critical part of the acronym ('QC') in our introduction of the term, but have now fixed this in the 'Standards used for evaluating prioritization, fig 1.' subsection of the Sample Preparation section of the Methods:

"In order to provide a controlled comparison of MaxQuant.Live's default global targeting method and the prioritized sample analysis method shown in Fig. 1, a Standardized TMT-labeled Quality Control sample was used (hereafter abbreviated as an 'SQC sample')."

Thanks for giving us the opportunity to clarify this!

9) What is the difference between the GitHub site for pSCoPE (<https://github.com/SlavovLab/pSCoPE>) and the lab specific link?: scp.slavovlab.net/pSCoPE

a) In general, for long term accessibility, it is better to use (and point people to) public data repositories rather than lab specific ones for data sharing.

We strongly agree that all data and code should be deposited on public repositories and did so. In addition to the public repositories, we provide more descriptions and intermediate data outputs on scp.slavovlab.net/pSCoPE. However, using scp.slavovlab.net/pSCoPE is not necessary for accessing the

raw data and reproducing our analysis. Thanks to your recommendation, we have also made use of Zenodo to archive both the figure-generating code and the data necessary for recreating the figures.

10) On page 7, this sentence is confusing owing to having additional/adding three times in one sentence: "iMBR facilitated the identification and quantification of approximately 170 additional precursors per shotgun experiment compared to 2,595 additional precursors per experiment added by prioritization, Extended Data Fig. 4."

We have edited this sentence for clarity as suggested:

"iMBR facilitated the identification and quantification of approximately 170 additional precursors per shotgun experiment compared to 2,595 precursors per experiment added by prioritization, Extended Data Fig. 4."

Final Decision Letter:

Dear Nikolai,

I am pleased to inform you that your Article, "Prioritized mass spectrometry increases the depth, sensitivity, and data completeness of single-cell proteomics", has now been accepted for publication in Nature Methods. Your paper is tentatively scheduled for publication in our April print issue, and will be published online prior to that. The received and accepted dates will be 24th March 2022 and 27th February 2023. This note is intended to let you know what to expect from us over the next month or so, and to let you know where to address any further questions.

Over the next few weeks, your paper will be copyedited to ensure that it conforms to Nature Methods style. Once your paper is typeset, you will receive an email with a link to choose the appropriate publishing options for your paper and our Author Services team will be in touch regarding any additional information that may be required.

Your paper will now be copyedited to ensure that it conforms to Nature Methods style. Once proofs are generated, they will be sent to you electronically and you will be asked to send a corrected version within 24 hours. It is extremely important that you let us know now whether you will be difficult to

contact over the next month. If this is the case, we ask that you send us the contact information (email, phone and fax) of someone who will be able to check the proofs and deal with any last-minute problems.

If, when you receive your proof, you cannot meet the deadline, please inform us at rjsproduction@springernature.com immediately.

Once your manuscript is typeset and you have completed the appropriate grant of rights, you will receive a link to your electronic proof via email with a request to make any corrections within 48 hours. If, when you receive your proof, you cannot meet this deadline, please inform us at rjsproduction@springernature.com immediately.

Once your paper has been scheduled for online publication, the Nature press office will be in touch to confirm the details.

Content is published online weekly on Mondays and Thursdays, and the embargo is set at 16:00 London time (GMT)/11:00 am US Eastern time (EST) on the day of publication. If you need to know the exact publication date or when the news embargo will be lifted, please contact our press office after you have submitted your proof corrections. Now is the time to inform your Public Relations or Press Office about your paper, as they might be interested in promoting its publication. This will allow them time to prepare an accurate and satisfactory press release. Include your manuscript tracking number NMETH-A48778B and the name of the journal, which they will need when they contact our office.

About one week before your paper is published online, we shall be distributing a press release to news organizations worldwide, which may include details of your work. We are happy for your institution or funding agency to prepare its own press release, but it must mention the embargo date and Nature Methods. Our Press Office will contact you closer to the time of publication, but if you or your Press Office have any inquiries in the meantime, please contact press@nature.com.

If you are active on Twitter, please e-mail me your and your coauthors' Twitter handles so that we may tag you when the paper is published.

Please note that Nature Methods is a Transformative Journal (TJ). Authors may publish their research with us through the traditional subscription access route or make their paper immediately open access through payment of an article-processing charge (APC). Authors will not be required to make a final

decision about access to their article until it has been accepted. Find out more about Transformative Journals

Authors may need to take specific actions to achieve compliance with funder and institutional open access mandates. If your research is supported by a funder that requires immediate open access (e.g. according to Plan S principles) then you should select the gold OA route, and we will direct you to the compliant route where possible. For authors selecting the subscription publication route, the journal's standard licensing terms will need to be accepted, including self-archiving policies. Those licensing terms will supersede any other terms that the author or any third party may assert apply to any version of the manuscript.

To assist our authors in disseminating their research to the broader community, our SharedIt initiative provides you with a unique shareable link that will allow anyone (with or without a subscription) to read the published article. Recipients of the link with a subscription will also be able to download and print the PDF. As soon as your article is published, you will receive an automated email with your shareable link.

Please note that you and your coauthors may order reprints and single copies of the issue containing your article through Nature Portfolio 's reprint website, which is located at <http://www.nature.com/reprints/author-reprints.html>. If there are any questions about reprints please send an email to author-reprints@nature.com and someone will assist you.

Best regards,
Arunima